# Muon Outperforms Adam in Tail-End Associative Memory Learning

**Shuche Wang**[1][*] **Fengzhuo Zhang**[1][*][†] **Jiaxiang Li**[2][*] **Cunxiao Du**[3] **Chao Du**[3]
**Tianyu Pang**[3] **Zhuoran Yang**[4] **Mingyi Hong**[2] **Vincent Y. F. Tan**[1]
[1]National University of Singapore   [2]University of Minnesota
[3]Sea AI Lab   [4]Yale University
shuche.wang@u.nus.edu  fzzhang@u.nus.edu  jasonljx96@gmail.com
cnsdunm@gmail.com  {duchao,tianyupang}@sea.com
zhuoran.yang@yale.edu  mhong@umn.edu  vtan@nus.edu.sg

## Abstract

The Muon optimizer is consistently faster than Adam in training Large Language Models (LLMs), yet the mechanism underlying its success remains unclear. This paper demystifies this mechanism through the lens of associative memory. By ablating the transformer components optimized by Muon, we reveal that the associative memory parameters of LLMs, namely the Value and Output (VO) attention weights and Feed-Forward Networks (FFNs), are the primary contributors to Muon's superiority. Motivated by this associative memory view, we then explain Muon's superiority on real-world corpora, which are intrinsically heavy-tailed: a few 'head' classes are extremely frequent, while a vast number of 'tail' classes are individually rare. The superiority is explained through two key properties: (i) its update rule consistently yields a more isotropic singular spectrum than Adam; and as a result, (ii) on heavy-tailed data, it optimizes tail classes more effectively than Adam. Beyond empirical evidence, we theoretically confirm these findings by analyzing a one-layer associative memory model under class-imbalanced data. We prove that Muon consistently achieves balanced learning across classes regardless of feature embeddings, whereas Adam can induce large disparities in learning errors depending on embedding properties. In summary, our empirical observations and theoretical analyses reveal Muon's core advantage: its update rule aligns with the outer-product structure of linear associative memories, enabling more balanced and effective learning of tail classes in heavy-tailed distributions than Adam.

## 1 Introduction

The effectiveness of Adam (Kingma & Ba, 2015) across diverse training scenarios has made it one of the most widely used optimizers for neural networks, serving as a cornerstone of the tremendous successes of Large Language Models (LLMs). Building on this foundation, Muon (Jordan et al., 2024) has emerged as a matrix-parameter optimizer designed to surpass Adam. Empirical studies show that Muon is nearly 2 times faster than Adam across a wide range of model sizes and architectures (Liu et al., 2025; Jordan et al., 2024). Its key innovation is to replace the raw gradient with the sum of its normalized orthogonal factors, which can be interpreted as performing steepest descent with respect to the spectral norm (Bernstein & Newhouse, 2024).

However, despite its empirical success, a rigorous understanding of why and how Muon outperforms Adam in transformers remains incomplete. In particular, the steepest gradient descent interpretation does not clarify why optimization with respect to the spectral norm, as in Muon, should outperform optimization with respect to the infinity norm (for vectors), as in Adam. Consequently, convergence analyses of Muon derived from this interpretation fail to account for its observed superiority over Adam (Li & Hong, 2025; Shen et al., 2025).

---

[*]Equal contribution.
[†]Project Lead. Fengzhuo Zhang conducted this work during his internship at Sea AI Lab.

This paper takes the first step toward understanding the mechanisms underlying Muon's superiority over Adam in training LLMs. Specifically, we ask the following two questions:

> *1. Which transformer components benefit most from Muon's matrix-norm–based optimization compared to Adam?*
>
> *2. What structural features of the transformer allow Muon to optimize these components more effectively?*

To answer the first question, we apply Muon to different transformer components. Our experiments consistently show that the more rapid convergence of the validation loss using the Muon optimizer compared to Adam is primarily due to the former's focus on the value-output (VO) matrices of the attention mechanism and the Feed-Forward Networks (FFN) blocks. This leads to our first key insight: VO and FFN blocks, which serve as the primary associative memory stores in the model (Geva et al., 2020; Bietti et al., 2023), are the main beneficiaries of Muon's optimization strategy.

Building on this, we address the second question linking Muon's update mechanism to the learning dynamics of associative memory. Prior work suggests that the behavior of these memory components can be modeled as a sum of outer products representing stored facts (Meng et al., 2022a). Since Muon's update assigns equal update magnitudes to each outer product of the gradient corresponding to orthogonal singular directions, we hypothesize that it optimizes associative memories more effectively than Adam because: (i) Muon's spectral normalization procedure balances the rates of learning of these outer products. (ii) Thus, when training on heavy-tailed data (i.e., where a few 'head' classes are extremely frequent, while a vast number of 'tail' classes are individually rare), Muon reduces the dominance of frequent (head) facts and enables more effective learning from infrequent (tail) facts compared to Adam.

We validate these hypotheses through a combination of empirical analysis and theoretical modeling. Empirically, we conduct two experiments. First, we measure the singular value spectra of the weight matrices and show that Muon consistently yields more isotropic representations than Adam, indicating that its normalization prevents spectral energy from concentrating in dominant components. Second, we evaluate the performance of both optimizers on a knowledge-intensive, heavy-tailed task to demonstrate the practical benefit of Muon's more balanced updates: while both optimizers perform well on head classes (frequent in training data), Muon outperforms Adam on tail classes (rare in training data), leading to more stable and uniform convergence.

Theoretically, we focus on a one-layer linear associative memory model to rigorously explain these empirical findings. Under class imbalance in the training data, mimicking a heavy-tailed distribution, we show that Muon maintains balanced learning across classes, regardless of the feature embeddings. In contrast, we prove that Adam's performance is unstable and strongly dependent on the embedding structure, which can lead to large disparities in learning error across classes. By closely examining the parameter updates, we find that the singular spectrum of weight matrices trained by Muon is nearly isotropic, whereas Adam's is uneven.

Summarizing the empirical and theoretical findings, we identify a clear mechanism underlying Muon's superiority: **The Muon update rule is aligned with the outer-product structure of linear associative memories, enabling more balanced and effective learning of tail classes in heavy-tailed distributions as compared with Adam.**

## 2 PRELIMINARIES

**Muon** (Jordan et al., 2024) is an optimizer tailored for matrix parameters that replaces the raw (or momentum) gradient with the sum of its *normalized orthogonal factors*, producing a scale-invariant, norm-controlled update direction. For a weight matrix $W \in \mathbb{R}^{m \times n}$ at step $t$, let $G_t = \nabla_W \mathcal{L}(W_t)$ denote its gradient. Muon maintains a momentum accumulator of gradients as $B_t = \mu B_{t-1} + G_t$ with $B_0 = 0$, and $\mu \in [0, 1)$. At each step, Muon computes the Singular Value Decomposition (SVD) of $B_t$ as $B_t = U_t S_t V_t^\top$ with $U_t \in \mathbb{R}^{m \times r_t}$, $V_t \in \mathbb{R}^{n \times r_t}$, where $r_t = \text{rank}(B_t)$, and form the nearest (semi)–orthogonal matrix $O_t = U_t V_t^\top$. Then Muon updates the parameter as $W_{t+1} = W_t - \eta_t O_t$. In practice, one can approximate $O_t$ using a fixed number (e.g., 5) of Newton–Schulz iterations applied to $B_t (B_t^\top B_t)^{-1/2}$, which avoids the full SVD while preserving the scale normalization effect. Detailed introduction of Muon is in the related works section (Appendix C).

**Transformers** serve as the backbone of LLMs, predicting the probability of the next token given a sequence of $N$ tokens. A sequence of $N$ tokens is embedded into a matrix $X^{(0)} \in \mathbb{R}^{d \times N}$. The first layer takes $X^{(0)}$ as the input, and each subsequent layer takes the previous layer's output as its input. Every layer $\ell \in [L]$ processes its input through two sequential components: an attention module and a FFN module. The attention module computes

$$H^{(\ell)} = X^{(\ell-1)} + \sum_{h=1}^{H} W_{O,h}^{(\ell)} W_{V,h}^{(\ell)} X^{(\ell-1)} \operatorname{sm}\big(X^{(\ell-1),\top} W_{K,h}^{(\ell),\top} W_{Q,h}^{(\ell)} X^{(\ell-1)}\big), \qquad (2.1)$$

where $\operatorname{sm}(\cdot)$ is the column-wise softmax operator, $H$ is the number of attention heads, $W_{Q,h}^{(\ell)}, W_{K,h}^{(\ell)} \in \mathbb{R}^{d_k \times d}$ capture token relationships, and $W_{V,h}^{(\ell)} \in \mathbb{R}^{d_v \times d}, W_{O,h}^{(\ell)} \in \mathbb{R}^{d \times d_v}$ apply linear transformations. The feed-forward module then updates the representation as

$$X^{(\ell)} = H^{(\ell)} + \operatorname{ff}(H^{(\ell)}, W_{\text{in}}^{(\ell)}, W_{\text{out}}^{(\ell)}) = H^{(\ell)} + W_{\text{out}}^{(\ell)} \sigma(W_{\text{in}}^{(\ell)} H^{(\ell)}), \qquad (2.2)$$

where $\sigma(\cdot)$ is the element-wise activation function, and $W_{\text{in}}^{(\ell)} \in \mathbb{R}^{d_f \times d}, W_{\text{out}}^{(\ell)} \in \mathbb{R}^{d \times d_f}$ are learnable parameters. A gated variant replaces the standard form with

$$\operatorname{ff}_{\text{gate}}(H^{(\ell)}, W_{\text{in}}^{(\ell)}, W_{\text{out}}^{(\ell)}, W_{\text{gate}}^{(\ell)}) = W_{\text{out}}^{(\ell)} \big(\sigma(W_{\text{in}}^{(\ell)} H^{(\ell)}) \odot (W_{\text{gate}}^{(\ell)} H^{(\ell)})\big),$$

where $\odot$ is the Hadamard product, and $W_{\text{gate}}^{(\ell)} \in \mathbb{R}^{d_f \times d}$ is an additional mapping. After $L$ layers, the final hidden state of the last token, $X_{-1}^{(L)}$, is projected by the language model head $E_{\text{head}} \in \mathbb{R}^{K \times d}$ to produce logits $E_{\text{head}} X_{-1}^{(L)}$, which has a vocabulary of size of $K$.

**Associative memory** refers to architectures that store and retrieve patterns based on learned associations between inputs and outputs. Recent research has examined *linear* associative memory in LLMs. Specifically, consider a triplet $(s, r, o)$, where $s$ is the subject, $r$ the relation, and $o$ the object (e.g., $s =$"The United Nations headquarters", $r =$"is located in", $o =$"New York City"). A linear associative memory $W$ maps a key vector $e_s$ encoding $(s, r)$ to a value vector $e_o$ encoding $o$, such that $e_o = W e_s$ holds for all possible $(s, r, o)$. Under the orthogonality of embeddings $e_s$ and $e_o$, $W$ can be expressed as $W = \sum_i e_{o_i} e_{s_i}^\top$, where the summation is taken over the indexes of facts. These facts naturally emerge in the token association in the pretraining data, e.g., the coappearance of "SpaceX" and "Elon Musk", and are learned by LLMs in the form of associative memories. Prior work has investigated associative memory in both attention and FFN modules. In the attention module, Bietti et al. (2023) showed that the parameter $W_O$ can serve as a linear associative memory when $W_V$ is fixed. Since $W_O$ and $W_V$ play symmetric roles, we also treat $W_V$ as part of the associative memory parameters. It is therefore natural to consider VO jointly: several works (Lin et al., 2024; Wang et al., 2025) have shown that the value and output matrices play similar roles and can be analyzed together in practice, even in multi-query attention (MQA) and grouped-query attention (GQA) settings. In FFN, works on knowledge editing (Geva et al., 2020; Dai et al., 2021; Meng et al., 2022a;b) have identified the module as functioning as an associative memory, which can be well approximated by linear associative memory models. In fact, they demonstrate that we can manually update the knowledge in Large Language Models (LLM)s using least squares on the FFN parameters (Meng et al., 2022a;b; Fang et al., 2024). Thus, throughout this paper, we refer to $W_O, W_V$, and FFN in LLMs as the *associative memory parameters*.

## 3 MAIN RESULTS

### 3.1 ASSOCIATIVE MEMORIES ARE MAIN BENEFICIARIES OF MUON

In this section, we identify the transformer components that benefit most from Muon by measuring validation loss on the FineWeb dataset using a 160M NanoGPT model. We adopt a two–stage protocol. First, in the "Independent Blocks" setting, we apply Muon to a single block at a time while keeping all other blocks on Adam, covering the attention projections $W_Q, W_K, W_V, W_O$ and the feed-forward matrices $W_{\text{in}}, W_{\text{out}}$. Second, in the "Combined Configurations" setting, we apply Muon to the most impactful subsets identified in the first stage to examine whether a partial application can recover the performance gains of full Muon. As introduced in Section 2, we evaluate both gated and non-gated FFN variants of NanoGPT. The experimental details are in Appendix F.

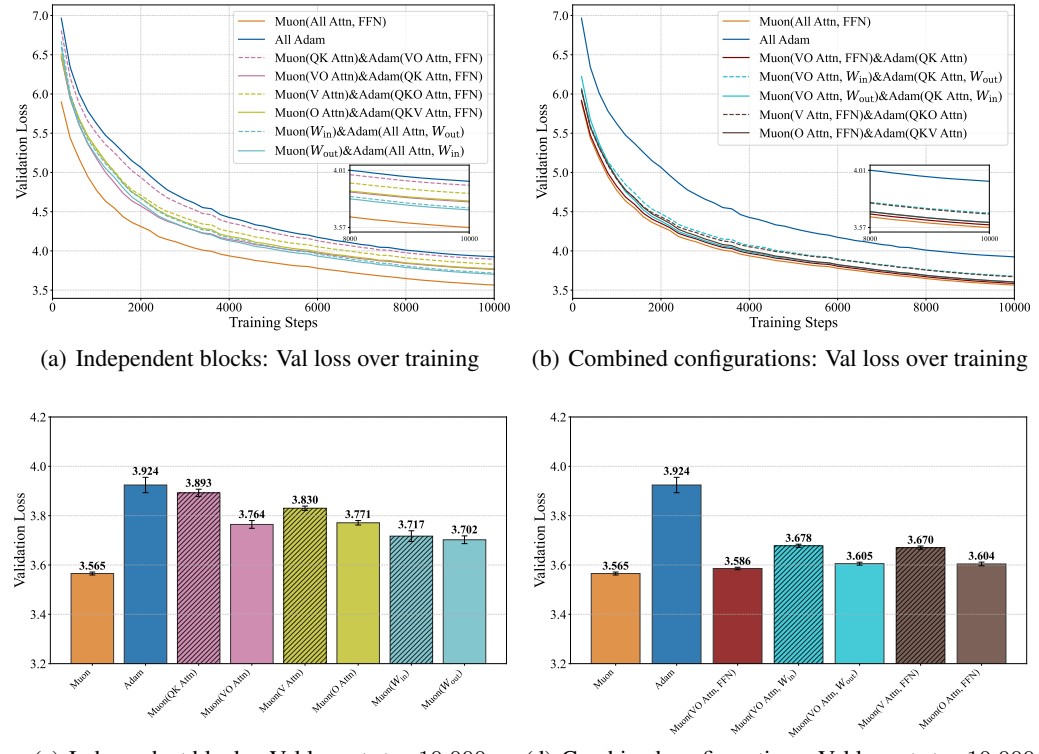

(a) Independent blocks: Val loss over training

(b) Combined configurations: Val loss over training

(c) Independent blocks: Val loss at step 10,000

(d) Combined configurations: Val loss at step 10,000

Figure 1: Validation loss comparison on the 160M NanoGPT model with non-gated FFN under different Muon/Adam assignments. Panels (a) and (b) show the validation loss over training steps for the Independent Blocks and Combined Configurations settings, respectively. Panels (c) and (d) report the corresponding validation loss at step 10,000 for each mode, summarizing the final performance of the Independent Blocks and Combined Configurations.

Figure 1 presents our results. We first examine the independent-block experiments for attention. From Figures 1(a) and 1(c), the VO weights $W_V, W_O$ (Muon on VO / Adam on QK and FFN) show substantially larger gains under Muon than the QK weights $W_Q, W_K$ (Muon on QK/Adam on VO and FFN). Notably, applying Muon to only $W_V$ or only $W_O$ already yields much larger gains than applying it to QK. For the FFN, we find that $W_{in}$, $W_{gate}$, and $W_{out}$ all benefit from Muon, with $W_{out}$ yielding stronger improvements than $W_{in}$. As we show in Appendix G.2, these trends persist even after controlling for parameter count.

After identifying the importance of each module, the combined configurations aim to quantify their contributions to the full Muon. Guided by the independent-block findings, we first observe that VO+FFN already closely tracks—and in our runs nearly recovers—the full-Muon trajectory in Figure 1(b). This indicates that applying Muon to QK contributes little to its overall performance. Importantly, this effect is not due to the logit explosion reported by Team et al. (2025) in large Mixture of Experts (MoE) models; logit values for our setting do not explode, as reported in Appendix G.1. The small remaining gap between full Muon and VO+FFN may stem from the fact that VO+FFN adopts the same learning rate as full Muon without further tuning.

To isolate the contributions of $W_O$ and $W_V$ within VO+FFN, we perform ablations starting from the VO+FFN setting: we keep Muon on FFN and on only one of $W_O$ or $W_V$, reverting the other to Adam (i.e., V+FFN and O+FFN). Both ablations degrade performance, with the V+FFN variant dropping more, indicating that $W_O$ is more influential than $W_V$. Overall, applying Muon to VO+FFN is critical for recovering full-Muon performance. The same qualitative patterns hold for the gated FFN variant reported in Appendix G.3, and are further confirmed on a larger 0.7B model in Appendix G.4, demonstrating the robustness of our findings.

> **Observation 1:** Muon is most effective when applied to VO and FFN; in particular, applying Muon to only VO+FFN almost recovers the full-Muon trajectory.

**Remark 3.1.** We emphasize that this observation is not a trivial consequence of parameter counting; although QK and VO are equal in size, VO proves substantially more influential.

As introduced in Section 2, prior works discover that the common role of VO and FFN is that they both serve as the associative memories for transformers, which store facts and knowledge. Furthermore, Bietti et al. (2023) and Meng et al. (2022a) show that the linear associative memories well approximate them. Specifically, for a set of facts represented by key-value pairs $\{(e_{s_i}, e_{o_i})\}$, the memory matrix $W$ can be constructed as a sum of outer products, i.e., $W = \sum_{i=1}^{K} e_{o_i} e_{s_i}^\top$, where the summation is taken over the index $i$ of $K$ facts. To make this more concrete, consider a toy example with two orthogonal facts in $\mathbb{R}^2$:

- Fact 1: ("the capital of France") $e_{s_1} = [1, 0]^\top$, ("Paris") $e_{o_1} = [1, 0]^\top$.
- Fact 2: ("the capital of Italy") $e_{s_2} = [0, 1]^\top$, ("Rome") $e_{o_2} = [0, 1]^\top$.

The resulting memory matrix is $W = e_{o_1} e_{s_1}^\top + e_{o_2} e_{s_2}^\top = I_{2,2}$ which correctly stores these facts since $W e_{s_i} = e_{o_i}$ for $i = 1, 2$.

Learning linear associative memories is particularly well-suited to Muon's update mechanism. Concretely, the gradient $G \in \mathbb{R}^{d \times d}$ of the loss with respect to the linear associative memory weight $W$ can be expressed as a sum of outer products via SVD as $G = USV^\top = \sum_{i=1}^{d} s_i u_i v_i^\top$. Muon computes its update (without momentum) by normalizing away the singular values, forming the orthogonal factor $O = UV^\top = \sum_{i=1}^{d} u_i v_i^\top$. Following the toy example, consider training the memory parameter $W$ with $\ell_2$ loss, i.e., $c_1 \|e_{o_1} - W e_{s_1}\|^2 + c_2 \|e_{o_2} - W e_{s_2}\|^2$, where $c_1, c_2 > 0$ represent the importance or frequency of each fact in the current training batch. The corresponding gradient is $G = c_1 \cdot e_{o_1} e_{s_1}^\top + c_2 \cdot e_{o_2} e_{s_2}^\top = \mathrm{diag}(c_1, c_2)$. Consequently, Muon's normalized update factor becomes $O = UV^\top = I_{2,2} = e_{o_1} e_{s_1}^\top + e_{o_2} e_{s_2}^\top$, which is simply the sum of the constituent facts' outer products. Crucially, the update $O$ assigns equal weight to both Fact 1 and Fact 2, regardless of their original coefficients $c_1$ and $c_2$ in the gradient. This illustrates how Muon normalizes the updates across orthogonal facts, allowing it to learn both frequent (large $c_1$) and infrequent (small $c_2$) facts uniformly. Comparing this with the linear associative memory $\sum_{i=1}^{K} e_{o_i} e_{s_i}^\top$, we see that Muon updates all "orthogonal" facts at the same rate. Later, we will see that the singular values $S$ of the gradient $G$ encode the frequencies of knowledge in the training data under cross-entropy loss in Sections 3.3 and 4. By normalizing away $S$ to form its update, Muon can therefore learn both frequent and infrequent facts more uniformly than gradient-magnitude-based optimizers like Adam.

We verify this insight from two perspectives. First, from the view of weight spectra, the weight matrices learned with Muon exhibit a more isotropic singular-value spectrum than those learned with Adam, indicating that knowledge, regardless of its frequency, is represented with comparable magnitude. Second, at the level of overall knowledge acquisition, Muon yields more balanced learning across entities and frequencies (head and tail) than Adam. We examine these two consequences in the following sections.

### 3.2 MUON CONSISTENTLY LEARNS MORE ISOTROPIC WEIGHTS THAN ADAM

To validate that Muon can shape the weight matrices more evenly across directions, we conducted a spectral analysis of them. For a weight matrix with $n$ non-zero singular values $\sigma = (\sigma_1, \sigma_2, \dots, \sigma_n)$, we define the normalized singular energy distribution $q = (q_1, q_2, \dots, q_n)$, where each component $q_i$ is $q_i = \sigma_i^2 / \sum_{j=1}^{n} \sigma_j^2$. This distribution represents the fraction of energy captured by each corresponding singular vector. Based on this, we introduce several metrics to characterize the isotropy of the spectrum: normalized SVD entropy defined as $H_{\mathrm{norm}}(\sigma) = -\frac{1}{\log n} \sum_{i=1}^{n} q_i \log q_i$, effective rank defined as $\mathrm{eRank}(\sigma) = \exp\left(-\sum_{i=1}^{n} q_i \log q_i\right)$, Top-$k$ energy fraction defined as $\mathrm{TopE}_k(\sigma) = \sum_{i=1}^{k} \sigma_i^2 / \sum_{j=1}^{n} \sigma_j^2$, and eigenvalue quantile ratio defined as $\{\sigma_i^2\}_{i=1}^{n} : Q_{75/25}(\sigma) = Q_3(\{\sigma_i^2\}) / Q_1(\{\sigma_i^2\})$. Detailed explanations of these metrics are in Appendix F.2. Intuitively, more isotropic weights correspond to larger values of normalized SVD entropy and effective rank, and smaller Top-$k$ energy fraction and eigenvalue quantile ratio.

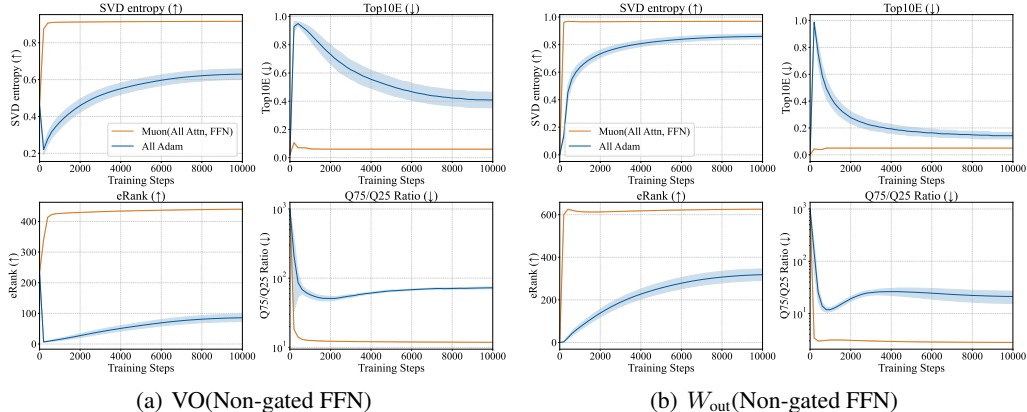

(a) VO(Non-gated FFN)     (b) $W_{\text{out}}$(Non-gated FFN)

Figure 2: Spectral Dynamics of Transformer Weight Matrices During Training. Each panel reports four metrics characterizing singular value distributions: SVD entropy, Top10E, eRank, and Q75/Q25 ratio. The four subplots correspond to different weight matrix groups: (a) VO and (b) $W_{\text{out}}$.

The spectral analysis in Figure 2, focusing on the key associative memory components from Observation 1, shows that Muon systematically reshapes the learned weight matrices relative to Adam. The results, averaged over 10 random seeds, demonstrate that: (i) Muon produces a much more isotropic singular spectrum than Adam from the start of training, whereas Adam's isotropy fluctuates significantly over the course of optimization. (ii) The isotropy of Muon is stable across random initializations, as indicated by the negligible error bars in Figure 2, while Adam is more sensitive to initialization. These findings suggest that Muon consistently promotes richer and more diverse features in the model's most critical memory components, a conclusion we summarize below. The results for the gated FFN architecture and other weights are in Appendix G.3 and G.5, respectively.

> **Observation 2:** Muon consistently yields more isotropic weight matrices with broadly distributed spectral energy than Adam, both throughout training and across random initializations, thereby supporting richer feature representations.

Empirically, we also find that Muon learns more isotropic QK weights than Adam. However, as discussed in Section 3.1, QK weights are not part of the linear associative memory mechanism and are therefore not expected to benefit from the isotropic property of the weight matrices.

Our results differ fundamentally from the spectral analysis in Liu et al. (2025) for three reasons. First, we decompose the parameters according to associative memories, whereas Liu et al. (2025) aggregates them, obscuring the essential components driving Muon's behavior. Second, we investigate the instability of Adam under random initialization (i.e., random seeds), which we further establish theoretically in Section 4. Finally, our analysis focuses on dense architectures, while Liu et al. (2025) centers on Mixture-of-Experts (MoE) models.

### 3.3 MUON ACQUIRES KNOWLEDGE MORE EVENLY COMPARED TO ADAM

Our previous findings indicate that the Muon optimizer is particularly important for the associative memory components of the model, where it learns more isotropic weights. To examine the overall effects of learning associative memories, we turn to a knowledge-intensive question-answering (QA) task. The task is based on a synthetic QA dataset containing biographical information (e.g., name, birthday, and company) for over 200,000 individuals (Allen-Zhu & Li, 2024). To capture the heavy-tailed nature of real-world knowledge, we control the frequency of each individual's appearance in the training set so that it follows a power-law distribution (Figure 3(a)), thereby inducing varying levels of difficulty in learning knowledge about different individuals. A 160M NanoGPT model is trained to answer questions about this biographical information. The performance is evaluated via the First Token Accuracy (FTA) on the answers, following Allen-Zhu & Li (2024). Further details on the dataset are provided in Appendix F.3. We include SGD as a baseline for Adam and Muon.

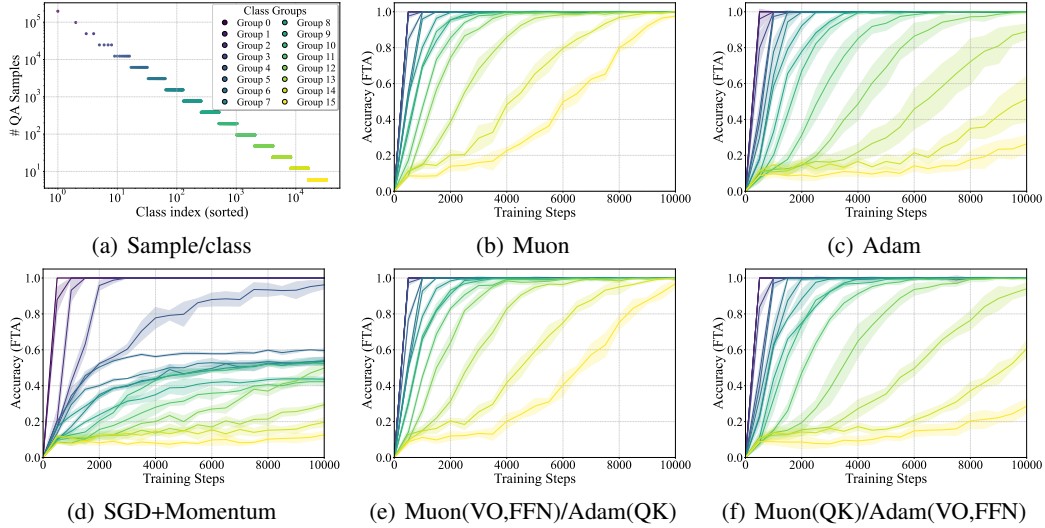

Figure 3: Performance comparison of different optimizers for transformers with non-gated FFN on a heavy-tailed knowledge task. (a) Sample distribution per class, following a power law. (b–d) Performance of Muon, Adam, and SGD+Momentum. (e) Muon applied to VO and FFN, with Adam on QK. (f) Muon applied to QK, with Adam on VO and FFN.

The results in Figure 3 lead to an unequivocal conclusion about the efficacy of different optimizers under data imbalance. In high-frequency (head) classes, all optimizers perform well, with Muon, Adam, and even SGD+Momentum rapidly reaching near-perfect accuracy (Figure 3(b–d)). Consistent with prior work on heavy-tailed distributions (Kunstner et al., 2024), Adam maintains a clear advantage over SGD, which struggles with tail classes. Our key finding, however, is that Muon substantially outperforms Adam on low-frequency (tail) data, achieving faster and more uniform convergence across all frequencies. Moreover, the consistently tighter error bars for Muon—especially relative to Adam—reflect lower variance and a more stable learning process.

Furthermore, the hybrid configurations in Figure 3(e–f) clarify where Muon matters most. Applying Muon to VO+FFN (with QK on Adam) yields strong gains on rare classes and markedly reduces the head–tail gap, whereas applying Muon only to QK (with VO+FFN on Adam) yields only limited improvement. This mirrors Observation 1: VO+FFN is the most effective target set, as it concentrates the model's associative memory. Results for the gated FFN, which show the same pattern, are provided in Appendix G.7. Additional experiments in Appendix G.8 vary the degree of fact imbalance, and show that the average FTA gap between Muon and Adam shrinks as the data distribution becomes more uniform. Together with the Wikitext103 results in Appendix G.9, which exhibit the same qualitative behavior on a standard language modeling benchmark, these findings further support the view that Muon's advantage is tightly linked to heavy-tailed imbalance. We summarize these findings as Observation 3.

**Observation 3:** In heavy-tailed, knowledge-intensive tasks, Muon matches Adam's strong performance in the head classes while substantially improving learning on tail classes, narrowing the head-tail gap and accelerating convergence.

In addition to the knowledge acquisition task, whose success primarily depends on learning the associative-memory parameters (VO and FFN), we also evaluate an in-context linear regression task in Appendix G.10, which primarily depends on learning the QK parameters. In contrast to the above observation, Muon achieves performance on the tail class similar to that of Adam in this task. This is consistent with Observation 1, which indicates that the QK parameters are not the main source of Muon's superiority.

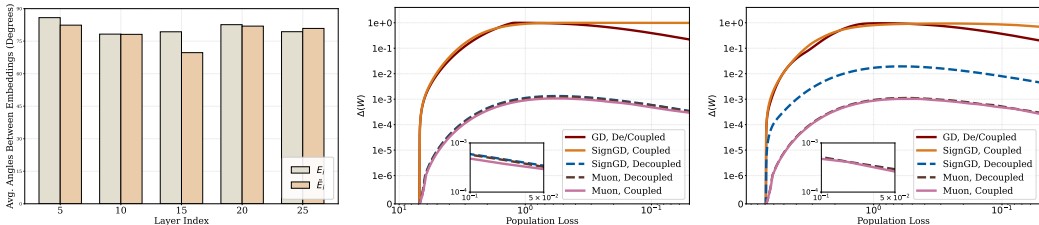

(a) Average Angles Between $E_i/\widetilde{E}_i$    (b) One-step Optimization Results    (c) Multi-step Optimization Results

Figure 4: (a) Average angles between $E_i$ or $\widetilde{E}_i$ in FFN at layers 5, 10, 15, 20, 25 of Llama3-8b-instruct. (b) Results of one-step GD, SignGD, and Muon with both coupled and decoupled embeddings. For GD, the outcomes under the two embedding types coincide. We use different step sizes to obtain different levels of population loss and $\Delta(W)$.(c) Results of multi-step GD, SignGD, and Muon with both coupled and decoupled embeddings. We use different numbers of optimization steps to obtain different levels of population loss and $\Delta(W)$.

## 4   CASE STUDY OF ONE-LAYER MODELS

We now analyze three optimizers—Adam, Muon, and Gradient Descent (GD) (as a baseline)—to complement the preceding empirical observations. We first introduce an abstraction that captures their key dynamics and then present both empirical and theoretical results. As shown in Eqns. (2.1) and (2.2), a structural property of associative memory parameters is that their output is added directly to the hidden states, which are subsequently processed by the language model head. Motivated by this property, our abstraction retains the associative memory and language model head, while replacing all preceding modules with given feature embeddings.

Consider $K$ triplets $\{(s_i, r_i, o_i)\}_{i=1}^{K}$ (introduced in Section 2), where subject-relation pairs $(s_i, r_i)$ and objects $o_i$ are embedded into the columns of matrices $E \in \mathbb{R}^{d_s \times K}$ and $\widetilde{E} \in \mathbb{R}^{d_o \times K}$, respectively. A linear associative memory $W \in \mathbb{R}^{d_o \times d_s}$ predicts the object for a query $E_k$ with probabilities $f_W(E_k) = \mathrm{sm}(\widetilde{E}^\top W E_k) \in \mathbb{R}^K$. The objective is to minimize the population cross-entropy loss $\mathcal{L}(W) = -\sum_{k=1}^{K} p_k \log[f_W(E_k)]_k$, where $p_k$ is the frequency or probability of the $k$-th triplet. We analyze three optimizers initialized at $W_0 = 0$, all simplified by disabling momentum for clarity. **(i) GD:** $W_{t+1} = W_t - \eta \nabla_W \mathcal{L}(W_t)$. **(ii) Adam:** Following prior work (Kunstner et al., 2024; Bernstein & Newhouse, 2024), we set $\beta_1 = \beta_2 = 0$, reducing it to SignGD: $W_{t+1} = W_t - \eta \, \mathrm{sign}(\nabla_W \mathcal{L}(W_t))$. **(iii) Muon:** The update is $W_{t+1} = W_t - \eta U_t V_t^\top$, where $U_t \Sigma_t V_t^\top$ is the SVD of $\nabla_W \mathcal{L}(W_t)$. This simplified form, $U_t V_t^\top$, is the projection of the gradient onto the nearest orthogonal matrix. We then state the assumptions for our results.

**Assumption 4.1.** The embeddings $E$ and $\widetilde{E}$ are orthonormal, i.e., $E^\top E = \widetilde{E}^\top \widetilde{E} = I_{K,K}$.

The unit-norm requirement rules out feature-level imbalance, which would otherwise couple with the imbalance induced by $p_k$ and complicate the analysis. Our techniques can be directly applied even without this unit-norm requirement. The orthogonality assumption is intuitively plausible, as different concepts are independent and do not influence one another. We empirically verify this on Llama3-8b-instruct (Dubey et al., 2024). Following Fang et al. (2024), we extract $E_i$ and $\widetilde{E}_i$ in FFN across layers for $3,000$ knowledge items of Counterfact (Meng et al., 2022a) and compute average angles between them (see Appendix F.4 for details). As shown in Figure 4(a), these angles are near $90°$, confirming approximate orthogonality. For $K$ independent concepts, orthogonality requires $d_r, d_s \geq K$. For simplicity, we set $d_r = d_s = K$ in what follows.

**Assumption 4.2.** The first $L$ triplets share the same probability and together contribute a total mass of $\alpha$, i.e., $p_k = \alpha/L$ for $k \in [L]$. The remaining triplets also share the same probability and together contribute a total mass of $1 - \alpha$, i.e., $p_k = (1 - \alpha)/(K - L)$ for $k > L$.

This assumption states that the data imbalance is between two classes among the $K$ triplets. Defining $\beta = L/K$, the ratio $\alpha/\beta$ quantifies the degree of balance: if $\alpha > \beta$, the first $L$ triplets appear more frequently during learning, and vice versa. This simplified two-class setting is sufficient to capture

the primary differences between optimizers; the multi-class case follows directly from our proof by extending the SVD calculation.

Throughout Section 4 we will also refer to the *imbalance ratio*, defined as the ratio between the minimal and maximal frequencies of triplets, i.e., $r := \frac{\min_{k \in [K]} p_k}{\max_{k \in [K]} p_k} \in (0, 1]$. Under Assumption 4.2 with parameters $\alpha$ and $\beta = L/K$, this reduces to $r(\alpha, \beta) = \min\left\{\frac{\alpha(1-\beta)}{\beta(1-\alpha)}, \frac{\beta(1-\alpha)}{\alpha(1-\beta)}\right\}$. We keep the two-mass $(\alpha, \beta)$ parametrization because it allows us to write the gradient and its SVD in closed form while capturing the same dependence on class imbalance as using $r$ directly; the multiclass case follows from the same SVD calculation.

## 4.1 EXPERIMENTAL RESULTS

Under Assumptions 4.1 and 4.2, we evaluate GD, SignGD, and Muon for $\alpha = 0.8$, $\beta = 0.2$, considering two embeddings for $E$ and $\widetilde{E}$: (i) support-decoupled: the supports (indices of non-zero entries) of different $E_i$ or $\widetilde{E}_i$ are disjoint; (ii) support-coupled: supports may overlap. We study two optimization protocols, initializing $W_0 = 0_{d_o \times d_s}$: (i) one-step: take a single update with a scaled step size to obtain a range of $\mathcal{L}(W)$ values; (ii) multi-step: run multiple updates to reduce $\mathcal{L}(W)$, varying the number of steps. Experimental details are in Appendix F.5. To quantify *learning imbalance* across $K$ knowledge items, we examine the relationship between population loss $\mathcal{L}(W)$ and *maximal probability gap* $\Delta(W) := \max_{i,j \in [K]} [f_W(E_i)]_i - [f_W(E_j)]_j$, where $[f_W(E_i)]_i$ denotes the probability assigned to the correct item $i$. A larger $\Delta(W)$ indicates greater imbalance.

Across both optimization-step protocols and embeddings (Figures 4(b), 4(c)), we observe that (i) For all optimizers, $\Delta(W)$ first *increases* and then *decreases* as $\mathcal{L}(W)$ decreases. Early in training, when correct probabilities are near 0, imbalance is pronounced; later, when all items are well learned (e.g., probabilities $\geq 0.9$), imbalance diminishes. (ii) For both embedding regimes, GD and Muon behave consistently: GD exhibits a substantial imbalance, whereas Muon remains much more balanced across items. (iii) SignGD also demonstrates unstable behavior; its imbalance resembles GD in the coupled embedding case and Muon in the decoupled embedding case.

Because one-step and multi-step experiments align qualitatively, we first analyze the **one-step** setting for clarity. This simplification is common in theoretical studies of neural network dynamics (Ba et al., 2022; Dandi et al., 2023), and our techniques extend directly—albeit with more algebra—to the multi-step case. As a demonstration, Theorem 4.4 provides a multi-step analysis of Muon.

## 4.2 THEORETICAL RESULTS

For each optimizer, we choose a step size $\eta$ so that *some* class already attains correct-class probability at least $1 - \epsilon$ after one update, and then we report the *smallest* correct-class probability across classes at the same $\eta$. Equation 4.1 formalizes this procedure.

$$\varrho_{\text{opt}}^\epsilon = \inf_{\eta \geq 0} \left\{ \min_{k \in [K]} [f_{W_\eta}(E_k)]_k \;\middle|\; \max_{k \in [K]} [f_{W_\eta}(E_k)]_k \geq 1 - \epsilon, \; W_\eta = W_0 - \eta \cdot G_{\text{opt}}(W_0) \right\}. \quad (4.1)$$

where opt $\in \{\text{GD}, \text{SignGD}, \text{Muon}\}$ and $G_{\text{opt}}(W_0)$ denotes the parameter update of optimizer opt at $W_0$; and $W_\eta$ denotes the parameter obtained after one step of optimizer opt with step size $\eta$ starting from $W_0$, i.e., $W_\eta = W_0 - \eta \cdot G_{\text{opt}}(W_0)$. Specifically, $G_{\text{GD}}(W_0) = \nabla_W \mathcal{L}(W_0)$, $G_{\text{SignGD}}(W_0) = \text{sign}(\nabla_W \mathcal{L}(W_0))$, and $G_{\text{Muon}}(W_0) = U_0 \text{norm}(\Sigma_0) V_0^\top$ where $U_0 \Sigma_0 V_0^\top$ is the SVD of $\nabla_W \mathcal{L}(W_0)$. Note that $\varrho_{\text{opt}}^\epsilon \in [0, 1 - \varepsilon]$ and $\Delta(W)$ are related as $\Delta(W) = 1 - \epsilon - \varrho_{\text{opt}}^\epsilon \geq 0$. When $\varrho_{\text{opt}}^\epsilon \approx 1 - \epsilon$, opt achieves balanced learning across facts; in contrast, when $\varrho_{\text{opt}}^\epsilon \approx 0$, imbalanced learning ensues.

**Theorem 4.3.** Let $r := \min_{k \in [K]} p_k / \max_{k \in [K]} p_k$ (under Assumption 4.2, $r = r(\alpha, \beta)$). If Assumptions 4.1 and 4.2 hold, with fixed $\alpha, \beta$ such that $\alpha \neq \beta$, and $K$ goes to infinity, we obtain the following results for one-step GD, Muon, and Adam.

- **GD**: For any $\widetilde{E}$ and $E$ satisfying Assumption 4.1, we have

$$\varrho_{\text{GD}}^\epsilon = O(\epsilon^{-r(\alpha,\beta)} K^{r(\alpha,\beta)-1}), \text{ where } r(\alpha, \beta) = \frac{\min_k p_k}{\max_k p_k} = \min\left\{\frac{\alpha(1-\beta)}{\beta(1-\alpha)}, \frac{\beta(1-\alpha)}{\alpha(1-\beta)}\right\} < 1.$$

- **Muon**: For any $\widetilde{E}$ and $E$ satisfying Assumption 4.1, we have

$$\varrho_{\mathrm{Muon}}^{\epsilon} \geq 1 - \epsilon\left(1 + O\left(\frac{\log K}{K}\right)\right), \text{ and } G_{\mathrm{Muon}}(W_0) = -\widetilde{E}E^{\top} + O\left(\frac{1}{K}\widetilde{E}J_{K,K}E^{\top}\right),$$

where $J_{K,K} \in \mathbb{R}^{K \times K}$ is the matrix with all elements equal to 1. The big-$O$ notation for matrices means that for $A = O(B)$, each entry satisfies $A_{ij} = O(B_{ij})$ for all $i, j$.

- **Adam**: There exist $\widetilde{E}$ and $E$ satisfying Assumption 4.1 such that $\varrho_{\mathrm{SignGD}}^{\epsilon} \geq 1 - \epsilon$. There also exist $\widetilde{E}'$ and $E'$ satisfying Assumption 4.1 such that

$$\varrho_{\mathrm{SignGD}}^{\epsilon} = O(\epsilon^{-0.7}K^{-0.3}), \text{ and } \frac{\sigma_{\min}\left(G_{\mathrm{SignGD}}(W_0)\right)}{\sigma_{\max}\left(G_{\mathrm{SignGD}}(W_0)\right)} \leq 25\%,$$

where $\sigma_{\max}$ and $\sigma_{\min}$ are the largest and smallest singular values, respectively.

**Interpretation of Theorem 4.3.** *These theoretical results align with Observations 2 and 3, and Figures 4(b) and 4(c): Muon maintains balanced learning with near-isotropic updates, GD is highly sensitive to data imbalance, and Adam varies widely across embeddings.* At the one-step update, when the maximum correct-class probability across items is at least $1 - \epsilon$, the item with the minimum correct-class probability satisfies: (i) Muon: $\geq 1 - \epsilon\left(1 + O(\frac{\log K}{K})\right)$, which indicates learning is essentially balanced across items with a near-isotropic update (singular values nearly equal); (ii) GD: $O\left(\epsilon^{-r(\alpha,\beta)}K^{r(\alpha,\beta)-1}\right)$, which is strongly controlled by data imbalance via $r(\alpha, \beta)$ (balanced when $r = 1$, severe imbalance when $r \ll 1$); (iii) Adam: embedding dependent; it can match Muon with disjoint supports (e.g., $\widetilde{E} = E = I_{K,K}$), achieving $1 - \epsilon$, but can drop to $O(\epsilon^{-0.7}K^{-0.3})$ with overlap; its update may exhibit pronounced spectral decay ($\sigma_{\min}/\sigma_{\max} \leq 25\%$), unlike the near-uniform singular values of Muon. A detailed discussion of Theorem 4.3 is provided in Appendix E.

In the following, we extend our techniques of one-step analysis to the multi-step analysis of Muon. Parallel to Eqn. (4.1), we define the infimum correct-class probability for the multi-step optimizer as $\varrho_{\mathrm{opt}}^{\epsilon} = \inf_t\{\min_{k \in [K]}[f_{W_t}(E_k)]_k \mid \max_{k \in [K]}[f_{W_t}(E_k)]_k \geq 1 - \epsilon$, where $W_t = W_{t-1} - \eta_t \cdot G_{\mathrm{opt}}(W_{t-1})\}$. Here, we assume that the learning rates $\{\eta_t\}_{t \geq 1}$ are determined by a fixed schedule prior to optimization. Although the quantity implicitly depends on this schedule, we omit it from the notation for $\varrho_{\mathrm{opt}}^{\epsilon}$ for brevity. We emphasize that different schedules may affect the value of $t$ that attains the infimum in $\varrho_{\mathrm{opt}}^{\epsilon}$, but they do not influence the balance behavior that we present.

**Theorem 4.4.** If Assumptions 4.1 and 4.2 hold, then multi-step Muon achieves

$$\varrho_{\mathrm{Muon}}^{\epsilon} \geq 1 - \epsilon\left(1 + O\left(\frac{\log K}{K}\right)\right), \text{ and } G_{\mathrm{Muon}}(W_t) = -\widetilde{E}E^{\top} + O\left(\frac{1}{K}\widetilde{E}J_{K,K}E^{\top}\right) \text{ for any } t \geq 0.$$

The proof is provided in Appendix I. We note that the multi-step analysis of Muon shares similar characteristics as the one-step version in Theorem 4.3.

## 5 CONCLUSION

Our work takes the first step toward unveiling why and how Muon outperforms Adam. Through ablations of Muon's effect on different Transformer components and by relating these results to the balanced learning of associative memories, we conclude that the Muon update rule is aligned with the outer-product structure of linear associative memories, enabling more balanced and effective learning of tail classes in heavy-tailed distributions. Intuitively, this property of Muon may extend beyond outer products to higher-order tensor products, an exciting direction for future work.

ACKNOWLEDGMENTS

S. Wang and V. Y. F. Tan are supported by a Singapore Ministry of Education (MOE) AcRF Tier 2 grant under grant number A-8004062-00-00. M. Hong and J. Li are supported partly by NSF grant ECCS-2450988, NSF CIF-2414372, NSF EPCN-2311007.

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

## A    USE OF LARGE LANGUAGE MODELS (LLMS)

Large language models (LLMs) were used solely to aid and polish the writing of this paper. The authors generated all research ideas, methods, analyses, and results independently. LLM assistance was limited to improving clarity, grammar, and readability of the manuscript text. No content was fabricated or introduced by the LLM beyond these language refinements.

## B    NOTATIONS

Let $[N]$ for the set $\{1, \ldots, N\}$. For a matrix $X \in \mathbb{R}^{d \times N}$, $X_i$ is its $i$-th column and $X_{:,-1}$ is its last column. $I_{K,K}$ is the $K \times K$ identity matrix, $\mathbb{1}_K$ is all-ones vector and $J_{K,K}$ is the all-ones matrix. $\odot$ denotes the element-wise product.

## C    RELATED WORKS

**Adam**, proposed by Kingma & Ba (2015), was designed to make GD adaptive to the complex optimization landscape of neural networks. Existing works analyze Adam from two primary perspectives: online optimization and feature learning. The online convex optimization view focuses on Adam's properties when optimizing convex or non-convex loss functions. From this perspective, Chen et al. (2019) and Zhou et al. (2018) derive non-convex convergence results for Adam, and a series of subsequent works continuously relaxed the required assumptions for Adam's convergence while tightening its convergence rate. For instance, Zou et al. (2019) proposes a set of easy-to-verify sufficient conditions for Adam's update rules to guarantee convergence. Défossez et al. (2020) derives the tightest dependency on the heavy ball momentum parameters. More recently, Zhang et al. (2022) demonstrates that Adam can converge without modification of its procedures, and Li et al. (2023) relaxes the smoothness assumption by employing an adaptive Lipschitz constant for gradients. The feature learning view, on the other hand, highlights the relationship between deep learning characteristics and Adam, focusing more on how Adam's mechanisms influence the properties of learned features within deep networks. For example, Pan & Li (2023) examines the sharpness of GD and Adam and relates Adam's superiority to its low sharpness. Kunstner et al. (2024) finds that Adam is better at learning heavy-tailed distributions than GD. Furthermore, Zhang et al. (2024a) shows that Adam is adaptive to heterogeneous Hessian structures, thus optimizing faster than GD. In a spirit similar to our work, recent studies have also used ablation experiments to deconstruct Adam's effectiveness. For instance, Zhao et al. (2024) and Zhang et al. (2024b) conduct detailed ablations on Adam's hyperparameters and components, identifying that its benefits are particularly pronounced for the first and last embedding layers of language models. While these works focus on understanding Adam's existing components, our study applies a similar ablation methodology to understand the impact of a different optimizer, Muon, on the internal modules of a Transformer. More literature on Adam is included in the survey by Abdulkadirov et al. (2023).

**Muon**, proposed by Jordan et al. (2024), applies spectral normalization of the gradient to update parameters. At a high level, Muon can be understood as steepest descent with respect to the matrix operator norm (Bernstein & Newhouse, 2024). Alternatively, it can be viewed as maximizing the feature update subject to a parameter update constraint (Yang et al., 2023). Experiments show that Muon consistently outperforms Adam across diverse model sizes and architectures, including dense transformers and Mixture-of-Experts (Liu et al., 2025; Jordan et al., 2024). Building on this, Si et al. (2025) introduces an adaptive variant of Muon. To explain its advantages, Lau et al. (2025) introduces a unifying preconditioning framework, distinguishing optimizers that address curvature anisotropy (like Adam) from those that address gradient anisotropy (like Muon), and proposes a generalized optimizer class named PolarGrad. Sato et al. (2025) and Shah et al. (2025) examine the critical batch size of Muon, while other works analyze its convergence in convex and non-convex settings (Li & Hong, 2025; An et al., 2025; Kovalev, 2025; Pethick et al., 2025; Shen et al., 2025).Kim & Oh (2026) derives the convergence bound of Muon, including the influence of NS steps. Furthermore, Grishina et al. (2025) proposes accelerating these NS steps via Chebyshev-optimized coefficients. Concurrently, Vasudeva et al. (2025) studies Muon on shallow ViTs for computer vision, grounding their results for gradient descent and Muon in linear regression. In contrast, we investigate Muon in the context of LLMs, focusing on its effects on associative memory in next-token prediction. Recent works have also investigated the scalability of the Muon optimizer.

For instance, Wen et al. (2025) reports that the benefits of Muon diminish with scale (dropping from $1.4\times$ gain at 0.1B to $1.1\times$ at 1.2B), whereas Liu et al. (2025) observes that Muon maintains a $\approx 2\times$ FLOP-efficiency advantage over Adam even on 32B models.

**Associative Memories** have a long history in neural network design and knowledge storage (Hopfield, 1982; Kohonen, 2009; Willshaw et al., 1969). They have inspired architectures capable of retaining long histories, including RNNs (Orvieto et al., 2023) and Mamba (Zhang et al., 2024c). With the success of transformers, recent work has examined them through the lens of associative memories. Geva et al. (2020) and Dai et al. (2021) show that feed-forward modules store knowledge in $W_{\text{out}}$, while Bietti et al. (2023) demonstrates that the attention output matrix $W_O$ also encodes associations of knowledge. Building on these findings, a series of works edit knowledge directly by modifying these weights (Meng et al., 2022b; Fang et al., 2024). Beyond empirical results, theoretical analyses have further clarified how transformers leverage associative memories: Bietti et al. (2023) conducts a dynamic analysis of memory formation, while Nichani et al. (2024) constructs explicit associative memory mechanisms in both attention and feed-forward modules.

## D    STEEPEST DESCENT VIEW UNDERSTANDING MUON AND ADAM

Bernstein & Newhouse (2024) showed that many popular deep learning optimizers can be understood through the unifying framework of *steepest descent*, once their exponential moving averages (EMAs) are disabled. This perspective shifts the focus from heuristic or second-order motivations to a more fundamental, geometric view: the choice of an optimizer is equivalent to choosing a specific *norm* to measure the "size" of the weight update.

**The Steepest Descent Framework.** The core idea is to find a weight update, $\Delta \mathbf{w}$, that minimizes a local quadratic approximation of the loss function. This is formulated as the following optimization problem:

$$\Delta \mathbf{w}^* = \operatorname*{argmin}_{\Delta \mathbf{w}} \left[ \mathbf{g}^\top \Delta \mathbf{w} + \frac{\lambda}{2} \|\Delta \mathbf{w}\|^2 \right],$$

where $\mathbf{g}$ is the gradient of the loss, $\lambda > 0$ is a "sharpness" parameter that controls the step size, and $\|\cdot\|$ is a chosen norm.

The solution to this problem can be expressed as:

$$\Delta \mathbf{w}^* = -\eta \cdot \mathbf{d},$$

where the step size $\eta = \frac{\|\mathbf{g}\|_*}{\lambda}$ and the update direction $\mathbf{d} = \arg\max_{\|\mathbf{t}\|=1} \mathbf{g}^\top \mathbf{t}$. Here, $\|\cdot\|_*$ denotes the *dual norm* of $\|\cdot\|$ (defined as $\|\mathbf{y}\|_* = \sup_{\|\mathbf{x}\|\leq 1} \mathbf{y}^\top \mathbf{x}$). The key insight is that different choices of the norm $\|\cdot\|$ lead to different update directions $\mathbf{d}$, recovering the update rules of well-known optimizers.

**Muon as Steepest Descent under Spectral Norm.** The update rule of the Muon optimizer is derived by applying the steepest descent framework to weight matrices equipped with the *spectral norm*, denoted in the paper as the $\|\cdot\|_{\ell_2 \to \ell_2}$ operator norm (defined as its largest singular value, $\|\mathbf{A}\|_{\ell_2 \to \ell_2} = \sigma_{\max}(\mathbf{A}) = \sup_{\|\mathbf{x}\|_2=1} \|\mathbf{A}\mathbf{x}\|_2$). For a gradient matrix $\mathbf{G}$, the problem is to find the update $\Delta \mathbf{W}$ that solves:

$$\Delta \mathbf{W}^* = \operatorname*{argmin}_{\Delta \mathbf{W}} \left[ \langle \mathbf{G}, \Delta \mathbf{W} \rangle_F + \frac{\lambda}{2} \|\Delta \mathbf{W}\|^2_{\ell_2 \to \ell_2} \right].$$

The solution to this problem is directly determined by the Singular Value Decomposition (SVD) of the gradient, $\mathbf{G} = \mathbf{U}\mathbf{\Sigma}\mathbf{V}^\top$. The resulting update direction, which maximizes alignment with the gradient under the spectral norm constraint, is shown to be $\mathbf{U}\mathbf{V}^\top$. The corresponding dual norm of the gradient, $\|\mathbf{G}\|^*_{\ell_2 \to \ell_2}$, which scales the step size, is found to be $\text{tr}(\mathbf{\Sigma})$, the sum of the singular values. Combining these components yields the final steepest descent update rule:

$$\Delta \mathbf{W}^* = -\frac{\text{tr}(\mathbf{\Sigma})}{\lambda} \cdot \mathbf{U}\mathbf{V}^\top.$$

This demonstrates that Muon's core operation is a principled descent step where the singular vectors of the gradient determine the direction, and the sum of its singular values scales the step size.

**Adam as Steepest Descent under $\ell_\infty$ Norm.** Adam can be understood as steepest descent on the flattened parameter vector $\mathbf{w}$ when the space is equipped with the vector *infinity norm* ($\ell_\infty$) (defined as the maximum absolute value of its elements, $\|\mathbf{x}\|_\infty = \max_i |x_i|$). For a gradient vector $\mathbf{g}$, the optimization problem is to find the update $\Delta \mathbf{w}$ that solves:

$$\Delta \mathbf{w}^* = \underset{\Delta \mathbf{w}}{\arg\min} \left[ \mathbf{g}^\top \Delta \mathbf{w} + \frac{\lambda}{2} \|\Delta \mathbf{w}\|_\infty^2 \right].$$

The update direction that maximizes alignment with the gradient $\mathbf{g}$ under the infinity norm constraint is the sign of the gradient, $\text{sign}(\mathbf{g})$. The corresponding dual norm of the gradient, $\|\mathbf{g}\|_\infty^*$, which scales the step size, is the $\ell_1$ norm, $\|\mathbf{g}\|_1$ (the sum of the absolute values of its elements, $\|\mathbf{x}\|_1 = \sum_i |x_i|$). Combining these components yields the final steepest descent update rule:

$$\Delta \mathbf{w}^* = -\frac{\|\mathbf{g}\|_1}{\lambda} \cdot \text{sign}(\mathbf{g}).$$

This reveals that Adam's fundamental operation corresponds to a descent step where each parameter moves with the same magnitude, determined only by its gradient's sign.

# E   DETAILED DISCUSSION OF THE THEOREM 4.3

The proof of Theorem 4.3 is provided in Appendix H. We now explain the results for the three optimizers separately. For GD, the quantity $r(\alpha, \beta) \le 1$ measures the imbalance of the data distribution: $r(\alpha, \beta) = 1$ corresponds to perfectly balanced data, while $r(\alpha, \beta) \ll 1$ indicates severe imbalance. The results show that if one set of $(s, r, o)$ triplets is learned with the correct-class probability $[f_W(E_k)]_k$ of at least $1 - \epsilon$, then there exists another triplet whose correct-class probability is $O(\epsilon^{-r(\alpha, \beta)} K^{r(\alpha, \beta)-1})$. Thus, GD is highly sensitive to data imbalance: as the training distribution becomes more imbalanced, the dispersion of correct-class probabilities across items increases, i.e., the maximal probability gap $\Delta(W)$ grows and $\min_{k \in [K]} [f_W(E_k)]_k$ decreases. This mirrors the message in Figure 4(b), 4(c), and Figure 3(d) in Section 3.3.

In contrast, Muon learns in a balanced fashion, unaffected by data imbalance for any embeddings $\widetilde{E}$ and $E$. Our results show that when the best-learned triplet achieves a correct-class probability of at least $1 - \epsilon$, the worst-learned triplet has a comparable correct-class probability at least $1 - \epsilon(1 + O(\log K/K))$. This justifies Observation 3. Furthermore, consistent with Observation 2, Muon's update $G_{\text{Muon}}$ rule allocates equal strength to all update directions; equivalently, the singular values of $G_{\text{Muon}}(W_0)$ are nearly identical.

Our analysis shows that Adam's performance is *unstable* with respect to the embeddings $\widetilde{E}$ and $E$, as reflected by the large error bars in Observations 2 and 3. Adam's element-wise normalization disrupts the inherent matrix structure of the gradient. When embeddings of different triplets have disjoint supports (e.g., $\widetilde{E} = E = I_{K,K}$), Adam can optimize parameters in a balanced manner. However, when embeddings overlap, the sign operator in Adam can introduce imbalance. In particular, the worst-optimized triplet may then have correct-class probability $O(\epsilon^{-0.7} K^{-0.3})$. These exponents $(0.3, 0.7)$ are intrinsic to Adam's update under certain embeddings and are independent of $\alpha$ or $\beta$. Moreover, the Adam update $G_{\text{SignGD}}(W_0)$ exhibits pronounced spectral decay—for example, its smallest singular value can be less than $25\%$ of the largest—unlike the nearly uniform singular values of Muon. This spectral decay explains the poor isotropy reported in Observation 2.

# F   EXPERIMENTAL DETAILS

## F.1   EXPERIMENTAL DETAILS OF TRAINING ON FINEWEB

When training 160M models on FineWeb, we disable weight decaying and Nesterov acceleration for both Adam and Muon. Thus, we only compare their performance along. To set the learning rate, we conduct a grid search on $1 \times 10^{-1}, 5 \times 10^{-2}, 2 \times 10^{-2}, 1 \times 10^{-2}, 5 \times 10^{-3}, 2 \times 10^{-3}, 1 \times 10^{-3}, 5 \times 10^{-4}, 2 \times 10^{-4}$. When conducting the "Independent Blocks " and "Combined Configuration " experiments in Section 3.1, we just fix the learning rate of Muon. We set $\beta_1 = 0.8, \beta_2 = 0.95$ for Adam and set $\beta = 0.95$ for Muon. When training 0.7B models on FineWeb, we conduct a grid

search of learning rate on $2 \times 10^{-3}, 1 \times 10^{-3}, 5 \times 10^{-4}, 2 \times 10^{-4}$. We set $\beta_1 = 0.9, \beta_2 = 0.95$ for Adam and set $\beta = 0.95$ for Muon. We do not adopt group query attention in the structure; thus, the parameter sizes of $W_Q, W_K, W_V$, and $W_O$ are the same. We conduct experiments on 8 A100 with 80 GB memory.

## F.2 ISOTROPICITY METRICS EXPLANATIONS

**Normalized SVD Entropy.** This metric, adapted from Alter et al. (2000), quantifies the uniformity of the singular energy distribution. A higher entropy value indicates a more isotropic matrix where energy is distributed evenly across many directions. It is defined as the Shannon entropy of the distribution $q$, normalized by the maximum possible entropy: $H_{\text{norm}}(\sigma) = -\frac{1}{\log n} \sum_{i=1}^{n} q_i \log q_i$.

**Effective Rank.** The effective rank (Roy & Vetterli, 2007) provides a continuous measure of the number of significant singular dimensions used by the matrix. It is calculated as the exponentiation of the unnormalized Shannon entropy, which corresponds to the perplexity of the energy distribution: $\text{eRank}(\sigma) = \exp\left(-\sum_{i=1}^{n} q_i \log q_i\right)$.

**Top-$k$ Energy Fraction.** This metric measures the concentration of energy within the Top-$k$ principal singular components. Assuming the singular values are sorted in descending order ($\sigma_1 \geq \sigma_2 \geq \cdots \geq \sigma_n$), it is the cumulative sum of the first $k$ energy fractions: $\text{TopE}_k(\sigma) = \frac{\sum_{i=1}^{k} \sigma_i^2}{\sum_{j=1}^{n} \sigma_j^2}$.

**Eigenvalue Quantile Ratio.** To measure the spread of the singular energy distribution while being robust to extreme outliers, we compute the ratio of the 75th percentile ($Q_3$) to the 25th percentile ($Q_1$) of the eigenvalues $\{\sigma_i^2\}_{i=1}^{n}$: $Q_{75/25}(\sigma) = \frac{Q_3(\{\sigma_i^2\})}{Q_1(\{\sigma_i^2\})}$.

## F.3 DATASET DETAILS FOR THE HEAVY-TAIL KNOWLEDGE TASK

Following Allen-Zhu & Li (2024), the foundation of our knowledge-intensive task is a set of question-answering (QA) pairs derived from synthetically generated biographies. Each biography is constructed from a combination of seven key attributes: name, birthdate, birthplace, educational institution, major, employer, and workplace. The attribute values are sampled from predefined lists, creating a diverse set of entities. Specifically, we use approximately 400 first names, 1000 surnames, 300 educational institutions, 100 majors, and 300 employers. Each synthetic individual is assigned a unique combination of these attributes, forming a distinct biographical profile. For example, a generated biography might look like this:

> ***Ashton Hilda Older*** *has a birthday that falls on* ***February 01, 2063***. ***Miami, FL*** *is the birthplace of he. He is an alumnus of* ***Saddleback College***. *He has a **General Literature** education. He works closely with **BlockFi***. *For professional growth, he chose to relocate to **Jersey City***.

This text is generated by combining the **structured attributes** (name, date, location, etc.) with a set of sentence templates.

A predefined set of QA templates is then used to generate the final training data. These templates contain placeholders corresponding to the biographical attributes. By formatting these templates with the information from each synthetic biography, we generate a collection of concrete QA pairs for each entity. For example, for the entity "Ashton Hilda Older", we can generate the following six QA pairs:

1. What is the birth date of Ashton Hilda Older?
   **Answer: February 01, 2063.**

2. What is the birth city of Ashton Hilda Older?
   **Answer: Miami, FL.**

3. Which university did Ashton Hilda Older study?
   **Answer: Saddleback College.**

4. What major did Ashton Hilda Older study?
   **Answer: General Literature.**

5. Which company did Ashton Hilda Older work for?
   **Answer: BlockFi.**

6. Where did Ashton Hilda Older work?
   **Answer: Jersey City.**

To evaluate the optimizers on a knowledge-intensive task with data imbalance, we constructed a synthetic dataset where the number of question-answering (QA) samples per class follows a power-law distribution. This is designed to simulate real-world scenarios where a few entities (the "head") are highly represented, while most entities (the "tail") are rare.

The generation process is controlled by an integer parameter, $m$. The classes are organized into $m + 1$ groups, indexed from $g = 0$ to $m$.

- Group $g$ contains $N_g$ classes, where $N_0 = 1$ and $N_g = 2^{g-1}$ for $g > 0$.

- Each class within group $g$ is allocated a specific number of "selections," $S_g = 2^{m-g}$.

- For each selection, we generate $n_{qa}$ unique QA pairs by formatting templates with biographical information corresponding to that class.

Thus, the total number of QA samples for any given class in group $g$ is $S_g \times n_{qa}$. This structure ensures that the single class in group 0 has the most samples, while the numerous classes in group $m$ have the fewest.

In our experiment, we set the parameters to $m = 15$ and $n_{qa} = 6$. This results in a dataset with a total of $2^{15} = 32,768$ classes. The number of samples per class ranges from $196,608$ for the head class (group 0) down to just 6 for each of the $16,384$ tail classes (group 15). The final distribution is visualized in Figure 3(a) in the main text.

To evaluate the model's performance on this pure memory task, we measure the First Token Accuracy (FTA) on the answers. This metric assesses the model's ability to correctly recall information by checking if the first generated token of the answer matches the ground truth. Furthermore, to understand how optimizers handle data imbalance, we analyze the FTA across different data frequency groups, from high-frequency (head) to low-frequency (tail) data.

### F.4 EXPERIMENTAL DETAILS ABOUT ANGLES BETWEEN ASSOCIATIVE MEMORIES EMBEDDINGS

Following Fang et al. (2024), we analyze the associative memories in the FFN modules. To obtain $E_i$, we use the activations within the feed-forward modules, and for $\widetilde{E}_i$, we take the corresponding module outputs. We evaluate knowledge items from two widely used datasets: Counterfact (Meng et al., 2022a) and ZsRE (Levy et al., 2017). Results on Counterfact are shown in Figure 4(a), while results on ZsRE are provided in Figure 18 in Appendix G.11.

### F.5 EXPERIMENTAL DETAILS OF ONE-LAYER MODELS

We set the hyperparameters as $K = d = 999$, $\alpha = 0.8$, $\beta = 0.2$. For the support-decoupled setting, we set $E$ and $\widetilde{E}$ as identity matrices. For the support-coupled setting, we set $E$ and $\widetilde{E}$ according to the construction presented in the proof of Theorem 4.3 in Appendix H.

# G  ADDITIONAL EXPERIMENTAL RESULTS

## G.1  MAXLOGIT PER LAYER ON THE 160M NANOGPT MODEL VIA MUON OPTIMIZER

In this subsection, we present the MaxLogit values for each layer of the 160M NanoGPT model trained using the Muon Optimizer. Following Gemma 3 (Kamath et al., 2025), we introduce RM-SNorm to the attention mechanism. The attention mechanism in our model is defined as follows:

$$O = \text{softmax}(\widetilde{Q}\widetilde{K}^T)V, \quad \widetilde{Q} = \text{RMSNorm}(Q), \quad \widetilde{K} = \text{RMSNorm}(K)$$

where RMSNorm is defined as $\text{RMSNorm}(x) = \frac{x}{\sqrt{\frac{1}{d}\sum_{i=1}^{d} x_i^2}}$, with $d$ being the dimension of $x$. MaxLogit is defined as:

$$S_{\max} = \max_{i,j} \widetilde{q}_i \cdot \widetilde{k}_j$$

representing the maximum value in the attention scores before softmax normalization.

The MaxLogit values for each layer are summarized in Table 1.

Table 1: MaxLogit values per layer on the 160M NanoGPT model via Muon Optimizer.

| Layer | 1 | 2 | 3 | 4 | 5 | 6 | 7 | 8 | 9 | 10 | 11 | 12 |
|---|---|---|---|---|---|---|---|---|---|---|---|---|
| MaxLogit | 8.396 | 6.880 | 6.009 | 7.676 | 6.349 | 5.890 | 7.688 | 6.314 | 6.205 | 5.613 | 6.033 | 6.371 |

Recent reports Team et al. (2025) have shown a potential "MaxLogit explosion" phenomenon, where $S_{\max}$ grows steadily (often near-linearly) during training, leading to overly peaked attention, gradient spikes, and degraded optimizer comparisons. We included this measurement to rule out the possibility that Muon's comparatively smaller impact on the QK blocks (relative to VO/FFN) is simply due to suppressing such an instability. In our 160M setting, with RMSNorm applied to both $Q$ and $K$ (following Gemma 3), the per-layer MaxLogit values remain moderate and show no runaway growth. Thus, for this model size and normalization scheme, differences in Muon's effectiveness across components cannot be attributed to avoiding a MaxLogit explosion in attention.

## G.2  CONTROLLING FOR PARAMETER COUNT IN COMPONENT-WISE ABLATIONS

A potential confounding factor in our ablation studies (Section 3.1) is that different model components contain different numbers of parameters. One might argue that applying Muon to a larger component naturally yields greater gains simply because more parameters are being optimized differently. To disentangle the effect of component type from the effect of parameter count, we measure the performance gain *per parameter*.

We measure the validation-loss improvement at 10,000 steps when applying Muon to a single component (QK, VO, $W_{\text{in}}$, or $W_{\text{out}}$) relative to a full-Adam baseline. This gain is then normalized by the number of parameters in that specific component. For the 160M model, the parameter counts satisfy $|W_V| = |W_O| = |W_Q| = |W_K|$ and $|W_{\text{in}}| = |W_{\text{out}}| = 4 \times |W_Q|$.

Figure 5(a) reports the validation loss at 10,000 steps, normalized by the number of parameters in each component. This result shows that the normalized gain for VO Attn is approximately 5 times greater than that for QK Attn, even though both components have the same number of parameters. The gains for $W_{\text{in}}$ and $W_{\text{out}}$ are also substantially higher (over 3x) than for QK Attn. Although $W_{\text{in}}$ and $W_{\text{out}}$ have twice as many parameters as QK, their normalized gains are far more than half the gain of QK.

To provide even more direct evidence, we designed the second experiment where the number of parameters optimized by Muon is held exactly equal across different components. We achieve this by comparing three configurations: Muon applies to QK matrices in all layers; but Muon applies to $W_{\text{in}}$ and $W_{\text{out}}$ matrices in only the odd-numbered layers. In this setup, the total number of parameters of QK, $W_{\text{in}}$ and $W_{\text{out}}$ optimized by Muon are identical. The results in Figure 5(b) show that even when optimizing an identical number of parameters, the gain from applying Muon to $W_{\text{in}}$ or $W_{\text{out}}$ is much more than the gain from applying it to the QK blocks.

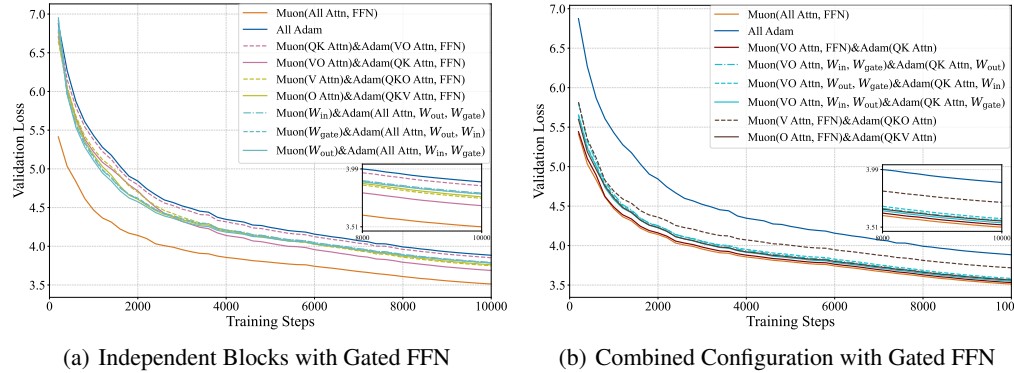

(a) Independent Blocks with Gated FFN   (b) Combined Configuration with Gated FFN

Figure 6: Validation loss comparison on the 160M NanoGPT model with gated FFN under different Muon/Adam assignments. Panels (a) and (b) show the validation loss over training steps for the Independent Blocks and Combined Configurations settings, respectively.

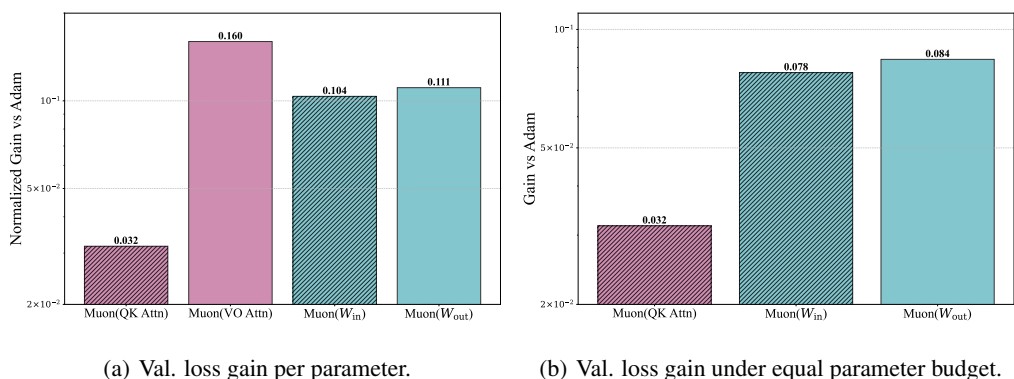

(a) Val. loss gain per parameter.   (b) Val. loss gain under equal parameter budget.

Figure 5: Component-wise validation-loss gain of Muon over Adam at 10,000 steps.

These results demonstrate that Muon's effectiveness is not simply about the quantity of parameters, but is highly specific to the *function* of the parameters. The associative memory components (VO, $W_{in}$, $W_{out}$) derive a much larger benefit per parameter, reinforcing our central claim that Muon excels at optimizing these specific parts of the Transformer architecture.

### G.3 ADDITIONAL RESULTS FOR GATED FFN ON FINEWEB

To verify that our findings in Section 3.1 are not specific to the non-gated FFN architecture, we repeat the same "Independent Blocks" and "Combined Configurations" experiments on the 160M NanoGPT model with a gated FFN. The results are presented in Figure 6.

The conclusions are almost identical to those from the non-gated setting (Figure 1). Specifically, in both the independent and combined settings, applying Muon to VO+FFN yields the most significant validation loss reduction, closely tracking the performance of full Muon. In contrast, applying Muon only to the QK blocks provides minimal benefit over the Adam baseline. This confirms that our finding—that the associative memory components (VO and FFN) are the primary beneficiaries of Muon—is robust to variations in the Transformer architecture, holding for both gated and non-gated FFNs.

Furthermore, we analyze the spectral dynamics of the weight matrices for the gated FFN model, with results for the VO and $W_{out}$ matrices shown in Figure 7. The trends are consistent with Observation 2 from the main text: for both matrices, Muon leads to significantly higher SVD entropy and effective rank (eRank) compared to Adam. This indicates that Muon encourages the learning

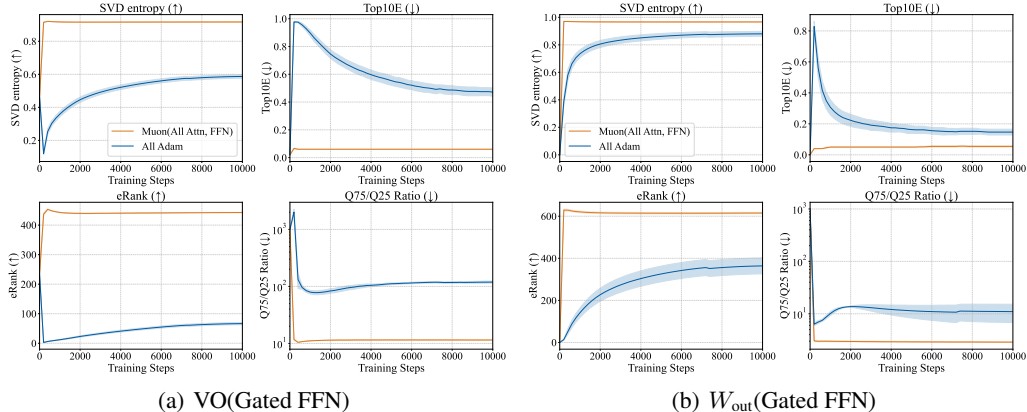

(a) VO(Gated FFN)        (b) $W_{\text{out}}$(Gated FFN)

Figure 7: Spectral Dynamics of Transformer Weight Matrices During Training. Each panel reports four metrics characterizing singular value distributions: SVD entropy, Top10E, eRank, and Q75/Q25 ratio. The four subplots correspond to different weight matrix groups: (a) VO and (b) $W_{\text{out}}$.

of more distributed, higher-dimensional representations in the associative memory components, a finding that holds true for the gated FFN architecture as well.

## G.4 SCALING TO THE 0.7B NANOGPT MODEL

To evaluate the scalability of our findings, we extend our experiments from the 160M model to a larger 0.7B parameter model. This section presents the results of this scaled-up analysis, examining whether the advantages of Muon observed in the smaller model persist at a larger scale.

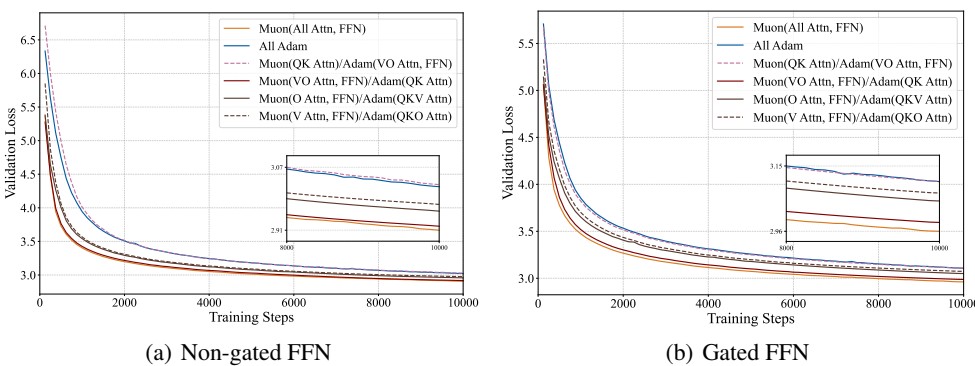

(a) Non-gated FFN        (b) Gated FFN

Figure 8: Validation loss comparison on the 0.7B NanoGPT model. (a) Combined configuration with non-gated feed-forward networks.(b) Combined configuration with gated feed-forward networks.

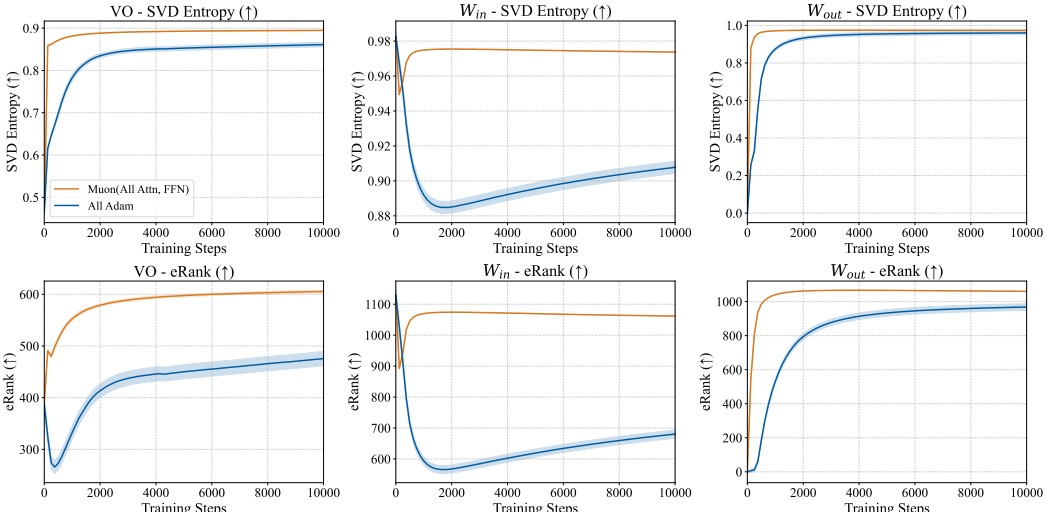

Figure 9: Spectral Dynamics of Weight Matrices During Training on the 0.7B NanoGPT model.

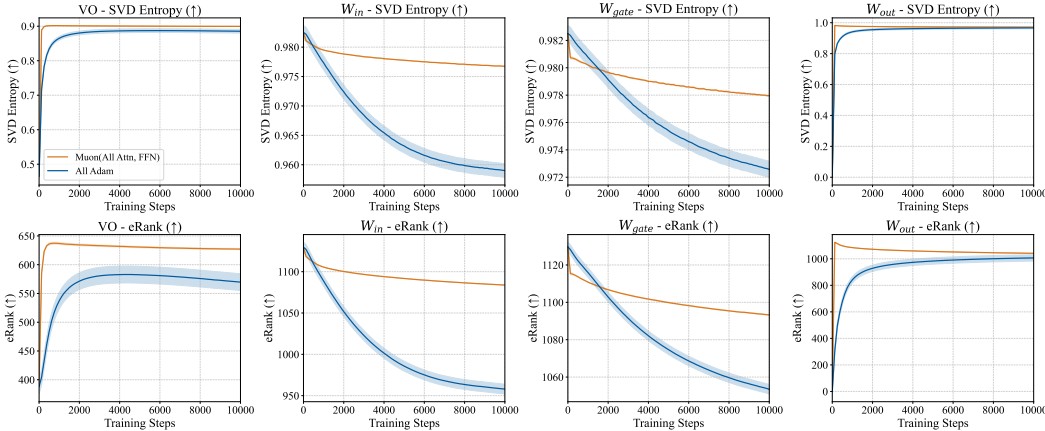

Figure 10: Spectral Dynamics of Weight Matrices During Training on the 0.7B NanoGPT model with the Gated FFN.

Figure 8 shows the validation loss curves for various optimizer configurations. Consistent with our findings on the 160M model, applying Muon to all components achieves the lowest validation loss, outperforming Adam baseline. The hybrid experiments further reinforce our earlier conclusions: applying Muon to only the VO and FFN components yields performance nearly identical to that of the full Muon optimizer, whereas applying it only to the QK components offers little advantage over Adam.

The spectral dynamics, shown in Figures 9 and 10, also align with Observation 2. For the VO, $W_{\text{in}}$, $W_{\text{gate}}$ (in model with Gated FFN) and $W_{\text{out}}$ matrices, Muon leads to higher SVD entropy and eRank compared to Adam, indicating that it encourages the learning of more distributed, higher-dimensional representations. Overall, these results demonstrate that the benefits of Muon and the underlying mechanisms scale to larger models.

## G.5 ADDITIONAL RESULTS ABOUT SPECTRAL DYNAMICS OF TRANSFORMER WEIGHT MATRICES DURING TRAINING

To complement the main-text analysis (Fig. 2), we also evaluate spectral dynamics during training for the 160M NanoGPT model with both non-gated and gated feed-forward networks (Fig. 11).

The analysis includes $W_{\text{in}}$ for both configurations, as well as the gate matrix $W_{\text{gate}}$ for the gated version. The conclusions are consistent across all three matrices and mirror the non-gated setting: with Muon, SVD entropy and eRank increase, while Top-$k$ energy and the $Q_{75/25}$ ratio decrease, consistent with Observation 2 in the main text.

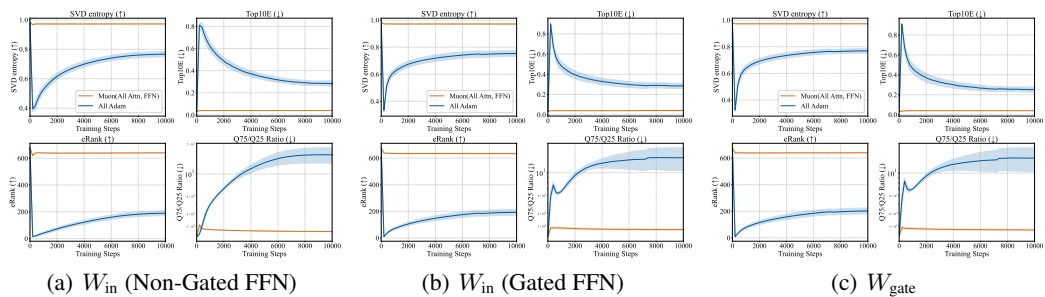

(a) $W_{\text{in}}$ (Non-Gated FFN)  (b) $W_{\text{in}}$ (Gated FFN)  (c) $W_{\text{gate}}$

Figure 11: Spectral Dynamics of FFN Weight Matrices During Training on the 160M NanoGPT model. Each panel reports four metrics characterizing singular value distributions: SVD entropy, Top10E, eRank, and Q75/Q25 ratio. The subplots correspond to different weight matrices: (a) $W_{\text{in}}$ (non-gated), (b) $W_{\text{in}}$ (gated), and (c) $W_{\text{gate}}$ (gated).

### G.6 DETAILED EXPERIMENT RESULTS ABOUT HEAVY-TAIL IMBALANCE KNOWLEDGE TASK

To complement the qualitative trends shown in Section 3.3 (Fig. 3), we report the exact First Token Accuracy (FTA) for selected tail groups at three training checkpoints (2k, 5k, 10k steps). We focus on groups $g = 11, 13, 15$, which represent increasingly rare (mid–tail, tail, extreme tail) frequency bands in the power-law distribution (recall that larger $g$ implies fewer samples per class). The tables contrast full Muon, Adam, SGD+Momentum, and two hybrid configurations (Muon applied only to VO&FFN or only to QK). The numbers highlight: (i) Muon's rapid convergence on rare groups (already strong by 2k, near-saturated by 5k), (ii) Adam's persistent head–tail gap, and (iii) the dominant contribution of applying Muon to VO&FFN for tail generalization (the VO&FFN hybrid closely tracks full Muon, whereas the QK-only hybrid lags). These quantitative results substantiate Observation 3 that Muon delivers more balanced learning.

Table 2: Heavy-tail knowledge task: Group performance by optimizer (2,000 steps)

| Group | Optimizer | | | | |
|---|---|---|---|---|---|
| | Muon | Adam | SGD+Mom. | Muon(VO, FFN) | Muon(QK) |
| 11 | **0.854 ± 0.029** | 0.312 ± 0.043 | 0.156 ± 0.037 | 0.814 ± 0.022 | 0.472 ± 0.041 |
| 13 | **0.386 ± 0.029** | 0.146 ± 0.015 | 0.120 ± 0.012 | 0.256 ± 0.030 | 0.154 ± 0.032 |
| 15 | **0.140 ± 0.027** | 0.090 ± 0.031 | 0.082 ± 0.013 | 0.114 ± 0.023 | 0.086 ± 0.037 |

Table 3: Heavy-tail knowledge task: Group performance by optimizer (5,000 steps)

| Group | Optimizer | | | | |
|---|---|---|---|---|---|
| | Muon | Adam | SGD+Mom. | Muon(VO, FFN) | Muon(QK) |
| 11 | **0.996 ± 0.006** | 0.936 ± 0.039 | 0.314 ± 0.021 | 0.992 ± 0.005 | 0.970 ± 0.007 |
| 13 | **0.964 ± 0.023** | 0.298 ± 0.074 | 0.148 ± 0.013 | 0.934 ± 0.015 | 0.354 ± 0.032 |
| 15 | **0.320 ± 0.028** | 0.110 ± 0.027 | 0.084 ± 0.011 | 0.254 ± 0.026 | 0.118 ± 0.019 |

Table 4: Heavy-tail knowledge task: Group performance by optimizer (10,000 steps)

| Group | Optimizer | | | | |
|---|---|---|---|---|---|
| | Muon | Adam | SGD+Mom. | Muon(VO, FFN) | Muon(QK) |
| 11 | 1.000 ± 0.000 | 1.000 ± 0.000 | 0.422 ± 0.023 | 1.000 ± 0.000 | 1.000 ± 0.000 |
| 13 | **1.000 ± 0.000** | 0.890 ± 0.042 | 0.294 ± 0.013 | 0.998 ± 0.002 | 0.940 ± 0.034 |
| 15 | **0.976 ± 0.006** | 0.264 ± 0.048 | 0.126 ± 0.021 | 0.954 ± 0.021 | 0.286 ± 0.039 |

## G.7 Additional Experiment Results about Heavy-Tail Imbalance Knowledge Task with Gated Feed-Forward Networks

This subsection complements the main heavy-tail results in Section 3.3 by studying the gated feed-forward networks (Gated FFN) variant. We follow the same presentation order as in the main text: first an overview figure (sample distribution and learning curves under different optimizers), then tables reporting the exact First-Token Accuracy (FTA) for tail groups $g \in \{11, 13, 15\}$ at three training checkpoints (2k, 5k, 10k steps). The findings mirror the non-gated setting: (i) full Muon consistently outperforms Adam and SGD+Momentum on rare classes and reaches high accuracy earlier; (ii) the VO&FFN-hybrid (Muon applied to VO and FFN while Adam is used for QK) closely tracks full Muon, indicating that VO&FFN are the primary levers for tail generalization; (iii) the QK-only hybrid offers limited gains. Overall, the gated FFN does not change the qualitative conclusions about where Muon helps most. See Fig. 12 and Tables 5–7 for details.

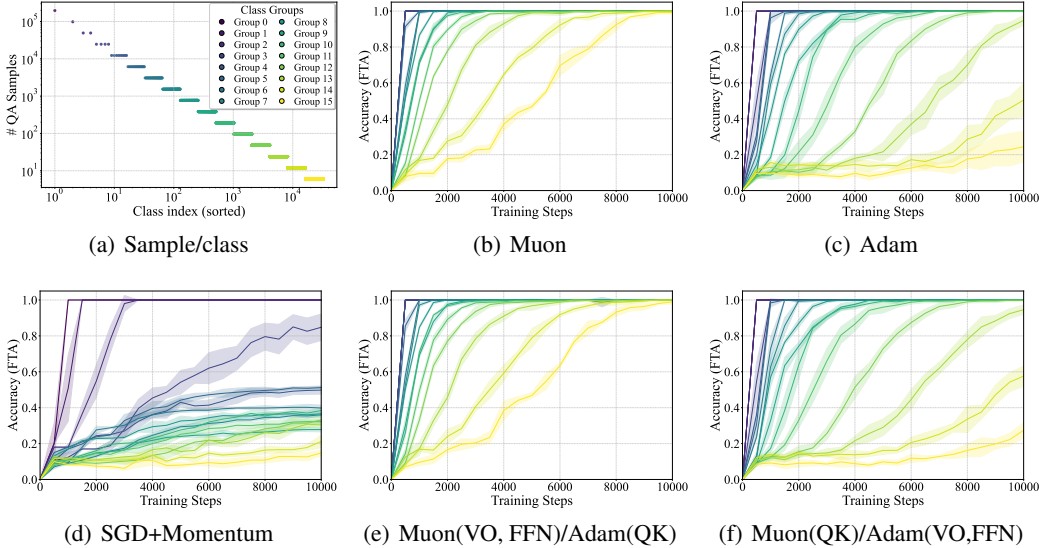

Figure 12: Performance comparison of different optimizers on a heavy-tailed knowledge task with gated feed-forward networks. (a) The distribution of samples per class follows a power law. (b-d) Performance of Muon, Adam, and SGD+Momentum optimizers. (e) Muon (VO, FFN)/Adam (QK). (f) Muon (QK)/Adam (VO, FFN).

Table 5: Heavy-tail knowledge task with the Gated FFN: Group performance by optimizer (2,000 steps)

| Group | Optimizer | | | | |
|---|---|---|---|---|---|
| | Muon | Adam | SGD+Mom. | Muon(VO, FFN) | Muon(QK) |
| 11 | **0.896 ± 0.009** | 0.214 ± 0.063 | 0.146 ± 0.018 | 0.892 ± 0.021 | 0.330 ± 0.042 |
| 13 | **0.478 ± 0.034** | 0.116 ± 0.030 | 0.110 ± 0.007 | 0.458 ± 0.037 | 0.140 ± 0.019 |
| 15 | **0.178 ± 0.018** | 0.086 ± 0.013 | 0.074 ± 0.009 | 0.166 ± 0.017 | 0.090 ± 0.020 |

Table 6: Heavy-tail knowledge task with the Gated FFN: Group performance by optimizer (5,000 steps)

| Group | Optimizer | | | | |
|---|---|---|---|---|---|
| | Muon | Adam | SGD+Mom. | Muon(VO, FFN) | Muon(QK) |
| 11 | **0.998 ± 0.002** | 0.928 ± 0.024 | 0.252 ± 0.016 | 0.990 ± 0.010 | 0.960 ± 0.032 |
| 13 | **0.990 ± 0.010** | 0.216 ± 0.052 | 0.156 ± 0.024 | 0.968 ± 0.028 | 0.290 ± 0.046 |
| 15 | **0.510 ± 0.039** | 0.092 ± 0.015 | 0.080 ± 0.016 | 0.468 ± 0.016 | 0.098 ± 0.013 |

Table 7: Heavy-tail knowledge task with the Gated FFN: Group performance by optimizer (10,000 steps)

| Group | Optimizer | | | | |
|---|---|---|---|---|---|
| | Muon | Adam | SGD+Mom. | Muon(VO, FFN) | Muon(QK) |
| 11 | 1.000 ± 0.000 | 0.998 ± 0.002 | 0.322 ± 0.011 | 1.000 ± 0.000 | 1.000 ± 0.000 |
| 13 | **1.000 ± 0.000** | 0.948 ± 0.027 | 0.304 ± 0.017 | **1.000 ± 0.000** | 0.946 ± 0.026 |
| 15 | **0.994 ± 0.006** | 0.244 ± 0.085 | 0.148 ± 0.015 | 0.990 ± 0.010 | 0.274 ± 0.042 |

### G.8 IMPACT OF DATA IMBALANCE LEVEL

To further investigate how the degree of data imbalance affects the performance gap between Muon and Adam, we conduct an ablation study on the heavy-tail knowledge task with varying levels of class imbalance. We compare three settings:

- **High Imbalance (base = 2.0):** This is the default setting used in our main experiments (Section 3), where the number of samples per class follows the power-law construction in Section F.3 with base 2.0.
- **Medium Imbalance (base = 1.2):** A less skewed version of the same construction, where the base is reduced to 1.2 so that the head–tail ratio is smaller.
- **Uniform:** A balanced setting where each group contains the same number of classes and each class is assigned the same number of QA samples.

The results are presented in Figure 13. From left to right, the panels correspond to the high-imbalance, medium-imbalance, and uniform settings, each plotting the average First Token Accuracy (FTA) over all groups for Adam and Muon. As the data distribution becomes more uniform, the performance gap between Muon and Adam steadily shrinks, and in the uniform case the two optimizers behave very similarly, indicating that Muon's advantage is most pronounced in highly imbalanced, heavy-tailed regimes.

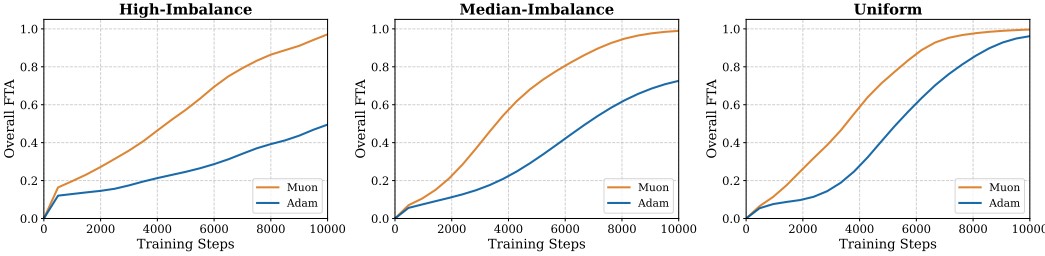

Figure 13: Comparison of Muon and Adam under different levels of class imbalance on the heavy-tail knowledge task. From left to right, the panels correspond to the High Imbalance (base = 2.0), Medium Imbalance (base = 1.2), and Uniform settings.

The datasets mixed with different levels of heavy-tailedness exhibit two properties: (1) Figure 14(a) shows that the single-token distributions are not exactly the same, i.e., facts and tokens cannot be

perfectly decoupled; and (2) the token distribution in the uniform mixture still follows Zipf's law (Figure 14(b)). Thus, we conclude that, under different levels of heavy-tailedness in the pretraining data, the benefit of Muon over Adam varies even when the token distribution remains close to Zipf's law: the more uniform the mixture, the smaller the gain of Muon over Adam.

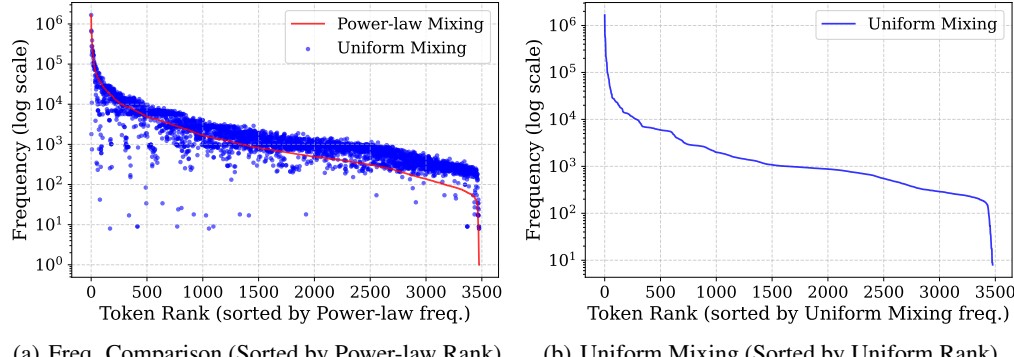

(a) Freq. Comparison (Sorted by Power-law Rank)  (b) Uniform Mixing (Sorted by Uniform Rank)

Figure 14: Token-frequency profiles for the synthetic heavy-tail QA task. (a) Compares the token frequencies of the original Power-law mixing (red curve) and a fact-balanced Uniform mixing (blue dots), using the token rank from the power-law mixing. (b) Shows the token frequency profile for the Uniform mixing data, re-sorted by its own token frequencies.

### G.9 ADDITIONAL RESULTS ON WIKITEXT103

To verify that our observations on FineWeb and the synthetic heavy-tail knowledge task transfer to a more standard language modeling benchmark, we additionally train 160M NanoGPT models on the Wikitext103 dataset. We keep the model architecture and most hyperparameters identical to the FineWeb setup and only retune the learning rate for each optimizer with a small grid search.

Figure 15 provides an overview of this setting. Panel (a) shows the empirical token frequency distribution of Wikitext103, which exhibits a clear heavy-tail pattern: a small number of tokens appear very frequently, while many tokens are rare. In the plot, the vocabulary is partitioned into ten frequency-based groups, each containing approximately 10% of the tokens (from most frequent to rarest), to make head and tail behavior more comparable. Panels (b) and (c) report the training loss curves for Adam and Muon, respectively. Consistent with our main results, Muon converges faster and reaches a lower training loss than Adam.

Figure 16 further highlights the difference between the two optimizers by plotting their training losses on the same axes. Looking from the head group to the tail group, the performance gap between Muon and Adam steadily widens: while the two optimizers behave similarly on high-frequency (head) tokens, Muon remains much stronger on mid- and low-frequency (tail) tokens. In addition, the error bars for Adam grow substantially toward the tail, indicating unstable generalization on rare tokens, whereas Muon stays consistently stable across all groups.

Figure 16 also reports the two hybrid configurations. The Muon(VO, FFN) variant, which applies Muon only to the value/output and feed-forward blocks while keeping Adam on QK, almost overlaps with the full Muon curve, showing that most of the improvement comes from these components. In contrast, the Muon(QK)-only variant is very close to the Adam baseline, suggesting that using Muon solely on the QK blocks brings limited benefit.

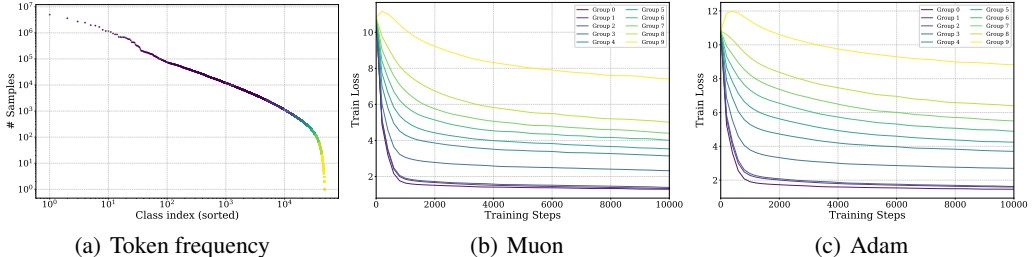

| (a) Token frequency | (b) Muon | (c) Adam |

Figure 15: Performance comparison of different optimizers on Wikitext103. (a) Token frequency distribution in the Wikitext103 training corpus, showing a pronounced heavy-tail structure. (b) Training loss curve for Muon. (c) Training loss curve for Adam.

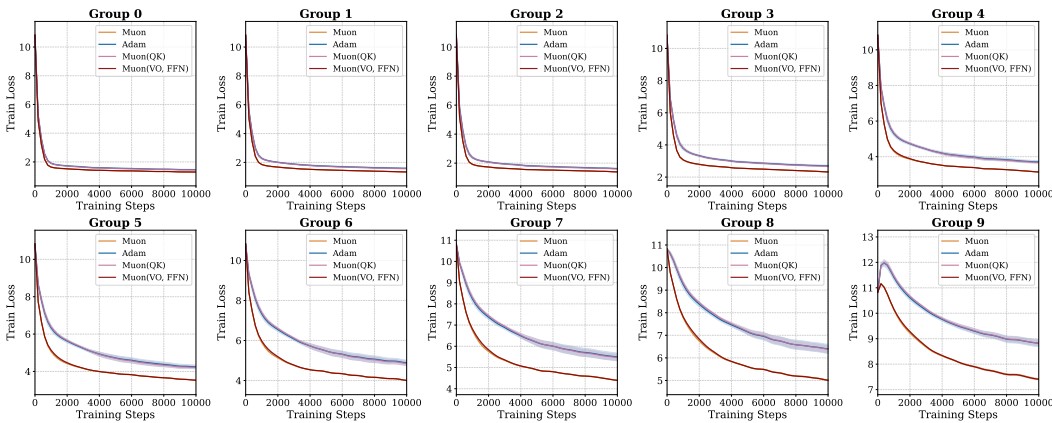

Figure 16: Training loss comparison on Wikitext103 across head and tail token groups under different optimizer configurations. The curves correspond to Adam, Muon, and two hybrid variants that apply Muon only to VO&FFN or only to QK. In these figures, the results of Muon(VO, FFN) coincide with those of Muon, while the results of Muon(QK) coincide with those of Adam.

## G.10 ADDITIONAL RESULTS ON LINEAR REGRESSION

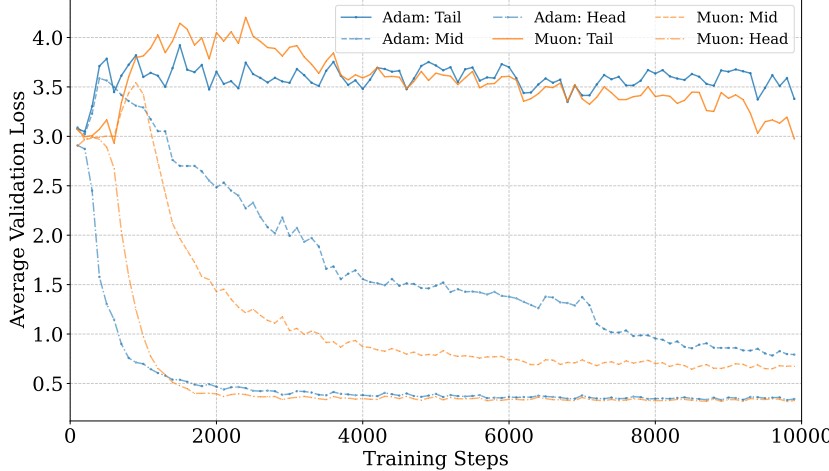

Figure 17: Validation loss on linear regression across head and tail groups under different Adam and Muon.

To further demonstrate that Muon is ineffective at optimizing the QK parameters in the attention module, we consider an in-context linear regression task (Garg et al., 2022), which heavily relies on the QK parameters. In this task, the model is prompted with a number of demonstrations $(x_i, y_i)_{i=1}^{K}$ with $y_i = x_i^\top w$ and a query $x_q$, where $x_i \in \mathbb{R}^d$ for $i \in [K]$ and $x_q, w \in \mathbb{R}^d$. The model is expected to output $x_q^\top w$. Intuitively, the QK parameters capture the correlations between the demonstrations and the query and use them to estimate $x_q^\top w$. Following (Garg et al., 2022), we train the model with $\ell_2$ loss. To test the efficacy of the optimizers under a heavy-tailed task distribution, we partition $w$ into groups supported on mutually orthogonal subspaces, which appear in the training data with different frequencies. We perform a grid search over learning rates for Adam and Muon and report the results in Figure 17.

Figure 17 shows that Adam and Muon achieve similar performance across different groups. In particular, both optimizers effectively learn the head class but barely improve on the tail class. This behavior is in sharp contrast to the results in Section 3.3, where Muon substantially outperforms Adam on tail classes. Hence, the linear regression experiment further supports our claim that the main benefit of Muon does not come from optimizing the QK parameters.

### G.11 ADDITIONAL RESULTS ABOUT ANGLES BETWEEN ASSOCIATIVE MEMORIES EMBEDDINGS

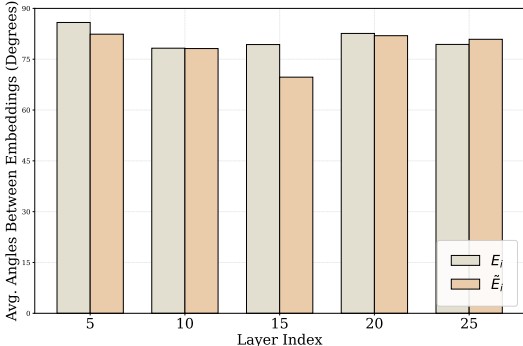

Figure 18: Average angles between $e_s$ or $e_o$ for items in ZsRE at layers 5, 10, 15, 20, 25 of Llama3-8b-instruct.

## H PROOF OF THEOREM 4.3

We separately derive the results for GD, Muon, and Adam in the following proof. For all of them, we define

$$\eta_{\text{opt}}^\epsilon = \inf \left\{ \eta \geq 0 \,\middle|\, 1 - \max_{k \in [K]} \left[ f_W(E_k) \right]_k \leq \epsilon, \text{ where } W = W_0 - \eta \cdot G_{\text{opt}}(W_0) \right\}. \quad \text{(H.1)}$$

The quantity $\eta_{\text{opt}}^\epsilon$ represents the minimal step size for at least one triplet to be learned with error probability less than $\epsilon$. From the definition, we have that

$$\varrho_{\text{opt}}^\epsilon \leq \min_{k \in [K]} \left[ f_{-\eta_{\text{opt}}^\epsilon G_{\text{opt}}}(E_k) \right]_k.$$

**Step 1: Calculations of GD.**

We define the score of $k'$-th object for the $k$-th subject-relation pair with the parameter $W$ as

$$s(k', k, W) = \frac{\exp(\widetilde{E}_{k'}^\top W E_k)}{\sum_{k''=1}^{K} \exp(\widetilde{E}_{k''}^\top W E_k)}.$$

At $W_0 = 0_{d_o, d_s}$, we have that

$$s(k', k, W_0) = \frac{1}{K} \text{ for all } k, k' \in [K].$$

Proposition J.1 shows that the gradient is

$$-\nabla_W \mathcal{L}(W_0) = \frac{\alpha}{L} \widetilde{E}_{1:L} E_{1:L}^\top + \frac{1-\alpha}{K-L} \widetilde{E}_{L+1:K} E_{L+1:K}^\top - \frac{\alpha}{LK} \widetilde{E} J_{K,L} E_{1:L}^\top$$
$$- \frac{1-\alpha}{(K-L)K} \widetilde{E} J_{K,K-L} E_{L+1:K}^\top. \tag{H.2}$$

From the gradient, it is easy to see that the first $L$ triplets $(s, r, o)$ share the same learning behavior, and the last $K - L$ triplets also share the same behavior. Thus, we calculate the results for $k = 1$ and $k = L + 1$. The calculation for $k = 1$ depends on evaluating its score function, which takes the form $\eta \cdot \widetilde{E}_{k''}^\top [-\nabla_W \mathcal{L}(W_0)] E_1$, for $k'' \in \{1, \ldots, K\}$. Based on the gradient in (H.2) and the orthonormality of the embeddings, it evaluates to $\alpha/L$ for the case $k'' = 1$, and to $0$ for all $k'' \neq 1$.

This leads to a numerator in the softmax score of $\exp(\eta \cdot \alpha/L)$, while the denominator sum consists of one term $\exp(\eta \cdot \alpha/L)$ and $K - 1$ terms of $\exp(0) = 1$. A similar calculation for $k = L + 1$ shows that the argument of the exponent for the correct object, $\eta \cdot \widetilde{E}_{L+1}^\top [-\nabla_W \mathcal{L}(W_0)] E_{L+1}$, evaluates to $\eta \cdot (1 - \alpha)/(K - L)$. By defining $\gamma_1 = \alpha/(\beta K)$ and $\gamma_2 = (1 - \alpha)/((1 - \beta)K)$ based on the problem setup ($L = \beta K$), we have that

$$\left[ f_{-\eta \nabla_W \mathcal{L}}(E_1) \right]_1 = \frac{\exp(\eta \gamma_1)}{\exp(\eta \gamma_1) + K - 1}, \quad \left[ f_{-\eta \nabla_W \mathcal{L}}(E_{L+1}) \right]_{L+1} = \frac{\exp(\eta \gamma_2)}{\exp(\eta \gamma_2) + K - 1},$$

where $\gamma_1$ and $\gamma_2$ are defined as

$$\gamma_1 = \frac{\alpha}{\beta K}, \quad \gamma_2 = \frac{1 - \alpha}{(1 - \beta)K}.$$

Then we derive that

$$\eta_{\text{GD}}^\epsilon = \frac{1}{\max\{\gamma_1, \gamma_2\}} \log \left[ (\epsilon^{-1} - 1)(K - 1) \right]. \tag{H.3}$$

To calculate the desired quantity, we define the quantity $r(\alpha, \beta)$ to evaluate the balance of data as

$$r(\alpha, \beta) = \min\{\gamma_1/\gamma_2, \gamma_2/\gamma_1\} = \min \left\{ \frac{\alpha(1 - \beta)}{\beta(1 - \alpha)}, \frac{\beta(1 - \alpha)}{\alpha(1 - \beta)} \right\}.$$

Some basic calculations show that

$$1 - \min_{k \in [K]} \left[ f_{-\eta_{\text{GD}}^\epsilon G_{\text{GD}}}(E_k) \right]_k = \frac{\epsilon}{\epsilon + (1 - \epsilon)^{r(\alpha,\beta)} \epsilon^{1-r(\alpha,\beta)} (K - 1)^{r(\alpha,\beta)-1}}. \tag{H.4}$$

When $r < 1$, with the fact that $\frac{1}{x+1} = 1 - x + O(x^2)$ as $x \to 0$, we have that

$$\min_{k \in [K]} \left[ f_{-\eta_{\text{GD}}^\epsilon G_{\text{GD}}}(E_k) \right]_k = O(\epsilon^{-r(\alpha,\beta)} K^{r(\alpha,\beta)-1}).$$

Thus, the proof for the convergence of GD has been established.

**Step 2: Calculations of Muon.**

For Muon, we first calculate the SVD of the gradient. In fact, we can write the gradient in Eqn. (H.2) as

$$-\nabla_W \mathcal{L}(W_0) = \widetilde{E} \left\{ \text{diag}\left( \frac{\alpha}{L} \mathbb{I}_L, \frac{1-\alpha}{K-L} \mathbb{I}_{K-L} \right) - \frac{1}{K} \mathbb{I}_K \cdot \left[ \frac{\alpha}{L} \mathbb{I}_L^\top, \frac{1-\alpha}{K-L} \mathbb{I}_{K-L}^\top \right]^\top \right\} E^\top$$
$$= \widetilde{E} X E^\top.$$

The SVD calculation of $X = U\Sigma V^\top$ can be directly derived from Proposition J.3. Thus, the SVD of the gradient is $-\nabla_W \mathcal{L}(W_0) = (\widetilde{E} \cdot U)\Sigma(E \cdot V)^\top$. The update quantity $G_{\text{Muon}}(W_0) =$

$U_0 \mathrm{norm}(\Sigma_0) V_0^\top$ of Muon is

$$
\begin{aligned}
& - G_{\mathrm{Muon}}(W_0) \\
&= \widetilde{E}_{1:L} R_{L,L-1} R_{L,L-1}^\top E_{1:L}^\top + \widetilde{E}_{L+1:K} R_{K-L,K-L-1} R_{K-L,K-L-1}^\top E_{L+1:K}^\top \\
&\quad + \frac{1}{\sqrt{K\left[\alpha^2(K-L)^3 + (1-\alpha)^2 L^3\right]}} \left((K-L)\widetilde{E}_{1:L}\mathbb{I}_L - L\widetilde{E}_{L+1:K}\mathbb{I}_{K-L}\right) \\
&\qquad\qquad\qquad\qquad\qquad\qquad \cdot \left(\frac{(K-L)\alpha}{L}\mathbb{I}_L^\top E_{1:L}^\top - \frac{L(1-\alpha)}{K-L}\mathbb{I}_{K-L}^\top E_{L+1:K}^\top\right) \\
&= \widetilde{E}_{1:L} E_{1:L}^\top + \widetilde{E}_{L+1:K} E_{L+1:K}^\top \\
&\quad + \frac{1}{K}\left\{ \frac{1}{\beta}\left(\frac{(1-\beta)^2\alpha}{\lambda} - 1\right)\widetilde{E}_{1:L} J_{L,L} E_{1:L}^\top \right. \\
&\qquad + \frac{1}{1-\beta}\left(\frac{\beta^2(1-\alpha)}{\lambda} - 1\right)\widetilde{E}_{L+1:K} J_{K-L,K-L} E_{L+1:K}^\top \\
&\qquad \left. - \beta(1-\alpha)\widetilde{E}_{1:L} J_{L,K-L} E_{L+1:K}^\top - \alpha(1-\beta)\widetilde{E}_{L+1:K} J_{K-L,L} E_{1:L}^\top \right\}, \quad \text{(H.5)}
\end{aligned}
$$

where $\lambda = \sqrt{\alpha^2(1-\beta)^3 + (1-\alpha)^2\beta^3}$, the matrices $R_{L,L-1}$ and $R_{K-L,K-L-1}$ are defined in Proposition J.3, and the second equality results from the following facts

$$
R_{L,L-1} R_{L,L-1}^\top = I_{L,L} - \frac{1}{L}\mathbb{I}_L \mathbb{I}_L^\top,
$$

$$
R_{K-L,K-L-1} R_{K-L,K-L-1}^\top = I_{K-L,K-L} - \frac{1}{K-L}\mathbb{I}_{K-L}\mathbb{I}_{K-L}^\top.
$$

Although the gradient is composed of heterogeneous components from $\widetilde{E}_{1:L}, E_{1:L}$ and $\widetilde{E}_{L+1:K}, E_{L+1:K}$, we can bound the convergence rate of $[f_{-\eta G_{\mathrm{Muon}}}(E_k)]_k$ for any $k$: an upper (resp. lower) bound is obtained by increasing (resp. decreasing) the coefficient of $\widetilde{E}_k E_k^\top$ while decreasing (resp. increasing) that of $\widetilde{E}_{k'} E_k^\top$ for $k' \neq k$. In fact, Eqn. (H.5) implies that there exists a constant $C > 0$ such that the dynamics of the fastest- and slowest-learning triplets are bounded by those along the following two update directions.

$$
-G_{\mathrm{Muon}}^+(W_0) = \left(1 + \frac{2C}{K}\right)(\widetilde{E}_{1:L} E_{1:L}^\top + \widetilde{E}_{L+1:K} E_{L+1:K}^\top) - \frac{C}{K} \cdot \widetilde{E} J_{K,K} E^\top
$$

$$
-G_{\mathrm{Muon}}^-(W_0) = \left(1 - \frac{2C}{K}\right)(\widetilde{E}_{1:L} E_{1:L}^\top + \widetilde{E}_{L+1:K} E_{L+1:K}^\top) + \frac{C}{K} \cdot \widetilde{E} J_{K,K} E^\top.
$$

Concretely, the rate of score increase for the correct object of the $k$-th triplet, which is given by the term $\widetilde{E}_k^\top [-G_{\mathrm{Muon}}(W_0)] E_k$ in the exponent of the softmax score, is bounded. The rate for the fastest-learning triplet is lower-bounded by the corresponding rate derived from $-G_{\mathrm{Muon}}^+(W_0)$, while the rate for the slowest-learning triplet is upper-bounded by that from $-G_{\mathrm{Muon}}^-(W_0)$. Thus, we only need to focus on $G_{\mathrm{Muon}}^+(W_0)$ and $G_{\mathrm{Muon}}^-(W_0)$ to calculate the desired quantity. Following the similar procedures for GD to derive Eqn. (H.4), we have that for any $\eta$ such that $\max_{k\in[K]} \left[f_{W_\eta}(E_k)\right]_k \geq 1 - \epsilon$ (where $W_\eta = W_0 - \eta \cdot G_{\mathrm{Muon}}(W_0)$), the following holds

$$
1 - \min_{k\in[K]} \left[f_{W_\eta}(E_k)\right]_k \leq \frac{\epsilon}{\epsilon + (1-\epsilon)^{r(K)}\epsilon^{1-r(K)}(K-1)^{r(K)-1}}, \quad \text{(H.6)}
$$

where $r(K) = (K - 2C)/(K + 2C)$. We further have that

$$
\begin{aligned}
&(1-\epsilon)^{r(K)}\epsilon^{1-r(K)}(K-1)^{r(K)-1} \\
&= (1-\epsilon)\exp\left(\frac{4C}{K+2C}\left(\log\frac{\epsilon}{1-\epsilon} - \log(K-1)\right)\right) \\
&= (1-\epsilon)\left[1 + \frac{4C}{K+2C}\left(\log\frac{\epsilon}{1-\epsilon} - \log(K-1)\right) + O\left(\frac{(\log K)^2}{K^2}\right)\right] \\
&= (1-\epsilon) + O\left(\frac{\log K}{K}\right), \quad \text{(H.7)}
\end{aligned}
$$

where the first equality results from the basic calculations, the second equality results from that $\exp(x) = 1 + x + O(x^2)$ when $x \to 0$. Combining Eqn. (H.6) and (H.7), we have that

$$\varrho_{\text{Muon}}^{\epsilon} \geq 1 - \epsilon\left(1 + O\left(\frac{\log K}{K}\right)\right).$$

Thus, we prove the desired results for Muon.

**Step 3: Calculations of Adam.**

The proof of the results for Adam is conducted under two cases. We will construct different embeddings $\widetilde{E}$ and $E$ in these two cases. In the first case, we set $\widetilde{E} = E = I_{K,K}$. With such embedding and sufficiently large $K$, we have that

$$-G_{\text{SignGD}}(W_0) = -\text{sign}(\nabla_W \mathcal{L}(W_0)) = 2I_{K,K} - J_{K,K}.$$

Under such a setting, all triplets share the same dynamic. Thus, we have that

$$\varrho_{\text{SignGD}}^{\epsilon} = 1 - \epsilon.$$

In the second case, we set $\widetilde{E}$ and $E$ as block-wise diagonal matrices. Here we set the block size as 3, i.e., requiring that $K \mod 3 = 0$. Such a requirement can be satisfied infinitely often when $K \to \infty$. Then the sufficient and necessary condition of Assumption 4.1 is that each $3 \times 3$ block contains an orthonormal basis. To achieve this, we define the following matrix.

$$R(a,b,c) = \begin{bmatrix} \cos a \cos b \cos c - \sin a \sin c & -\cos a \cos b \sin c - \sin a \cos c & \cos a \sin b \\ \sin a \cos b \cos c + \cos a \sin c & -\sin a \cos b \sin c + \cos a \cos c & \sin a \sin b \\ -\sin b \cos c & \sin b \sin c & \cos b \end{bmatrix}.$$

It is obvious that $R(a,b,c)^\top R(a,b,c) = I_{3,3}$. Then we set $\widetilde{E}$ and $E$ as

$$\widetilde{E} = I_{K/3,K/3} \otimes R(3.638, 2.949, 5.218), \quad E = I_{K/3,K/3} \otimes R(1.715, 0.876, 3.098),$$

where $\otimes$ is the Kronecker product. With these specifications and sufficiently large $K$, the Adam update matrix is

$$-G_{\text{SignGD}}(W_0) = I_{K/3,K/3} \otimes A + J_{K/3,K/3} \otimes B,$$

where $A$ and $B$ are specified as

$$A = \begin{bmatrix} 2 & 0 & 0 \\ 2 & 0 & 2 \\ -2 & -2 & -2 \end{bmatrix}, \quad B = \begin{bmatrix} -1 & -1 & -1 \\ -1 & -1 & -1 \\ 1 & 1 & 1 \end{bmatrix}.$$

These show that the diagonal block of $-G_{\text{SignGD}}(W_0)$ is

$$A + B = \begin{bmatrix} 1 & -1 & -1 \\ 1 & -1 & 1 \\ -1 & -1 & -1 \end{bmatrix}.$$

Since the $k$-th and $(k+3)$-th triplets share the same learning dynamics for all $k \in [K-3]$, we focus on the learning dynamics of $k = 1, 2, 3$. We have that

$$R(3.638, 2.949, 5.218)^\top \cdot (A+B) \cdot R(1.715, 0.876, 3.098)$$
$$= \begin{bmatrix} 1.46552253 & 1.0132908 & -0.11179563 \\ -0.0732561 & 1.00709257 & -1.26935805 \\ 0.0544114 & 0.89611102 & 1.54147329 \end{bmatrix},$$

$$R(3.638, 2.949, 5.218)^\top \cdot B \cdot R(1.715, 0.876, 3.098)$$
$$= \begin{bmatrix} -0.19288146 & -1.24460331 & -1.4058011 \\ -0.20112175 & -1.2977753 & -1.46585978 \\ -0.12780259 & -0.82466989 & -0.93147899 \end{bmatrix}.$$

From the last columns of these two matrices, following the similar procedures for GD to derive Eqn. (H.3), we have that

$$\eta_{\text{SignGD}}^{\epsilon} \leq \frac{1}{1.541 + 0.930} \log\left[(\epsilon^{-1} - 1)(K-1)\right] = \frac{1}{2.471} \log\left[(\epsilon^{-1} - 1)(K-1)\right].$$

Then, from the first columns of these matrices, we have that

$$1 - \min_{k \in [K]} \left[ f_{-\eta^\epsilon_{\text{SignGD}} G_{\text{SignGD}}}(E_k) \right]_k \geq \frac{\epsilon}{\epsilon + (1-\epsilon)^r \epsilon^{1-r} (K-1)^{r-1}},$$

where $r = \frac{1.466 + 0.202}{2.471} = \frac{1.668}{2.471}$.

Thus, we have that

$$\varrho^\epsilon_{\text{SignGD}} \leq O(\epsilon^{-r} K^{r-1}) \leq O(\epsilon^{-0.7} K^{-0.3}).$$

Then we calculate the singular values of $-G_{\text{SignGD}}(W_0)$. We define the eigen vectors of $I_{K,K}$ as $\widetilde{U}$, i.e., $\widetilde{U}^\top I_{K/3,K/3} \widetilde{U} = \text{diag}(K/3, 0 \cdots, 0)$. Using the orthogonal invariance of singular values, $-G_{\text{SignGD}}(W_0)$ shares the singular values with the following matrix

$$(\widetilde{U}^\top \otimes I_{3,3})\big(-G_{\text{SignGD}}(W_0)\big)(\widetilde{U} \otimes I_{3,3})$$
$$= I_{K/3,K/3} \otimes A + (\widetilde{U}^\top I_{K/3,K/3} \widetilde{U}) \otimes B$$
$$= \text{diag}(A - KB/3, A, \cdots, A).$$

Thus, the singular values of $A$ are also the singular values of $G_{\text{SignGD}}(W_0)$. We have that

$$\frac{\sigma_{\min}\big(G_{\text{SignGD}}(W_0)\big)}{\sigma_{\max}\big(G_{\text{SignGD}}(W_0)\big)} \leq \frac{\sigma_{\min}(A)}{\sigma_{\max}(A)} \leq 25\%.$$

Thus, we conclude the proof of Theorem 4.3.

## I  PROOF OF THEOREM 4.4

The proof of Theorem 4.4 takes two steps. In the first step, we derive the share form of $W_t$ along the whole optimization trajectory. In the second step, we build the desired results on the basis of step 1. Throughout the proof, we will write $W_t^{\text{Muon}}$ as $W_t$ for the ease of presentation.

**Step 1: Derive the shared forms of $W_t$ and $G_{\text{Muon}}$.**

We will derive the forms of $W_t$ along the optimization trajectory via the induction method. We first state our hypothesis and then prove it.

**Hypothesis 1** . For any optimization step index $t \in [T]$, the parameters $W_t$ can be expressed as

$$W_t = \widetilde{E} X_t E, \quad X_t = \Lambda_t + C_t,$$

where $\Lambda_t$ and $C_t$ are

$$\Lambda_t = \text{diag}(a_t \cdot \mathbb{I}_L, b_t \cdot \mathbb{I}_{K-L}), \quad C_t = \begin{bmatrix} c_t^{11} \cdot J_{L,L} & c_t^{12} \cdot J_{L,K-L} \\ c_t^{21} \cdot J_{K-L,L} & c_t^{22} \cdot J_{K-L,K-L} \end{bmatrix},$$

where $a_t, b_t, c_t^{11}, c_t^{12}, c_t^{21}, c_t^{22} \in \mathbb{R}$ are real numbers such that (1) $a_t = b_t \geq 0$, and (2) $c_t^{ij} = O(a_t/K)$ for $i, j \in [2]$.

When $t = 0$, it is obvious to verify that $W_0 = 0_{d_o,d_s}$ satisfying this hypothesis with $a_t = b_t = c_t^{11} = c_t^{12} = c_t^{21} = c_t^{22} = 0$. Then we assume that this hypothesis holds for $\{1, \cdots, t\}$, and we will prove that it holds for $t + 1$. Since $W_{t+1} = W_t - \eta_{t+1} U_t \text{norm}(\Sigma_t) V_t^\top$, we need to show that $-\eta_{t+1} U_t \text{norm}(\Sigma_t) V_t^\top$ satisfies the hypothesis. We define the score of $k'$-th object for the $k$-th subject-relation pair with the parameter $W$ as

$$s(k', k, W) = \frac{\exp(\widetilde{E}_{k'}^\top W E_k)}{\sum_{k''=1}^K \exp(\widetilde{E}_{k''}^\top W E_k)}.$$

According to the symmetry of $W_t$, we have that

- $s(k, k, W_t) = s(1, 1, W_t)$ for all $k \leq L$.
- $s(k, k, W_t) = s(K, K, W_t)$ for all $k > L$.

- $s(k', k, W_t) = s(2, 1, W_t)$ for all $k, k' \leq L, k' \neq k$.
- $s(k', k, W_t) = s(K, 1, W_t)$ for all $k \leq L, k' > L$.
- $s(k', k, W_t) = s(K - 1, K, W_t)$ for all $k, k' > L, k' \neq k$.
- $s(k', k, W_t) = s(1, K, W_t)$ for all $k > L, k' \leq L$.

Thus, Proposition J.1 shows that the gradient of $W_t$ is

$$-\nabla_W \mathcal{L}(W_t) = \widetilde{E}(\Gamma_t + B_t)E^\top,$$

where $\Gamma_t$ and $B_t$ are defined as

$$\Gamma_t = \mathrm{diag}\bigg(\frac{\alpha}{L}\big(1 + s(2, 1, W_t) - s(1, 1, W_t)\big)\mathbb{I}_L,$$

$$\frac{1-\alpha}{K-L}\big(1 + s(K-1, K, W_t) - s(K, K, W_t)\big)\mathbb{I}_{K-L}\bigg),$$

$$B_t = \begin{bmatrix} -\frac{\alpha}{L}s(2, 1, W_t) \cdot J_{L,L} & -\frac{1-\alpha}{K-L}s(1, K, W_t) \cdot J_{L,K-L} \\ -\frac{\alpha}{L}s(K, 1, W_t) \cdot J_{K-L,L} & -\frac{1-\alpha}{K-L}s(K-1, K, W_t) \cdot J_{K-L,K-L} \end{bmatrix}.$$

Thus, Proposition J.2 shows that

$$-G_{\mathrm{Muon}}(W_t) = \widetilde{E}\bigg(\mathrm{diag}(\mathbb{I}_K) + \begin{bmatrix} C_{11} \cdot J_{L,L} & C_{12} \cdot J_{L,K-L} \\ C_{21} \cdot J_{K-L,L} & C_{22} \cdot J_{K-L,K-L} \end{bmatrix}\bigg)E^\top,$$

where

$$C_{11} = \frac{\widetilde{U}_{1,1}\widetilde{V}_{1,1} + \widetilde{U}_{1,2}\widetilde{V}_{1,2} - 1}{\beta K}, \qquad\qquad C_{12} = \frac{\widetilde{U}_{1,1}\widetilde{V}_{2,1} + \widetilde{U}_{1,2}\widetilde{V}_{2,2}}{\sqrt{\beta(1-\beta)}K},$$

$$C_{21} = \frac{\widetilde{U}_{2,1}\widetilde{V}_{1,1} + \widetilde{U}_{2,2}\widetilde{V}_{1,2}}{\sqrt{\beta(1-\beta)}K}, \qquad\qquad C_{22} = \frac{\widetilde{U}_{2,1}\widetilde{V}_{2,1} + \widetilde{U}_{2,2}\widetilde{V}_{2,2} - 1}{(1-\beta)K}.$$

where $\widetilde{U}, \widetilde{V} \in \mathbb{R}^{2\times 2}$ are the orthonormal matrices defined in Proposition J.2. Since $W_{t+1} = W_t - \eta_{t+1}G_{\mathrm{Muon}}(W_t)$, it is obvious that $a_{t+1} = b_{t+1}$. The orthonormality of $\widetilde{U}$ and $\widetilde{V}$ implies that $|\widetilde{U}_{i,j}|, |\widetilde{V}_{i,j}| \leq 1$. Thus, we have

$$\frac{\widetilde{U}_{1,1}\widetilde{V}_{1,1} + \widetilde{U}_{1,2}\widetilde{V}_{1,2} - 1}{\beta K} = O\bigg(\frac{1}{K}\bigg).$$

This further implies that $c_{t+1}^{1,1} = O(a_{t+1}/K)$. The proofs for other $c_{t+1}^{ij}$ are similar. This completes the proof.

**Step 2: Establish the convergence results.**

We note that this analysis is very similar to the proof of Muon in Theorem 4.3. Concretely, for $W_t$, the coefficients $a_t, b_t, c_t^{11}, c_t^{12}, c_t^{21}, c_t^{22}$ from multiple-step optimization share the same property with those of the one-step results. It means that there exists a constant $C > 0$ such that the dynamics of the fastest- and slowest-learning triplets are bounded by those along the following two update directions in only one step.

$$-G_{\mathrm{Muon}}^+ = \bigg(1 + \frac{2C}{K}\bigg)(\widetilde{E}_{1:L}E_{1:L}^\top + \widetilde{E}_{L+1:K}E_{L+1:K}^\top) - \frac{C}{K} \cdot \widetilde{E}J_{K,K}E^\top$$

$$-G_{\mathrm{Muon}}^- = \bigg(1 - \frac{2C}{K}\bigg)(\widetilde{E}_{1:L}E_{1:L}^\top + \widetilde{E}_{L+1:K}E_{L+1:K}^\top) + \frac{C}{K} \cdot \widetilde{E}J_{K,K}E^\top.$$

The remaining analysis is then exactly the same as that of Theorem 4.3. Thus, we conclude the proof of Theorem 4.4.

## J  SUPPORTING PROPOSITIONS

**Proposition J.1.** We define the score of $k'$-th object for the $k$-th subject-relation pair with the parameter $W$ as

$$s(k', k, W) = \frac{\exp(\widetilde{E}_{k'}^\top W E_k)}{\sum_{k''=1}^K \exp(\widetilde{E}_{k''}^\top W E_k)}.$$

When the parameter $W$ is trained with loss

$$\mathcal{L}(W) = -\sum_{k=1}^K p_k \cdot \log\left[f_W(E_k)\right]_k,$$

the gradient of $W$ is

$$\nabla_W \mathcal{L}(W) = -\sum_{k=1}^K p_k \left\{ \left[1 - s(k, k, W)\right]\widetilde{E}_k E_k^\top - \sum_{k' \neq k} s(k', k, W)\widetilde{E}_{k'} E_k^\top \right\}.$$

*Proof of Proposition J.1.* The proof just follows from the basic calculus. Thus, we omit them here.  □

**Proposition J.2.** Let $X = \Lambda + C \in \mathbb{R}^{K \times K}$. The matrix $\Lambda = \mathrm{diag}(a \cdot \mathbb{I}_L, b \cdot \mathbb{I}_{K-L})$ is a diagonal matrix whose first $L$ diagonal elements are $a$ and the last $K - L$ elements are $b$ with $a, b > 0$. The matrix $C$ is a block-wise constant matrix defined as

$$C = \begin{bmatrix} c_{11} \cdot J_{L,L} & c_{12} \cdot J_{L,K-L} \\ c_{21} \cdot J_{K-L,L} & c_{22} \cdot J_{K-L,K-L} \end{bmatrix}.$$

Then $X = U\Sigma V^\top$. Here $\Sigma, V, U$ are defined as follows. All of them can be decomposed into three blocks, each corresponding to a subspace. The first subspace is

$$\mathcal{S}_1 = \left\{ \begin{bmatrix} x \\ 0_{K-L} \end{bmatrix} \,\middle|\, x^\top \mathbb{I}_L = 0, \text{ and } x \in \mathbb{R}^L \right\}.$$

The dimension of this space is $L - 1$. The singular value of $X$ corresponding to this subspace is $a$. The block of columns in both $U$ and $V$ that forms an orthonormal basis for this subspace is given by

$$\begin{bmatrix} R_{L,L-1} \\ 0_{K-L,L-1} \end{bmatrix},$$

where the columns of the matrix $R_{L,L-1} \in \mathbb{R}^{L \times (L-1)}$ form an orthonormal basis for the subspace $\{x \in \mathbb{R}^L | x^\top \mathbb{I}_L = 0\}$. The second subspace is

$$\mathcal{S}_2 = \left\{ \begin{bmatrix} 0_L \\ y \end{bmatrix} \,\middle|\, y^\top \mathbb{I}_{K-L} = 0, \text{ and } y \in \mathbb{R}^{K-L} \right\}.$$

The dimension of this space is $K - L - 1$. The singular value of $X$ corresponding to this subspace is $b$. The block of columns in both $U$ and $V$ that forms an orthonormal basis for this subspace is given by

$$\begin{bmatrix} 0_{L,K-L-1} \\ R_{K-L,K-L-1} \end{bmatrix},$$

where the columns of the matrix $R_{K-L,K-L-1} \in \mathbb{R}^{(K-L) \times (K-L-1)}$ form an orthonormal basis for the subspace $\{y \in \mathbb{R}^{K-L} | y^\top \mathbb{I}_{K-L} = 0\}$. The remaining 2-dimensional subspace is induced by a $2 \times 2$ matrix $M$ defined as

$$M = \begin{bmatrix} \alpha & \beta \\ \gamma & \delta \end{bmatrix} = \widetilde{U} \mathrm{diag}(s_1, s_2) \widetilde{V}^\top,$$

where the elements of $M$ are defined as

$$\alpha = a + Lc_{11}, \quad \beta = \sqrt{L(K-L)}\, c_{12}, \quad \gamma = \sqrt{L(K-L)}\, c_{21}, \quad \delta = b + (K-L)c_{22}.$$

The singular values $s_1, s_2$ are

$$s_{1,2} = \sqrt{\frac{T \pm \sqrt{T^2 - 4\Delta}}{2}}, \quad T = \alpha^2 + \beta^2 + \gamma^2 + \delta^2, \quad \Delta = (\alpha\delta - \beta\gamma)^2.$$

The singular values of $X$ in this subspace are $s_1$ and $s_2$. The corresponding right singular vectors ($v_i$) and left singular vectors ($u_i$), which form columns of $V$ and $U$ respectively, are given by:

$$v_i = \widetilde{V}_{1,i}e_1 + \widetilde{V}_{2,i}e_2, u_i = \widetilde{U}_{1,i}e_1 + \widetilde{U}_{2,i}e_2 \text{ for } i = 1, 2,$$

where the vectors $e_1$ and $e_2$ are defined as

$$e_1 = \begin{bmatrix} \frac{1}{\sqrt{L}}\mathbb{I}_L \\ 0_{K-L} \end{bmatrix}, \quad e_2 = \begin{bmatrix} 0_L \\ \frac{1}{\sqrt{K-L}}\mathbb{I}_{K-L} \end{bmatrix}.$$

In summary, the SVD of $X$ is

$$\Sigma = \mathrm{diag}(a \cdot \mathbb{I}_{L-1}, b \cdot \mathbb{I}_{K-L-1}, s_1, s_2),$$

$$V = \left[ \begin{bmatrix} R_{L,L-1} \\ 0_{K-L,L-1} \end{bmatrix}, \begin{bmatrix} 0_{L,K-L-1} \\ R_{K-L,K-L-1} \end{bmatrix}, v_1, v_2 \right],$$

$$U = \left[ \begin{bmatrix} R_{L,L-1} \\ 0_{K-L,L-1} \end{bmatrix}, \begin{bmatrix} 0_{L,K-L-1} \\ R_{K-L,K-L-1} \end{bmatrix}, u_1, u_2 \right].$$

*Proof of Proposition J.2.* We first prove the results for $\mathcal{S}_1$. For any vector $v$ in $\mathcal{S}_1$, it is direct to verify that

$$X^\top X \begin{bmatrix} v \\ 0_{K-L} \end{bmatrix} = a^2 \begin{bmatrix} v \\ 0_{K-L} \end{bmatrix}.$$

Thus, the singular value of $X$ corresponding to the subspace spanned by the vector $[v^\top, 0_{K-L}^\top]^\top$ is $a$, and the corresponding columns of $V$ form an orthonormal basis for $\mathcal{S}_1$. For the $U$ calculation, we have that

$$X \begin{bmatrix} v \\ 0_{K-L} \end{bmatrix} = a \begin{bmatrix} v \\ 0_{K-L} \end{bmatrix}.$$

Thus, the corresponding left singular vectors (columns of U) are identical to the right singular vectors for this subspace. A similar calculation can be done for $\mathcal{S}_2$. The remaining vectors are orthogonal to both $\mathcal{S}_1$ and $\mathcal{S}_2$ and thus take the form of

$$v_i = p_1 e_1 + p_2 e_2, \quad u_i = p_3 e_1 + p_4 e_2 \text{ for } i = 1, 2 \text{ with } p_1, p_2, p_3, p_4 \in \mathbb{R}.$$

By solving the equation $X^\top X v_i = \lambda v_i$, we can show that the corresponding singular values and coefficients $p_1, p_2, p_3, p_4$ coincide with those in the SVD of $M$, as can be verified by simple calculations. Thus, we conclude the proof of Proposition J.2. $\square$

**Proposition J.3.** Let $x = [a \cdot \mathbb{I}_L^\top, b \cdot \mathbb{I}_{K-L}^\top]^\top \in \mathbb{R}^K$, and $X = \mathrm{diag}(x) - K^{-1}\mathbb{I}_K \cdot x^\top \in \mathbb{R}^{K \times K}$, where $a, b > 0$. Then the SVD of $X = U\Sigma V^T$ is that

$$\Sigma = \mathrm{diag}\left(a \cdot \mathbb{I}_{L-1}, b \cdot \mathbb{I}_{K-L-1}, \sqrt{\frac{a^2 \cdot (K-L) + b^2 \cdot L}{K}}, 0\right),$$

$$V = \left[ \begin{bmatrix} R_{L,L-1} \\ 0_{K-L,L-1} \end{bmatrix}, \begin{bmatrix} 0_{L,K-L-1} \\ R_{K-L,K-L-1} \end{bmatrix}, v_1, v_2 \right],$$

$$U = \left[ \begin{bmatrix} R_{L,L-1} \\ 0_{K-L,L-1} \end{bmatrix}, \begin{bmatrix} 0_{L,K-L-1} \\ R_{K-L,K-L-1} \end{bmatrix}, u_1, u_2 \right].$$

Here, the columns of the matrix $R_{L,L-1} \in \mathbb{R}^{L \times (L-1)}$ form an orthonormal basis for the subspace of vectors in $\mathbb{R}^L$ orthogonal to $\mathbb{I}_L$. Similarly, the columns of $R_{K-L,K-L-1} \in \mathbb{R}^{(K-L) \times (K-L-1)}$ form

an orthonormal basis for the subspace of vectors in $\mathbb{R}^{K-L}$ orthogonal to $\mathbb{I}_{K-L}$. These correspond to the subspaces $\mathcal{S}_1$ and $\mathcal{S}_2$ defined as:

$$\mathcal{S}_1 = \left\{ \begin{bmatrix} x \\ 0_{K-L} \end{bmatrix} \,\middle|\, x^\top \mathbb{I}_L = 0, \text{ and } x \in \mathbb{R}^L \right\}, \quad \mathcal{S}_2 = \left\{ \begin{bmatrix} 0_L \\ y \end{bmatrix} \,\middle|\, y^\top \mathbb{I}_{K-L} = 0, \text{ and } y \in \mathbb{R}^{K-L} \right\}.$$

The vectors $v_1, v_2, u_1, u_2$ are

$$v_1 = \frac{1}{\sqrt{a^2(K-L)+b^2L}} \left( \frac{a\sqrt{K-L}}{\sqrt{L}} \begin{bmatrix} \mathbb{I}_L \\ 0_{K-L} \end{bmatrix} - \frac{b\sqrt{L}}{\sqrt{K-L}} \begin{bmatrix} 0_L \\ \mathbb{I}_{K-L} \end{bmatrix} \right)$$

$$v_2 = \frac{1}{\sqrt{a^2(K-L)+b^2L}} \left( b \begin{bmatrix} \mathbb{I}_L \\ 0_{K-L} \end{bmatrix} + a \begin{bmatrix} 0_L \\ \mathbb{I}_{K-L} \end{bmatrix} \right)$$

$$u_1 = \frac{1}{\sqrt{KL(K-L)}} \left( (K-L) \begin{bmatrix} \mathbb{I}_L \\ 0_{K-L} \end{bmatrix} - L \begin{bmatrix} 0_L \\ \mathbb{I}_{K-L} \end{bmatrix} \right)$$

$$u_2 = \frac{1}{\sqrt{K}} \mathbb{I}_K.$$

*Proof of Proposition J.3.* This proposition is a direct corollary of Proposition J.2. The matrix $X = \mathrm{diag}(x) - K^{-1}\mathbb{I}_K \cdot x^\top$ is an instance of the general form $\Lambda + C$ from Proposition J.2.

The diagonal part is $\Lambda = \mathrm{diag}(x) = \mathrm{diag}(a \cdot \mathbb{I}_L, b \cdot \mathbb{I}_{K-L})$. The off-diagonal part is $C = -K^{-1}\mathbb{I}_K \cdot x^\top$. We can write $C$ in block form:

$$C = -\frac{1}{K} \begin{bmatrix} \mathbb{I}_L \\ \mathbb{I}_{K-L} \end{bmatrix} \begin{bmatrix} a\mathbb{I}_L^\top & b\mathbb{I}_{K-L}^\top \end{bmatrix} = -\frac{1}{K} \begin{bmatrix} aJ_{L,L} & bJ_{L,K-L} \\ aJ_{K-L,L} & bJ_{K-L,K-L} \end{bmatrix}.$$

This corresponds to setting the block-wise constants in Proposition J.2 to:

$$c_{11} = -a/K, \quad c_{12} = -b/K, \quad c_{21} = -a/K, \quad c_{22} = -b/K.$$

Substituting these into the formulas for $\alpha, \beta, \gamma, \delta$ from Proposition J.2 gives:

$$\alpha = a + L(-a/K) = a(K-L)/K$$
$$\beta = \sqrt{L(K-L)}(-b/K)$$
$$\gamma = \sqrt{L(K-L)}(-a/K)$$
$$\delta = b + (K-L)(-b/K) = bL/K$$

These coefficients define the $2 \times 2$ matrix $M$ from Proposition J.2 for this specific case. We now analyze this matrix $M$. A key observation is that its determinant is zero:

$$\det(M) = \alpha\delta - \beta\gamma = \frac{a(K-L)}{K}\frac{bL}{K} - \left( \frac{L(K-L)}{K^2} \right)(-b)(-a) = 0.$$

Since the determinant is zero, one of its singular values must be zero. The other singular value, $s_1$, can be calculated from the squared Frobenius norm (sum of squares of elements), which is also the sum of squared singular values ($s_1^2 + s_2^2$):

$$s_1^2 + 0^2 = \alpha^2 + \beta^2 + \gamma^2 + \delta^2 = \frac{a^2(K-L)^2}{K^2} + \frac{L(K-L)b^2}{K^2} + \frac{L(K-L)a^2}{K^2} + \frac{b^2L^2}{K^2}$$
$$= \frac{a^2(K-L)+b^2L}{K}.$$

This confirms the singular values stated in the proposition. The singular vectors $v_1, v_2, u_1, u_2$ can be derived by performing the SVD on this specific $2 \times 2$ matrix $M$. $\qquad\square$

## K  HEAVY-TAILEDNESS OF GRADIENT OF LLMS

In this section, we discuss how our insight about the gradient in the one-layer model generalizes to the multi-layer model. In the following analysis, we focus on the FFN modules in the model, and

the attention module can be similarly analyzed. The illustration of this multi-layer model of FFN modules is shown in the following Figure 19.

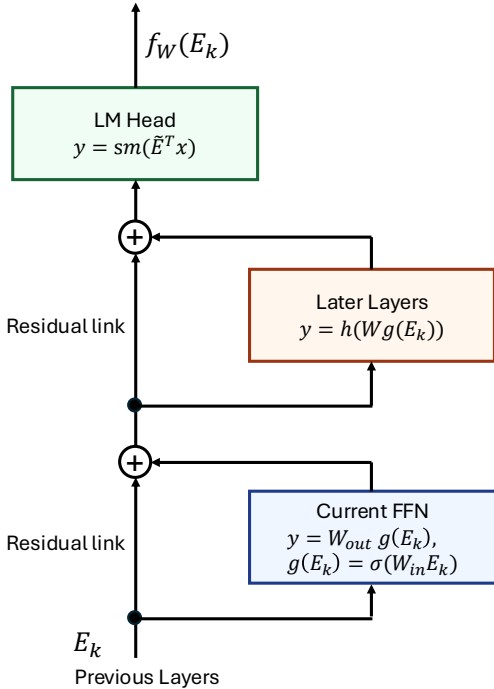

Figure 19: The illustration of the multi-layer model of FFN modules.

We abstract the feature extraction role of all the previous layers and the in-projection of the current FFN as a function as follows.

$$y = W_{\text{out}}\sigma(W_{\text{in}}x) = W_{\text{out}}g(x),$$

where $W_{\text{out}}, W_{\text{in}} \in \mathbb{R}^{d \times d}$ are weight matrices, $x \in \mathbb{R}^d$ is the output of all the previous layers, and $g : \mathbb{R}^d \to \mathbb{R}^d$ abstracts the role of feature learned in $W_{\text{in}}$. Abstracting all the later layers as a function $h : \mathbb{R}^d \to \mathbb{R}^d$, the function $h$ may also take all previous tokens as inputs, which we omit from the notation for brevity. The whole model is written as

$$f_W(E_k) = \text{sm}\left( \widetilde{E}^\top \left[ Wg(E_k) + h\big(Wg(E_k)\big) \right] \right),$$

where $\widetilde{E} \in \mathbb{R}^{d \times K}$ is the parameter of the language model head, $K$ is the alphabet size, and $E_k$ is the hidden state of the last token in the training context that precedes the $k$-th token in the alphabet, at the layer where associative memory is present. Without loss of generality, we assume that the next token is the $k$-th token in $\widetilde{E}$. Then the loss function on the pretraining data is

$$\mathcal{L}(W) = -\sum_{k=1}^{K} p_k \log[f_W(E_k)]_k,$$

where $p_k$ is the frequency of $k$-th token. We note that this is a simplification of what happens in the pretraining, where the frequencies of token associations instead of the single token matter. However, such simplification does not influence our main message. In the heavy-tailed dataset, e.g., WikiText103, $p_k$ decays as $p_k = \alpha \cdot k^{-1}$ for $k \in [K]$. Then the gradient of $W$ is

$$\nabla_W \mathcal{L}(W) = -\sum_{k=1}^{K} p_k \nabla_W \log[f_W(E_k)]_k,$$

$$\nabla_W \log[f_W(E_k)]_k = \left( I_{d,d} + J_h\big(Wg(E_k)\big) \right)^\top \left[ \widetilde{E}_k g(E_k)^\top - \sum_{i=1}^{K} [f_W(E_k)]_i \cdot \widetilde{E}_i g(E_k)^\top \right],$$

where $I_{d,d} \in \mathbb{R}^{d \times d}$ is the identity matrix and $J_h$ is the Jacobian of the function $h$. Two structural properties of the gradient are worth highlighting. First, $\nabla_W \mathcal{L}(W)$ is heavy-tailed, since it is a weighted sum of per-token gradients with geometrically decaying weights $p_k = \alpha \cdot k^{-1}$ for $k \in [K]$. Second, the gradient decomposes as a sum of outer products $\widetilde{E}_k \, g(E_k)^\top$. Our theoretical analysis in Section 4 focuses on the simplified setting $h = 0$ and $g(x) = x$. For general $h$ and $g$, the Jacobian $J_h$ acts as a preconditioner on the gradient, and each outer product is formed between the feature in the language-model head $\widetilde{E}$ and the transformed feature $g$ in each layer. Thus, our intuition extends to this more general multi-layer setting.

