# OpenReview forum: "Muon Outperforms Adam in Tail-End Associative Memory Learning"
_ICLR.cc/2026/Conference — ICLR 2026 Poster_

### Official Review · Reviewer_sugx · 2025-10-15

**Soundness:** 3
**Presentation:** 4
**Contribution:** 3
**Rating:** 6
**Confidence:** 4

**Summary:**

The paper studies why Muon often outperforms Adam in LLM-pretraining. The central claim is that Muon’s advantage concentrates in associative-memory components, specifically the Value/Output (VO) attention matrices and FFNs. Muon's update rule yields a more isotropic singular spectrum, which in turn improves learning on tail classes in heavy-tailed distributions. The authors support this with
1. component-wise optimizer ablations in Transformers,showing that VO+FFN accounts for most of the validation-loss gains for Muon.
2. measurements of more isotropic singular-value spectra under Muon than under Adam.
3. a knowledge-intensive QA task reflecting heavy-tailed data where Muon shows larger gains on tail examples.
4. a theoretical analysis of a one-layer linear associative memory under class imbalance showing Muon’s superiority.

**Strengths:**

1. **Focused component ablations.** The blockwise ablation is informative and directly isolates VO/FFN as primary beneficiaries, supporting the associative-memory lens.

2. **Heavy-tail perspective with tail-class gains**. The paper emphasizes tail performance and links it to spectral isotropy, providing a plausible causal chain: Muon → more balanced singular spectrum → better tail learning.

3. **Theory that matches the story.** The one-layer associative-memory model with class imbalance theoretically explains why Muon outperforms Adam under heavy-tailed class distributions, through the model is simple.

4. **Muon wins over Adam, and community does not know why!**
5. **LLM settings are diverse.** Covers both gated and non-gated 160M/700M LLM.

**Weaknesses:**

1. **Multiple kinds of heavy-tailness** The paper presents servral kinds of heavy-tailness: data, knowledge, parameter and update. There are evidence in the paper to connect the heavy-tailness of data with the heavy-tailness of knowledge, there are also evidence to connect the heavy-tailness of parameter with the heavy-tailness of update. However, it remains unclear how to connect the heavy-tailness of data and knowledge with the heavy-tailness of parameter and update in the LLM.
2. **$W_V$ as associative memory** Prior work rarely frames $W_V$ as an associative-memory parameter. Although the paper argues $W_V$ and $W_O$ play symmetric roles, that symmetry breaks under MQA/GQA. Treating VO jointly is understandable as a number of literatures[2][3] say that VO(Value-Output) are similar and can be considered together in practice even under the context of MQA/GQA, but a restatement is needed.
3. **Observation 2 seems to be known in literature** The figure 4 in [3] already points out Observation 2 that Muon consistently yields more isotropic weight matrices with broadly distributed spectral energy than Adam.
4. **FFN over QK may be the result of parameter counting** FFN over QK may be the result of parameter counting, not because of associative memory.
5. **Reproducibility** The `model` are missing from the released code, making the work not fully reproducible.

[1] Lin, C. H., Gao, S., Smith, J. S., Patel, A., Tuli, S., Shen, Y., ... & Hsu, Y. C. MoDeGPT: Modular Decomposition for Large Language Model Compression. In The Thirteenth International Conference on Learning Representations. \
[2] Wang, J., Wang, M., Zhou, Z., Yan, J., & Wu, L. The Sharpness Disparity Principle in Transformers for Accelerating Language Model Pre-Training. In Forty-second International Conference on Machine Learning. \
[3] Liu, J., Su, J., Yao, X., Jiang, Z., Lai, G., Du, Y., ... & Yang, Z. (2025). Muon is scalable for LLM training. arXiv preprint arXiv:2502.16982.

**Questions:**

1. **Capacity control** If you equalize parameter counts (e.g., only optimize $W_{in}$ parameter in some depth by Adam), do Muon-over-Adam gaps persist in QK vs. FFN?
2. **Multiple kinds of heavy-tailness** Can you connect the heavy-tailness of data and knowledge with the heavy-tailness of parameter and update in the context of LLM?
3. **Logic gap on associative-memory mechanism** The paper does not yet provide strong causal evidence that Muon surpasses Adam *specifically* because of associative-memory effects. The superiority on the knowledge-intensive QA task might not be attributable to associative memory per se. VO+FFN may contribute more ('may' is due to weakness 2 and 4), but the mechanistic link to associative memory remains suggestive rather than conclusive.

Overall, the paper discusses a vital problem in the optimizer community. However, currentlymy tentative recommendation is borderline Weak Accept/Accept. I’m open to revising my score if these concerns are addressed.

---

> ### Author Response · Authors · 2025-11-20
> **Official Comment by Authors (Round 1, Part 1/3)**
>
> We thank the reviewer for your positive review and suggestions, as well as many insightful suggestions. Below we respond to the comments in Weakness (W) and Questions (Q).
>
> ---
>
> ### **W1 & Q2. How to connect the heavy-tailness of data and knowledge with the heavy-tailness of parameter and update in the LLM?**
>
> Thank you for your insight comment.  We show the step-by-step logic that how to connect the heavy-tailness of data and knowledge with the heavy-tailness of parameter and update as follows:
>
> * **Data/Knowledge Tail $\rightarrow$ Gradient Tail**
> The chain starts with the data. Let the frequency of the $k$-th fact be $p_k$, following a heavy-tailed power law ($p_1 \gg p_K$). In our associative memory model, this data imbalance directly creates a heavy-tailed singular value spectrum in the gradient matrix $G = \nabla_W \mathcal{L}$. As derived in Proposition J.1, the gradient is dominated by the term:$G \approx -\sum_{k} p_k \tilde{E}_k E_k^\top$. Since the embeddings $E, \tilde{E}$ are near-orthogonal, the singular values of the gradient are directly determined by the data frequencies: $\sigma_k(G) \approx p_k$. Thus, heavy-tailed data ($p_k$) $\Rightarrow$ heavy-tailed gradient spectrum ($\sigma_k(G)$).
>
> * **Gradient Tail $\rightarrow$ Parameter/Update Tail (via Optimizer)**
> The optimizer determines the shape of the learned parameters because **parameters are simply the accumulation of updates over time** ($W_T = \sum_{t} \Delta W_t$). The update rule filters the gradient spectrum differently:
>
>   + **Muon** constructs the update using only the orthogonal vectors of the gradient ($G=U\Sigma V^\top$). This forces all singular values of the update to be uniform ($\sigma_k(\Delta W) = 1$). By accumulating these isotropic updates, the resulting weight matrix $W$ becomes near-isotropic.
>   + **Adam** is element-wise and is not aware of the matrix's spectral properties. This characteristic means it can preserve or even amplify the gradient's spectral skew, resulting in a highly anisotropic update that exhibits significant spectral decay.
>
> * **The Consequence: Balanced vs. Imbalanced Learning**
> This mechanical difference in the update (isotropic vs. anisotropic) directly causes the performance difference.
>
>   + Because **Muon's update** is isotropic, it applies equal "learning strength" to all facts and guarantees balanced learning.
>   + Because **Adam's update** is anisotropic, it focuses learning on the "head"  and fails to learn the "tail", resulting in imbalanced learning.
>
>
> Together, we make the full chain explicit:
> $\text{heavy‑tailed  data}\ \Rightarrow\ \text{heavy-tailed singular value spectrum in Gradient}$ $\xrightarrow{\text{update rule}}\
> \text{(near‑isotropic parameters (Muon), anisotropic parameters (Adam))}$\
> $\Rightarrow\ \text{balanced vs. imbalanced head–tail recall.}$
>
> ---
>
> ### **W2. Treating VO jointly is understandable as a number of literatures say that VO are similar and can be considered together in practice even under the context of MQA/GQA, but a restatement is needed.**
>
> We thank for this clarification. We modify this statement as follows. "In the attention module, (Bietti et al,2023) showed that the parameter $W_{O}$ can serve as a linear associative memory when $W_{V}$ is fixed. Since $W_{O}$ and $W_{V}$ play symmetric roles, we also treat $W_{V}$ as part of the associative memory parameters. It is therefore natural to consider VO jointly: several works (Lin et al,2024, Wang et al,2025) have shown that the value and output matrices play similar roles and can be analyzed together in practice, even in multi-query attention (MQA) and grouped-query attention (GQA) settings." in Line 133-136.
>
>
> ---

---

> ### Author Response · Authors · 2025-11-20
> **Official Comment by Authors (Round 1, Part 2/3)**
>
> ### **W3. Observation 2 seems to be known in literature The figure 4 in Liu et al. (2025) already points out**
>
> We thank the reviewer for this important comment. We agree that Liu et al. (2025) also reported that Muon can yield more isotropic weight spectra. However, **we respectfully clarify that our contribution is not only the observation of isotropy itself, but rather a deeper, mechanistic explanation of where this isotropy matters and why it emerges—a gap that Liu et al. (2025) does not address**.
>
> From the view of novel empirical findings, we have these two findings not cover by Liu et al. (2025).
>
> * Stability Focus: We are the first to investigate and demonstrate Muon's robust stability across random initializations (e.g., the tight error bars in Figure 2), in sharp contrast to Adam's high instability on this task—a finding we also connect to our theoretical results.
>
> * Precise Localization and Functional Disentanglement: While Liu et al. (2025) report spectral entropy for component groups such as AttnQO and AttnKV, their categorization combined functionally distinct parameters. Specifically, they group Query with Output, and Key with Value. This grouping obscures the critical mechanistic distinction between QK and associative memory (VO).
>
>     In contrast, our work disentangles these components. We specifically isolate VO from QK. Our component-wise ablations in Figure 1 show that the performance benefits stem almost entirely from the VO and FFN components. Applying Muon to QK contributes little. This functional localization is a key distinction not made in prior work.
>
> From the view of theoretical understanding, this serves as a firm basis.
>
> * Our paper is the first to provide a mechanistic link from the heavy-tailed nature of knowledge (Observation 3) to this isotropic spectra (Observation 2). We show that isotropy is not just a statistical artifact but a direct consequence of Muon's update rule, which is better aligned with the outer-product structure of associative memories. Our theoretical analysis (Theorems 4.3) proves this mechanism enables Muon to learn tail classes more effectively and uniformly than Adam, which explains why Muon's performance is superior.
>
> ---
>
> ### **W4 & Q1. FFN over QK may be the result of parameter counting**
>
> Thanks for your insightful comment.  We don't directly compare the importance between FFN with QK. Throughout our paper, we claim that **VO parameters are more important than QK paramters within attention metrics**.
>
> To rule out the possibility that the observed gains are simply due to parameter counting, we have added a new analysis in Appendix G.2 (see Figure 5 in the revised paper) that explicitly controls for parameter count in two complementary ways. First, Figure 5(a) reports the validation-loss gain of Muon over Adam **normalized by the number of parameters** in each component. For the 160M model, the parameter counts are related as follows: $|W_V| = |W_O| = |W_Q| = |W_K|$ and $|W_{\text{in}}| = |W_{\text{out}}| = 4 \times |W_Q|$.
>
> The results show that the normalized gain for VO attention is approximately $5\times$ larger than that for QK attention, even though both components have the same number of parameters. The normalized gains for $W_{\text{in}}$ and $W_{\text{out}}$ are also substantially higher (over $3\times$) than for QK; although $W_{\text{in}}$ and $W_{\text{out}}$ have twice as many parameters as QK, their **per-parameter** gains are far more than half of QK’s gain.
>
> Second, Figure 5(b) holds the number of Muon-optimized parameters **exactly equal** across components. We compare configurations where Muon is applied to (i) all QK matrices, or (ii) $W_{\text{in}}$ and $W_{\text{out}}$ only in the odd-numbered layers, so that the total number of parameters trained with Muon is identical in each case. Even under this equal-parameter budget, applying Muon to $W_{\text{in}}$ or $W_{\text{out}}$ yields much larger validation-loss improvements than applying it to QK.
>
> Taken together, these analyses show that Muon’s advantage on VO and FFN parameters cannot be explained by parameter counting alone; rather, it is closely tied to the specific associative-memory role of these parameters.
>
> ---
>
> ### **W5. The model are missing from the released code.**
>
> Thanks for your suggestion. We add the model into the Supplementary document.

---

> ### Author Response · Authors · 2025-11-20
> **Official Comment by Authors (Round 1, Part 3/3)**
>
> ### **Q3. Logic gap on associative-memory mechanism.**
>
> The mechanistic link between Muon’s superiority and associative memory is **conclusive**, and we investigate it using the **hypothetico-deductive method**: we propose a concrete hypothesis, deduce testable predictions, and evaluate the hypothesis by checking whether these predictions match empirical observations.
>
> Our hypothesis about the causal relationship between Muon’s superiority and associative memory is that **Muon normalizes the amplitude of each outer product in the SVD of the gradient—so that all outer products are learned at a similar pace—thereby achieving more balanced learning of facts in a linear associative memory, where each fact is represented by a single outer product** (Lines 248–264). We verify this hypothesis from both empirical and theoretical perspectives.
>
>
> We empirically verify this hypothesis with two experiments.
>
> * **Parameter level.** A key deduction of our hypothesis is that the associative-memory parameters learned by Muon should have a more isotropic singular-value spectrum than those learned by Adam. The results in Section 3.2 confirm this.
>
> * **End-to-end performance.** Muon should outperform Adam on the tail class in a knowledge-intensive, long-tailed QA task. This is supported by the results in Section 3.3. In addition, when we keep QK on Adam and apply Muon only to VO+FFN, the tail gains persist; the converse setting (Muon only on QK) does not.
>
> We also provide theoretical support for this hypothesis.
>
> * In a one-layer associative-memory model with class imbalance, we show that Muon’s update is near-isotropic and essentially balances learning across facts, whereas Adam/SignGD can be highly embedding-dependent and imbalanced. Equivalently, Muon’s update acts uniformly over stored outer products, which aligns precisely with the linear associative-memory structure.
>
> ___
> We thank the reviewer again for reading our response. We appreciate it if the reviewer could consider raising the score, if our rebuttal has addressed your concerns.

---

> ### Comment · Reviewer_sugx · 2025-11-22
>
> I can confirm my concerns W2, W4 & Q1, Q3 are resolved. I still want to discuss about W1&Q2, W5:
>
> W1&Q2: My question is how to connect heavy-tailness of data and knowledge with the heavy-tailness of parameter and update in the LLM. I'm still confused about how data tailness induces gradient tail. Your induction is **heavily based on**: "Under the assumption of linear associative memory model, data tailness can be directly converted to tailness over $E_k$ and $\tilde{E}_k$ based triplet." I did not see why this convertion can *possibly transfer* over LLM, where triplet convertion *does not exist at all*. I understand simple model is prevalent in theoretical study, but I want to understand how this convertion can transfer to complex model, at least informally.
>
> W5: utils.float_compute is still missing, and some files are not in the right directory, such as optimizers, models.
>
> Q3: resolved; still, confusion about theoretical supporting how data tailness induces gradient tail is addressed in W1&Q2.

---

> ### Author Response · Authors · 2025-11-23
> **Official Comment by Authors (Round 2)**
>
> We sincerely thank you for your reading and replying to our response, as well as raising insightful follow-up comments.
>
> ---
>
> ### **W1 & Q2: How to connect heavy-tailness of data and knowledge with the heavy-tailness of parameter and update in the LLM.**
>
> We would like to highlight that our theoretical intuitions of one layer model generalize to the general LLMs. We add this discussion in **Appendix K** of the updated revision. Specifically, we will show that the followings hold in general LLMs:
> * When the data is heavy-tailed, the gradient of parameters in general LLMs is also **heavy-tailed**.
> * The gradient of associative memory parameters in LLMs is the **sum of outer-products**.
>
> First, we introduce our model of a general LLM. We extend the simplified one-layer setup in Section 4 to a multi-layer architecture and focus on an arbitrary layer within it. In this layer, we study the matrix $W_{\text{out}}$ in the FFN module (the attention module can be analyzed analogously) and abstract the combined effect of the activation function and $W_{\text{in}}$ as a feature map $g: \mathbb{R}^d \rightarrow \mathbb{R}^d$, where $d$ is the hidden state dimension. To analyze the role of the remaining layers while keeping the notation concise, we abstract all preceding layers as a feature embedding and denote by $E_k \in \mathbb{R}^d$ the output of the previous layers for the $k$-th token. Similarly, we abstract all subsequent layers as a function $h: \mathbb{R}^d \rightarrow \mathbb{R}^d$. The function $h$ may also take all previous tokens as inputs, which we omit from the notation for brevity. Finally, we denote the language-model head parameters by $\tilde{E} \in \mathbb{R}^{d \times K}$, where $K$ is the vocabulary size. Based on these abstractions, for the $k$-th token we write the model output as:$$f_W(E_k)=\text{sm}\bigg(\tilde{E}^{\top}\Big[Wg(E_k)+h\big(Wg(E_k)\big)\Big]\bigg).$$
>
> The loss function on pretraining data is the cross-entropy loss weighted by token frequency $p_k$:$$\mathcal{L}(W) = -\sum_{k=1}^{K} p_k \log[f_W(E_k)]_k.$$ Consistent with heavy-tailed natural language distributions (e.g., WikiText103), for example, we can model the token frequency decay as $p_k = \alpha \cdot 2^{-k}$.
>
> Second, we derive the gradient of the loss with respect to $W$, denoted as $\nabla_W \mathcal{L}(W)$. The total gradient is a weighted sum of per-token gradients:$$\nabla_W \mathcal{L}(W) = -\sum_{k=1}^{K} p_k \nabla_W \log[f_W(E_k)]_k.$$ The total gradient $\nabla_W \mathcal{L}(W)$ remains **a linear combination of per-token gradients weighted by $p_k$**. Thus, it is heavy-tailed.
>
> The explicit form of the per-token gradient component is written as: $$\nabla_W \log [f_W(E_k)]\_k= \left(I_{d,d} + J_h(Wg(E_k))\right)^\top [\tilde{E} _k - \sum _{i=1}^{K} [f _W(E _k)] _i \cdot \tilde{E} _i ] g(E_k)^\top,$$
>
> where $I_{d,d}\in\mathbb{R}^{d\times d}$ is the identity matrix and $J_h$ is the Jacobi of the function $h$. The $J_h$ acts as a Jacobian preconditioner. It linearly transforms the gradient but **does not destroy the fundamental outer product structure** formed between the language model head features $\tilde{E}$ and the transformed input features $g$, i.e., $E$ of the one layer model in Section 4.
>
> Last, we briefly summarize why our one-layer model is representative of multi-layer models:
>
> * The outer-product structure originates from matrix–vector multiplications in the network, which appear in both one-layer and multi-layer architectures.
> * The heavy-tailedness of the gradient is inherited from the heavy-tailedness of the training data, which is exactly our data assumption.
>
> In our one-layer model, we prove that these two properties cause Muon to learn parameters in a more balanced way than Adam.
>
> ---
>
> W5: We update the full code that contains "utils.float_compute" and organize files in the correct folder.

---

> > ### Comment · Reviewer_sugx · 2025-11-26
> >
> > I've read Appendix K. Issues:
> > - Notation $k \in [K]$ should be category of alphabet. However, you write "$E_k$ is the output of the previous layers for the k-th token". In LLM, layers will only output $R^{d\times s}$ except for the last layer, where $s$ is the context lenght, there is no dimension $k$ in $E$.
> > - The loss notation
> > $$
> > \nabla_W \mathcal{L}(W)=-\sum_{k=1}^K p_k \nabla_W \log \left[f_W\left(E_k\right)\right]_k.
> > $$
> >
> >    I guess the loss here is for single token $s$? should be
> >    $$
> >    \nabla_W \mathcal{L}(W)=-\sum_{k=1}^K 1_{s=k} \nabla_W \log \left[f_W\left(E_k\right)\right]_k.
> >    $$
> > - Write "$\nabla_W$ is heavy-tailed since it is a weighted sum of per-token gradients with geometrically decaying weights $p_k = \alpha 2^{-k}$." This is not right, for example, if $\nabla_W\log [f( \cdot)] _k $ is the same for all $k $.
> >
> > I have a brief explanation though, by
> > $$
> > \mathbb{E} [\nabla_W \mathcal{L}(W)]=-\sum_{k=1}^K p_k \mathbb{E} [\nabla_W \mathcal{L}(W) | S = k].
> > $$
> >
> > $\mathbb{E} [\nabla_W \mathcal{L}(W) | S = k]$ as randomly drawn vectors, should be nearly perpendicular in high dimensions, by High-Dimensional Probability. geometrically decaying weights + nearly orthogonal can induce heavy tailness.
> >
> > This explanation is intuition, and resulting heavy-tailness is for vector not as heavy-tailness for matrix as Muon.
> >
> > Overall, I was satisfied with the author's previous response, but not this time. It's OK to write short, but keep meaningful.

---

> ### Author Response · Authors · 2025-11-27
> **Official Comment by Authors (Round 3)**
>
> Thanks for your insightful follow-up comments. We would like to clarify our points as follows.
>
> * (The meaning of $k$-th token) In our discussion, there are two notions of token indexing: (i) **its index in the alphabet (vocabulary)**, and (ii) its position in the sentence, i.e., the context mentioned by the reviewer. Throughout this section, **we only use the first notion**: the $k$-th token in the alphabet. Concretely, given a token-sequence prefix, we assume that the next token to be predicted is the $k$-th token in the alphabet. Let $H \in \mathbb{R}^{d \times s}$ denote the hidden features of the token sequence. We **only analyze the feature corresponding to the token to be predicted**, i.e., the last column of hidden state $H[:, -1] = E_k$ according to the transformer structure, while the influence of all preceding tokens on this prediction is absorbed into the function $h$.
>
>
> * (Discussions about our approximations) By writing $H[:,-1] = E_k$, we implicitly assume that the same token shares (approximately) the same feature embedding across all contexts. This assumption **holds exactly in the first layer of an LLM**, since $E_k$ there is just the token embedding, which is context-independent. It holds approximately in deeper layers, where $E_k$ can be viewed as the sum of the token embedding and the outputs of each layer.
>
> * (Organizing training loss with respect to the alphabet index) The training loss is the average over all tokens in the corpus. Rather than indexing this average by token positions within sentences (the sequence index discussed above), **we index it by the identity of the predicted token in the vocabulary**. Concretely, we write $\mathcal{L}(W) = -\sum_{k=1}^{K} p_k \log[f_W(E_k)]_k$, where $p_k$ is the empirical frequency of the $k$-th vocabulary token in the training corpus.
>
> * (Possibility of coincidentally same gradient) **It is almost impossible for different tokens to share exactly the same gradient**. Specifically, the gradient for the $k$-th token in the alphabet is
>     $$\nabla _W \log[f _W(E _k)] _k = \left(I _{d,d} + J _h(Wg(E _k))\right)^\top \left[ \tilde{E} _k - \sum _{i=1}^{K} [f _W(E _k)] _i \cdot \tilde{E} _i  \right]g(E _k)^\top.$$
>     Since different tokens have different embeddings $\tilde{E}_k$ in the language-model head and different feature embeddings $E_k$, it is highly unlikely that they share the same gradient.
>
> * (Alignment between gradient and Muon) We now explain in more detail the “nearly perpendicular in high dimensions” comment by the reviewer and its connection to Muon. Note that $W$ is a matrix, and the high-dimensional orthogonality concerns the embedding vectors: for different $k$, the vectors $\tilde{E}_k$ and $g(E_k)$ are (approximately) orthogonal. **This vector-level orthogonality makes the outer-product terms $\left[ \tilde{E} _k - \sum _{i=1}^{K} [f _W(E _k)] _i \cdot \tilde{E} _i  \right]g(E _k)^\top$ for different tokens $k$ approximately orthogonal under the Frobenius inner product.** Comparing this with the singular-value decomposition $\nabla _W \mathcal{L}(W) = \sum _{s=1}^{d} \sigma _{s} u _{s} v _{s}^\top$, where $\\{u_s\\} _{s=1}^{d}$ and $\\{v _s\\} _{s=1}^{d}$ form orthonormal bases, **we see that Muon will tend to balance the contributions of different tokens in the alphabet to the gradient, mitigating the effect of their differing frequencies $p_k$**, which are implicitly encoded in the singular values $\\{ \sigma _s \\} _{s=1}^{d} $.

---

> > ### Comment · Reviewer_sugx · 2025-11-27
> >
> > You first say "assume that the next token to be predicted is the $k$-th token in the alphabet.", later say "By writing $H[:,-1] = E_k$, we implicitly assume that the same token shares (approximately) the same feature embedding across all contexts. This assumption holds exactly in the first layer of an LLM, since $E_k$ there is just the token embedding, which is context-independent. ".
> >
> > $k$ is the alphabet to be predicted, right? However, casual transformer input only gets the token embedding before the predicted token. How come you get $E_k$? you only get type $k$ from $H[:,-1]$ in the final loss. As modded-nanogpt writes
> > ```
> > _inputs = buf[:-1]
> > _targets = buf[1:]
> > ```
> > I really do not get what you are saying.

---

> ### Author Response · Authors · 2025-11-27
> **Official Comment by Authors (Round 4)**
>
> Thank you for your follow-up questions and for continuing the discussion with us. We agree with your point regarding the causal nature of Transformers: the input must be the prefix context, and the target token must not be leaked into the input. We would like to clarify our definitions precisely and provide a concrete example.
>
> * (A detailed definition) **We define $E_k$ as the hidden state of the last token in the training context that precedes the $k$-th token in the alphabet, at the layer where associative memory is present.** In our previous response, we described $E_k$ as "the feature corresponding to the token to be predicted"; here, "corresponding" specifically refers to the feature of the context prior to the $k$-th token in the alphabet. We also highlight that $\tilde{E}_k$ is the feature at the language-model head of the $k$-th token in the alphabet itself.
>
> * (An example) To illustrate our notation, consider a training sample where the context is “A successful entrepreneur Elon” and the target to be predicted is “Musk” ( the $k$-th token in the alphabet is $\text{Musk}$). Here, **$E_{k}$ denotes the feature representation of “A successful entrepreneur Elon”**. In particular, it is the feature at the position of “Elon”, which also aggregates information from the preceding tokens “A successful entrepreneur” via attention in the previous layers. Feature $E_k$ does **not** refer to the embedding of the token “Musk”. However, $\tilde{E}_k$ is the language-model head feature of "Musk".
>
> * ("we implicitly assume that the same token shares (approximately) the same feature embedding across all contexts.")  Although a token can appear in many different contexts, we model all such contexts with the same feature vector. For example, we use $E_k$ to represent the feature corresponding to both “A successful entrepreneur Elon” and “A distinguished entrepreneur Elon”. This is justified for two reasons: (i) these contexts typically share many common words, such as “entrepreneur” and “Elon”; and (ii) **our results would remain unchanged even if we introduced a separate notation** $E_k^{\prime}$ for “A distinguished entrepreneur Elon.” For notational simplicity, we therefore use the same feature vector to represent all such contexts.
>
>
>
> * (Why this is plausible) **Adopting a single feature vector $E_k$ to represent a context, rather than a single token, is not only intuitive but also empirically grounded.** Existing works on knowledge editing explicitly construct such context-level representations in LLMs (Meng et al., 2022; Fang et al., 2024). For example, they can identify a vector corresponding to the concept “US president” and modify it so that the model’s output changes from “Biden” to “Trump”.
>
>
> * (Why we adopt this notation in our rebuttal) We admit that some notations let $E_k$ represent the token embedding of the $k$-th token in the alphabet itself, e.g., "Musk". However, **our work focuses on the token associations**, e.g., the relationship between "A successful entrepreneur Elon" and "Musk", which is stored in the associative memory of LLMs. Thus, for notational convenience, we let $E_k$ denote features that can induce the token “Musk,” e.g., “A successful entrepreneur Elon,” and let $\tilde{E}_k$ denote the language-model head features of the token “Musk” itself.
>
>
>
> Thank you again for your time, and please let us know if our response does not fully address your concerns.
>
> ---
> **Reference**
>
> Meng K, Bau D, Andonian A, et al. Locating and editing factual associations in gpt. Advances in neural information processing systems, 2022, 35: 17359-17372.
>
> Fang J, Jiang H, Wang K, et al. Alphaedit: Null-space constrained knowledge editing for language models. arXiv preprint arXiv:2410.02355, 2024.

---

> ### Comment · Reviewer_sugx · 2025-11-28
>
> I think I may finally understand your notation, I hope so
> - Should as write $k$-th type token, I interpret it as $k$-th position token at first, which's confusing.
> -
>    > By writing $H[:,-1] = E_k$, we implicitly assume that the same token shares (approximately) the same feature embedding across all contexts. This assumption holds exactly in the first layer of an LLM, since $E_k$ there is just the token embedding, which is context-independent.
>
>    In the 3rd respone, $E_k$ should be the token embedding of $k$-th type token, otherwise $E_k$ would not be the same in the first layer. In the 4th response, $E_k$ is the hidden state of the last token in the training context that precedes the $k$-th type token. Right? $H \in \mathbb{R}^{d \times s}$ actually denotes the hidden features of the token sequence before the predicted token, which should not include predicted token alphabet $k$.
> - $h$ is a function? It should be depending on $H[:,:-1]$ and $k$? $h$ can be similar under the same $k$ with the similar assumption of $E_k$ and $E_k'$.
>
> I really need to reconfirm your notation before reading the proof, since it's chaotic at the beginning. \
> Am I right about your final notation? No need to hurry. No need to write long. Please save my time with the clear notation.

---

> > ### Author Response · Authors · 2025-12-01
> >
> > Thanks for the continual discussion with us. We think that your understandings are largely correct and we want to further clarify some details.
> >
> > * ($k$-th type token) Yes, your understanding is correct. We mean the **$k$-th token in the alphabet**, i.e., the $k$-th type token.
> >
> > * (Relationship between hidden states and $E_k$) You are correct that $H \in \mathbb{R}^{d \times s}$ should not include the predicted token, and that $E_k$ is the hidden state of the last token in the training context that precedes the $k$-th type token. In both our 3rd and 4th responses, we do **not** define $E_k$ as the token embedding of the $k$-th type token. In contrast, when the **first layer** serves as the associative memory, $E_k$ **is the token embedding of the last token of the context**, i.e., exactly the hidden state of the last token in the training context.
> >
> >
> > * (An example for the relationship between hidden states and $E_k$) Consider the example where the context is “A successful entrepreneur Elon”, and the token to be predicted is “Musk”. If each word is a token, then $H\in\mathbb{R}^{d\times 4}$ is the hidden states of words "A successful entrepreneur Elon", and $E_k$ is the hidden state of "Elon". When the first layer serves as the associative memory, $E_k$ is the token embedding of "Elon", i.e., the input hidden state of the first layer.
> >
> >
> > * (Function $h$ explanation) **The function $h:\mathbb{R}^{d}\rightarrow \mathbb{R}^{d}$ depends on $H[:,:-1]$, but does not directly depend on $k$, i.e., it depends on $k$ through the context $H[:,:-1]$.** We adopt $h$ to summarize the combined effect of attention and FFN in later layers. In particular, within the attention submodule (which is part of $h$), the transformation applied to $E_k$ depends on the full context $H[:,:-1]$. For notational simplicity, we suppress the explicit dependence of $h$ on $H[:,:-1]$, **as this does not affect our conclusions**: the gradient still has an outer-product structure, and the weights in this structure are related to the heavy-tailedness of the data.
> >
> >
> >
> >
> > Thank you again for your time and for your thoughtful feedback. We hope that our response has addressed all of your concerns.

---

### Official Review · Reviewer_Yhie · 2025-10-30

**Soundness:** 2
**Presentation:** 2
**Contribution:** 2
**Rating:** 4
**Confidence:** 3

**Summary:**

The authors mainly did an abalation study of Muon. They showed two major and novel findings: 1. applying muon only on VO+FFN can recover the full muon training. 2. Muon narrows the head-tail performance gap when learning highly imbalance datasets. They also provided a preliminary theory to explain their findings.

**Strengths:**

1. The ablation study of this work is sound to me. Muon only works for VO and FFN. This finding is somewhat interesting and might inspire following works to promote Muon's ability over QK.
2. The heavy-tail findings is also interesting, which promotes the appliablity of Muon in practice.

**Weaknesses:**

**The novelty and contributions of this paper could be further improved.**
- Three observations are provided. Among these, the second observation is already presented in Liu et al. (2025). Though the differences between them are noted by the authors, such as Liu et al. (2025) conduced experiments on MoE models whereas this paper focues on Dense Models, these reasons cannot convince me about the novelty.
- The first observations are good, yet I expect more analysis. A central question is "Why Muon is more effective on VO+FNN than QK", yet this paper didn't address this question. They attribute this phenomenon to the associative memory property of VO and FFN, but the causal relationship between these is not clear. Is the heavy-tail task ability of Muon relate to the effectiveness of Muon on VO+FNN?
- In addition, via the third observation, the authors further highlighted the connection between Muon and the associative memory. However, I also doubt that if the third observation can fully supports that Muon acquires knowledge more evenly than AdamW. Muon indeed prompts the performance on heavy-tail tasks, but it might restrict to knowledge acquring tasks. I expect more experiments across various tasks.

**The toy theory seems to be intuitive**
- The theory seems to be intuitive. Muon produces updates evenly spread over all directions compared to GD and SignGD. Then Muon must give even/uniform weight matrices, and thereby in this simple associative memory modeling, mitigating the unbalance learning problem.

**Questions:**

N/A

---

> ### Author Response · Authors · 2025-11-20
>
> We thank the reviewer for your insightful suggestions. Below we respond to the comments in Weakness (W).
>
> ---
>
> ### **W1. The second observation is already presented in Liu et al. (2025).**
>
> We agree that Liu et al. (2025) also reported that Muon can yield more isotropic weight spectra. However, **we respectfully clarify that our contribution is not only the observation of isotropy itself, but rather a deeper, mechanistic explanation of where this isotropy matters and why it emerges—a gap that Liu et al. (2025) does not address**.
>
> From the view of **novel empirical findings**, we have these two findings not cover by Liu et al. (2025).
>
> * **Stability Focus:** We are the first to investigate and demonstrate Muon's robust stability across random initializations (e.g., the tight error bars in Figure 2), in sharp contrast to Adam's high instability on this task—a finding we also connect to our theoretical results.
>
> * **Precise Localization and Functional Disentanglement:** While Liu et al. (2025) report spectral entropy for component groups such as AttnQO and AttnKV, their categorization combined functionally distinct parameters. Specifically, they group Query with Output, and Key with Value. This grouping obscures the critical mechanistic distinction between QK and associative memory (VO).
>
>     In contrast, our work disentangles these components. We specifically isolate VO from QK. Our component-wise ablations in Figure 1 show that the performance benefits stem almost entirely from the VO and FFN components. Applying Muon to QK contributes little. This functional localization is a key distinction not made in prior work.
>
> From the view of **theoretical understanding**, this serves as a firm basis.
>
> * Our paper is the first to provide a mechanistic link from the heavy-tailed nature of knowledge (Observation 3) to this isotropic spectra (Observation 2). We show that isotropy is not just a statistical artifact but a direct consequence of Muon's update rule, which is better aligned with the outer-product structure of associative memories. Our theoretical analysis (Theorems 4.3) proves this mechanism enables Muon to learn tail classes more effectively and uniformly than Adam, which explains why Muon's performance is superior.
>
> ---

---

> ### Author Response · Authors · 2025-11-21
>
> ### **W2. A central question is "Why Muon is more effective on VO+FNN than QK", yet this paper didn't address this question.**
>
>
>
> >They attribute this phenomenon to the associative memory property of VO and FFN, but the causal relationship between these is not clear.
>
>
> In fact, our core contribution is precisely to explain "why Muon is more effective on VO+FFN than on QK". From both empirical and theoretical perspectives, our work supports the following causal relationship between Muon’s superiority and associative memories: **because Muon normalizes the amplitude of each outer product in the SVD of the gradient—so that all outer products are learned at a similar pace—it achieves more balanced learning of facts in a linear associative memory, where each fact is represented by a single outer product** (refer to the claim and added example in line 237-263 of the revised version).
>
> Before going into details, we highlight a key conclusion from the knowledge-editing literature (Bietti et al., 2023; Meng et al., 2022): **VO and FFN parameters can be modeled as linear associative memories that store the prior knowledge/facts of LLMs**. This point is introduced in Lines 129-135 of the original version and Lines 131-139 of the revised version.
>
> **Empirically**, we verify this relationship both from the internal behavior of LLMs (Section 3.2) and from their end-to-end performance (Section 3.3).
>
> * Concretely, Section 3.2 shows that Muon indeed achieves more balanced learning of facts at the parameter level: the singluar values of VO and FFN parameters learned by Muon are more isotropic than those learned by Adam. This indicates that the facts (i.e., outer products) have more comparable amplitudes in the weights learned by Muon than by Adam.
> * Section 3.3 then confirms this in an end-to-end manner. **Since knowledge is stored in VO and FFN parameters, i.e., in associative memories, the most direct way to test whether Muon achieves more balanced learning on associative memories is to examine whether it learns knowledge in a more balanced way across classes.** This is exactly the message of Figure 3: under a heavy-tailed distribution, Muon yields a much smaller accuracy gap between head and tail classes than Adam.
>
> This also affirmatively answers the reviewer’s question:
>
> > Is the heavy-tail task ability of Muon relate to the effectiveness of Muon on VO+FNN?
>
> We highlight two key points:
> (1) Muon’s performance on heavy-tail tasks that rely on associative memory depends on its effectiveness when optimizing the VO+FFN parameters;
> (2) in contrast, Muon may not outperform Adam on heavy-tail tasks that do *not* rely on associative memory.
>
> Our work verifies these two points separately.
>
> - **To verify (1)**, we show that the benefit of Muon over Adam on heavy-tailed tasks that depend on associative memory is contingent on Muon actually optimizing VO+FFN. In the knowledge acquisition task in Section 3.3, which requires the LLM to store knowledge in its associative memories, when Muon only optimizes the QK parameters, its performance is similar to Adam. However, when Muon optimizes VO+FFN, it recovers the performance of Muon that optimizes all parameters.
>
> - **To verify (2)**, in Appendix G.8 of the revision, we train LLMs with Muon and Adam on an in-context linear regression task that primarily depends on learning the QK parameters. The similar performance of Muon and Adam on this task shows that they behave similarly when optimizing QK.
>
> ---

---

> ### Author Response · Authors · 2025-11-21
>
> ### **W3. Muon indeed prompts the performance on heavy-tail tasks, but it might restrict to knowledge acquring tasks. I expect more experiments across various tasks.**
>
> Thanks for your comment. To address this, we have added a new experiment on the **WikiText103** language modeling benchmark in Appendix G.9 (Figures 15 & 16). We treat this standard language modeling task as a natural heavy-tailed learning problem by partitioning the tokens into frequency-based groups.
>
> Our findings on this real-world dataset perfectly mirror our synthetic results: As shown in Figure 15, Muon and Adam perform similarly on head groups. However, as token frequency decreases, the performance gap steadily widens, with **Muon substantially outperforming Adam on the tail groups**.
>
> Consistent with our theory, Adam shows growing instability (large error bars) on the tail groups, whereas Muon remains stable across the entire spectrum. On WikiText103, the **Muon(VO+FFN) hybrid configuration** nearly overlaps with the full Muon curve, confirming that the associative memory components are the primary drivers of this tail-learning capability.
>
> ---
>
> ### **W4. The toy theory seems to be intuitive**
>
> We thank the reviewer for this comment. While we appreciate the reviewer’s concern, we do not regard the fact that the toy model yields an intuitive picture as a weakness. On the contrary, we view intuitiveness as a desirable property of a theory: a good model should both match empirical behavior and distill it into a simple, interpretable mechanism.  In addition, the core of our theory is not only to confirm this intuition for Muon, but also to **provide the first quantitative, constructive proof of why and how Adam fails** in the associative-memory setting.
>
> These are non-trivial results that prior Muon and Adam theory does not capture:
>
> * **Novelty vs. Prior Adam Theory**. Existing theoretical work on Adam has primarily analyzed its convergence behavior in *online optimization* or *general non-convex* settings. These studies focus on bounding regret or establishing convergence guarantees under smoothness and Lipschitz assumptions, gradually relaxing these conditions while tightening rates. While such analyses clarify when Adam converges, they do not explain *what structure* of the learned solution emerges or *how Adam’s update geometry interacts with the learning dynamics* of associative memories.
>
>     We are the first to provide a **constructive, quantitative bad-case analysis** for Adam specifically in the **LLM associative-memory** regime. Proving why Adam fails (spectral anisotropy) and how much it fails is the key, non-intuitive theoretical contribution of this paper.
>
> * **Novelty vs. Prior Muon Theory**. Existing theoretical work on Muon mostly treats it as a generic optimizer and studies convergence in abstract optimization settings. Prior analyses focus on (i) convergence rates in convex problems or (ii) convergence to stationary points in general non-convex objectives.
>
>     Our work is the first to analyze Muon in a non-convex logistic-regression–style associative-memory model tailored to LLM knowledge recall. We prove that Muon converges to the optimal solution in this setting and that the resulting update is nearly isotropic across singular directions, which in turn yields balanced learning of head and tail facts.
>
> ---
> We thank the reviewer again for reading our response. We appreciate it if the reviewer could consider raising the score, if our rebuttal has addressed your concerns.
> ___
>
> **Reference**
>
> Meng K, Bau D, Andonian A, et al. Locating and editing factual associations in gpt[J]. Advances in neural information processing systems, 2022, 35: 17359-17372.
>
> Bietti A, Cabannes V, Bouchacourt D, et al. Birth of a transformer: A memory viewpoint[J]. Advances in Neural Information Processing Systems, 2023, 36: 1560-1588.

---

> ### Author Response · Authors · 2025-11-27
>
> Dear Reviewer,
>
> As the discussion period is approaching its end, **with less than a week remaining**, we would like to ensure that we have fully addressed all of your concerns. If you have any additional comments or suggestions, please feel free to share them. We are happy to clarify any remaining points and further improve our work.
>
> Thank you so much for your time and effort in reviewing our work.

---

### Official Review · Reviewer_W4iz · 2025-10-30

**Soundness:** 3
**Presentation:** 3
**Contribution:** 3
**Rating:** 6
**Confidence:** 4

**Summary:**

The paper studies what transformer components benefit most from Muon as compared to Adam. The authors find that the main benefit comes from the VO and FFN blocks, which correspond to the associative memory stores. The authors use a heavy-tailed knowledge task to compare Adam and Muon and provide theoretical insights on one-layer models.

**Strengths:**

- The imbalanced knowledge experiment is a nice set up and the ablations over which layers most benefit from Muon are interesting
- Provides an interesting connection between optimizer performance and prior work on associative memory
- The writing and research motivation are clear

**Weaknesses:**

- Experiments are limited to a synthetic heavy-tailed knowledge task. It would be interesting to see in addition a task that relies heavily on QK to support the claim that Muon is not as important to apply to QK.
- Language modeling results are limited to 160M parameter transformers. It would be interesting to see the main ablation (Muon only on VO+FFN) at a larger scale and performance measured on downstream tasks.

**Questions:**

- Do you have any intuition on why benefits from Muon seem to diminish with scale (/if this pertains to associative memory)?
- It seems like the Newton-Schulz application could be an interesting knob to turn here -- have you tried varying the Newton Schulz coefficients/iterations to vary how much of the spectrum Muon is capturing and test how this affects associative memory?
- What LR schedule did you use for Adam and Muon respectively?

---

> ### Author Response · Authors · 2025-11-20
>
> We thank the reviewer for your positive review and suggestions, as well as many insightful suggestions. Below we respond to the comments in Weakness (W) and Questions (Q).
>
> ---
>
> ### **W1. It would be interesting to see in addition a task that relies heavily on QK to support the claim that Muon is not as important to apply to QK.**
>
> Thank you for this insightful suggestion. We agree that an additional experiment on a QK-heavy task would strengthen our claim. In Appendix G.10 of the revised version, we train LLMs with Muon and Adam on an in-context linear regression task. The LLM is asked to infer a vector $w$ for a query $x_{\text{query}}$ from demonstrations $\{(x_i, y_i)\}_{i=1}^{K}$ in the prompt, where $y_i = w^\top x_i$. The value of $w$ varies across prompts, and we assign different frequencies to $w$ in different subspaces. Intuitively, the **LLM must rely on QK parameters to extract $w$ from the demonstrations**.
>
> However, as shown in Figure 17, **Adam and Muon achieve very similar performance on both head and tail classes**. This is in sharp contrast to our main findings (e.g., in Section 3.3), where Muon substantially outperforms Adam on tail classes. This experiment further supports our claim that the primary benefits of Muon do not stem from its application to the QK parameters, which is why we found it effective to exclude them.
>
> ---
>
> ### **W2.  It would be interesting to see the main ablation (Muon only on VO+FFN) at a larger scale and performance measured on downstream tasks.**
>
> Our work also includes experiments on **0.7B** models, reported in Appendix G.2 of the original version (Appendix G.4 of the revised version). These experiments display the **same overall pattern** as those for the 160M model:
>
> - **Module-level superiority:** **VO + FFN Muon** nearly matches **Full Muon**, whereas **QK-only** offers minimal improvement (Fig. 8).
> - **Parameter isotropy:** The 0.7B spectral metrics show **stronger isotropy** in VO/FFN parameters under Muon compared to Adam (Figs. 7).
>
> ---
>
> ### **Q1. Do you have any intuition on why benefits from Muon seem to diminish with scale (/if this pertains to associative memory)?**
>
> Thanks for this interesting question. Our work focuses on understanding **how the algorithmic design of different optimizers affects their performance**, and such design principles should, in principle, **apply across model scales**. In addition to algorithmic design, the practical **performance of optimizers at large scale also depends crucially on the hyperparameter tuning protocol** (see, e.g., Wen et al., 2025). They report that the benefits of Muon diminish with scale, from 1.4× at 0.1B models to 1.1× at 1.2B models, whereas Liu et al. (2025) find that Muon remains about 2× more FLOP-efficient than Adam on 32B models. This discrepancy may stem from differences in their hyperparameter tuning protocols. A systematic study of how hyperparameter tuning interacts with optimizer performance is important but **orthogonal to the focus of our current work**.
>
> ---
>
> ### **Q2. It seems like the Newton-Schulz application could be an interesting knob to turn here -- have you tried varying the Newton Schulz coefficients/iterations to vary how much of the spectrum Muon is capturing and test how this affects associative memory?**
>
> Thanks for this important question. **Our experiments include the coefficients of standard implementation of NS in Muon and the exact SVD**, and both results support our conclusion of the benefits of Muon on learning associative memories. Concretely, in our experiments of 160M and 0.7B model, we follow standard Muon practice and fix a **small, constant number of NS iterations (=5)** to approximate the orthogonal factor $O_t \approx B_t(B_t^\top B_t)^{-1/2}$, described in Line 97-98 of original version (Line 97-98 of revised version). In our experiment of the one layer model in Section 4, we implement the exact SVD for Muon implementation. Both results demontrate the superiorty of Muon on learning associative memories.
>
> The influence of coefficients of Newton-Schulz application is an interesting problem and is orthogonal to our contribution. The work "*Convergence of Muon with Newton-Schulz*" derives a convergence bound of Muon, including the influence of NS steps. Furthermore, "*Accelerating Newton-Schulz Iteration for Orthogonalization via Chebyshev-type Polynomials*" proposes accelerating these NS steps via Chebyshev-optimized coefficients. We added this in the related work section.
>
> ---
> ### **Q3. What LR schedule did you use for Adam and Muon respectively?**
>
> We apply the Warmup-Stable-Decay (WSD) schedule for both Adam and Muon. This schedule involves a linear warmup phase, followed by a constant (stable) learning rate phase, and concludes with a rapid decay phase. This setting is widely adopted in the LLM optimizer.

---

> ### Author Response · Authors · 2025-11-20
>
> We thank the reviewer again for reading our response. We appreciate it if the reviewer could consider raising the score, if our rebuttal has addressed your concerns.
>
> ---
> **Reference**
>
> Wen K, Hall D, Ma T, et al. Fantastic pretraining optimizers and where to find them. arXiv:2509.02046.
>
> Liu J, Su J, Yao X, et al. Muon is scalable for LLM training. arXiv:2502.16982.

---

> ### Author Response · Authors · 2025-11-27
>
> Dear Reviewer,
>
> As the discussion period is approaching its end, **with less than a week remaining**, we would like to ensure that we have fully addressed all of your concerns. If you have any additional comments or suggestions, please feel free to share them. We are happy to clarify any remaining points and further improve our work.
>
> Thank you so much for your time and effort in reviewing our work.

---

> > ### Comment · Reviewer_W4iz · 2025-11-27
> >
> > Thank you for the detailed rebuttal. I will maintain my current score. I think the ablation study is useful and clearly executed, and while the associative-memory interpretation feels somewhat narrow, it is reasonable within the scope of the work. I'd lean towards acceptance.
> >
> > Two minor notes regarding my prior comments:
> > - I don't see the question of scale being completely orthogonal to your work, but I understand if increasing the scale is beyond the scope of your current submission. I agree that the performance of optimizers depends on hyperparameter tuning, but I think it would be interesting to see whether the benefits for associative-memory benefits of standard Muon diminish with scale and how this fits into your interpretation.
> > - My point about Newton-Schulz was not a critique of your implementation (which is standard), I meant more so as a possible experimental knob to turn, where the parameters of the NS application can control how well the update gets orthogonalized and how strongly minor singular directions are suppressed. But this is not concerning any issue I have with your current paper.

---

> > > ### Author Response · Authors · 2025-11-27
> > >
> > > We are glad that our additional experiments have addressed your major concerns, and we thank you for your positive evaluation of our work.
> > >
> > > We would like to provide further clarification on your two minor comments.
> > >
> > > * We believe that whether the benefit of Muon diminishes with scale is **still an open question**. For example, Wen et al. (2025) report that Muon’s benefit shrinks to 1.1× on a 1.2B-parameter model, whereas Liu et al. (2025) find that Muon remains about 2× more FLOP-efficient than Adam even on 32B models. Thus, it is unclear whether the observed diminishing returns are an intrinsic property of Muon or a consequence of hyperparameter tuning. **A definitive answer to this question is beyond the scope of our work.**
> > >
> > > * Thank you for confirming that our experimental setting is plausible. We agree that a better implementation of NS could yield a better normalization, and we would like to explore this direction in future work.

---

### Official Review · Reviewer_E1d8 · 2025-11-05

**Soundness:** 2
**Presentation:** 2
**Contribution:** 3
**Rating:** 4
**Confidence:** 4

**Summary:**

The submission investigates the reason for the improved performance of Muon over Adam on transformers. It establishes empirically that the weights of a transformer architecture that benefit most from Muon-style updates are the non-key/query parameters of the attention modules and the parameters of the feedforward networks. To explain this behaviour, the authors propose an explanation based on different frequencies of "fact" representations in an associative memory module, and show theoretically that Muon-style updates lead to more uniform convergence across different frequencies.

**Strengths:**

To my knowledge, this is the submission that attempts to provide empirical evidence for the source of the benefit of Muon updates over Adam on language models. The identification of specific weights that benefit most from Muon and the corresponding explanations help us understand the difficulties of optimization in transformer-based architecture, and hopefully design better methods. The experiment on the synthetic dataset with imbalanced facts is well-designed an provides good evidence for the claim that Muon helps most on associative memories.

I am generally positive about the submission. My initial score for the review is low due to the issues listed below. Those should be addressable with some revisions to the presentation and an additional experiment. I am expecting to increase my score if that can be added.

**Weaknesses:**

My main issue is with the presentation of the results, specifically in Figure 1, 2, 4 and §4.1.Those are dense and require a lot of care on the reader's part to understand what is being shown. This complexity does not appear necessary. A pass on the issues are detailed below could improve the clarity of the presentation significantly.

On the evidence, the main argument of the submission, that Muon helps with imbalance in "facts" stored in an associative memory. It appears difficult to distinguish between this explanation and the more general imbalance in token frequencies in text data for the success of Adam. See below of a possible additional experiment that would strengthen the argument of the submission.

The submission would be much stronger with those changes, and I would be happy to increase my score if they are addressed.

## Distinguishing between frequency imbalance in "facts" vs. general imbalance in text

The argument presented by the submission is that the benefit of Muon over Adam stems from an imbalance in the frequencies of different "facts" stored in an associative memory.
The controlled experiment with a synthetic dataset in §4.1 is a very nice illustration of this point. However, we could also expect this imbalance to stem simply from the distribution of tokens in text data, which follow Zipf's law. Those have been shown to lead to a better performance of Adam over SGD (Kunstner et al., 2024) primarily due to the update in the first and last embedding layers (Zhao et al., 2024; Zhang et al., 2024). One hypothesis could be that Muon helps on additional layers for the same issue of imbalanced tokens, not specifically facts in associate memories. Adam cannot correct these imbalances beyond the embedding layers because they are no longer axis-aligned, while Muon could.

The main difference I am drawing between those two hypotheses is the whether the main difficulty is due to an imbalance in token frequencies or an imbalance in the frequencies of "facts" to be stored in associate memories. As facts are also represented as tokens, there is some overlap between those ideas, but there is a way to disentangle these two explanations. If the arguments of the submission is correct, that Muon target imbalances in "facts" and associate memories beyond mere imbalance in token frequencies, I would expect the performance gap between Muon and Adam to decrease as the fact imbalance is reduced. The synthetic dataset prepared by the authors could be modified to exhibit various levels of imbalance, varying the imbalance continuously between the approximate power-law distribution in Figure 3 to a uniform distribution. The imbalance in token frequencies should still follow a power-law (assuming the different templates are rich enough to approximate natural text, which should be checked), but the imbalance in "facts" would be controlled. If the performance gap between Muon and Adam shrinks as the "fact" imbalance becomes more uniform, this would give a lot more strength to the proposed fact-based explanation.



## Presentation of the results

The specific recommendations below are only one possible way to improve the presentation, meant to help communicate the difficulties I encountered in understanding the results. The authors know their work best and might have other ideas on how to improve the presentation that are more appropriate.


**Figure 1**:
There are too many lines to be able to easily tell which combination of architecture/optimizer is performing best.
This detailed plot could be useful to readers wanting to see the behaviour over time,
but the main point of Figure 1 would be better illustrated by a simple bar plot of the final performance,
using more legible distinctions (colors/makers) or having the description of the setup directly above the bar.

The figure also combines gated and non-gated variants of the architecture,
but it is not clear why this distinction is important for the point being made.
The plot could be made much simpler by focusing on one of the two variants,
and pointing the reader to "the same happens on the gated version, see Appendix X".

**Figure 2**
suffers from a similar issue.
The plot shows 4 metrics measuring the spread of the eigenvalues,
some which should be small while other should be large,
for 2 weights (VO, $W_{\mathrm{out}}$) on 2 architectures (gated, non-gated).
This makes it hard to extract the main point of the figure at a glance.
A simpler variant of the figure could focus on one architecture and one metric,
relegating the others to the appendix with the comment that a similar behaviour is observed,
or use this space to show the full distribution of the spectrum over time.

An interesting addition to Figure 2 could be to show the QK weights. The Muon update is apparently less helpful for those weights; does the spectrum give a hint as to why?

**Figure 4 (b, c)**:
I had a hard time understanding those figures, and what those metrics tell us about the update.
I am not sure which is the best way to improve the presentation, but here are things I found confusing:

The figure 4b is implicitly parameterized by a step-size which is being varied. This can only be understood from the main text (not the caption), which should be fixed. Even with this information, it is not easy to see the "step-size" on the plot. This might be because the lines "start" at the bottom right ($\eta = 0$ is at $(10^{.8}, 0)$) and go to the top left as $\eta$ increases, going against the standard left-to-right/top-to-bottom reading direction. Flipping the x and y axes might help. Another option could be to use a plot with 2 y-axes (or two subplots on top of each other) that would show the $\Delta$ vs. step-size and the corresponding loss vs. step-size. The same treatment could be applied to Figure 4c (with number of updates as the x-axis). Currently, 4b and 4c have a different experimental setting (varying step-size vs. varying time) but it is not clear from the figures/caption that they differ in this way.

**Theoretical results in §4.2**
Assumption 4.2 seems overly complex for its uses in Theorem 4.3. The parameters $\alpha$ and $\beta$ appear to impact the results only through the ratio $r(\alpha,\beta)$. Since it happens to be $\frac{\min_i p_i}{\max_i p_i}$, wouldn't it be easier to just define the degree of imbalance as $r = \min_i p_i / \max_i p_i$ directly, and avoid the extra notation of $\alpha, \beta$? If not, writing the overall loss after Assumption 4.2 as a function of $\alpha, \beta$ might be helpful.

The only place where I see their use is in the proof for Adam, but the range of behavior observed under the $(\alpha,\beta)$ assumption would also be observed on a model with imbalance ratio $r$ since it's strictly less restrictive. If I'm missing something and the $\alpha, \beta$ are important for the proof, they should still be mentioned, but an explanation of what $r(\alpha,\beta)$ is should be given around Assumption 4.2 to help the reader.

Could some of the notation be simplified by assuming that the frequencies are sorted? For example $\min_{k\in[K]}$ and $\max_{k\in[K]}$ in Eq. 4.1 could be replaced by $p_K$ and $p_1$ if we assume the probabilities are decreasing.

The description of the stopping condition would also benefit from a slightly more wordy explanation. The start of §4.2 takes a while to parse. Something like "For each optimizer, we take one step with step-sizes $\eta$, selected such that the accuracy of the most frequent class reaches $1-\epsilon$. We then look at the accuracy achieved on the least frequent class."


## Additional references

The submission should cite and discuss the following works which have done similar ablation studies on Adam to justify its performance on language models and identify that they help most with the first and last embedding layers. This does not detract from the contributions of the submission, but those works should be acknowledge.$

- Deconstructing What Makes a Good Optimizer for Language Models
  Rosie Zhao, Depen Morwani, David Brandfonbrener, Nikhil Vyas, Sham Kakade
  https://arxiv.org/abs/2407.07972
- Adam-mini: Use Fewer Learning Rates To Gain More
  Yushun Zhang, Congliang Chen, Ziniu Li, Tian Ding, Chenwei Wu, Diederik P. Kingma, Yinyu Ye, Zhi-Quan Luo, Ruoyu Sun
  https://arxiv.org/abs/2406.16793


## Minor/Typos

- "intrinsically heavy-tailed: a few classes (tail classes) appear far less frequently than others"
  Is this a typo? Presumably if only a few classes are rare, their impact on the overall loss is small and Muon would not help much.
- It is not clear where Theorem 4.3 ends. Could the paragraph breaks "For GD, ... For Muon, ... For Adam, ..." be placed in an itemize block to help that?
- "highly sensitive" as a description of Adam's behavior in Figure 2 seems exaggerated. "More sensitive" would be appropriate.

**Questions:**

**Why does Muon help more on the VO/FFN than on the QK weights?**
This is the main distinction found in the first set of experiments.
The theory tries to explain why the VO and FFN weights are helped by Muon by drawing on the connection to associative memories,
but I have some reservations about this explanation.

The QK weights are also part of the attention mechanism and form an outer-product based on the representation of the tokens. Why would the associative-memory explanation not apply to them as well? The FFN weights do not seem to follow this outer-product structure, so why is the associate-memory explanation relevant for them?

It might be that the associative-memory explanation is most relevant for the VO weights and do not explain the benefits on FFN weights or the reduced benefits on QK weights.
But if that is the case, this should be acknowledged more explicitly in the submission.

Beyond this theoretical explanations, it would be helpful for future work if the authors could look at the experiments and data they have collected to see if there is some empirical difference between the QK weights and the VO/FFN weights that could explain the different behavior, for example a different spread in the spectrum of the singular values of the inputs of each layer? This is not strictly necessary but even a speculative explanation (based on some empirical observation) might be helpful for future work,

**Why the focus on Gated vs. Ungated FFN?**
Is there a-priori a reason to believe that Muon would behave differently?

**How is the step-size set in Figure 1?**
It is not clear to me whether the results presented in Figure 1 use the same step-size for both Adam and Muon,
or use a tuned step-size for each optimizer. Could the authors clarify this point?

The later would seem more appropriate as it is not clear that the same step-size is optimal for both optimizers, especially when used at the same time for different weights.
Although it would then require a comparison against using Adam for both subset of weights with a different step-size for each subset to account for the possibility that the improvement comes from using different step-sizes.

---

> ### Author Response · Authors · 2025-11-20
>
> We sincerely thank the reviewer’s thorough and careful reading of our paper, as well as for many insightful suggestions. Below, we respond to the comments in Weakness (W) and Questions (Q).
>
> ---
>
> ### **W1. Distinguishing between frequency imbalance in "facts" vs. general imbalance in text**
>
> Thanks for the insights about the relationship between facts and tokens. We would like to explain the relationship between them by clarifying the three main points raised by the reviewer.
>
> >One hypothesis could be that Muon helps on additional layers for the same issue of imbalanced tokens, not specifically facts in associate memories.
>
> * (Definition of "fact") **We do not sharply separate "facts" from "tokens".** Our main claim is that Muon’s update rule better matches the outer-product structure of linear associative memories than Adam, improving learning of tail classes. Following Meng et al. (2022) and Dai et al. (2022), we call the learned parameters in these memories "facts" or "knowledge", which arise from token associations (e.g., "SpaceX"–"Elon Musk") rather than marginal token frequencies, and thus are not captured by single-token analyses such as Kunstner et al. (2024). Our knowledge acquisition task in Section 3.3 is designed to probe these associative facts (Lines 128–129).
>
> * (Relationship between fact and single token) It is somewhat imprecise to say that **individual tokens** contain knowledge or facts, since such knowledge primarily resides in the **association between tokens**. For example, to form a meaningful fact or piece of knowledge, we need to relate "Elon Musk" to "male" / "he" / "him". Thus, we focus on token--token associations, and we interpret the per-token loss curves in (Kunstner et al. 2024) as evidence that Adam can better predict less frequent tokens than SGD given the context, i.e., that it better learns the association between the target token and other tokens.
>
> >Adam cannot correct these imbalances beyond the embedding layers, while Muon could.
> * (Relationship between Muon and associative memories) Our work pinpoints why Muon optimizes beyond the embedding layer more effectively: **its update rule aligns with the outer-product structure of linear associative memories**. We distinguish (i) prompt-time associations (e.g., "sky color"–"red" in "If the color of the sky is red, then …", mainly in QK) from (ii) pretraining associations (e.g., "sky color"–"blue" in natural text, stored in VO/FFN and modeled as linear associative memories; Bietti et al., 2023; Meng et al., 2022). Our results indicate that this **second type is the main source of Muon’s superiority over Adam**, as shown in both realistic (Sec. 3.3) and simplified (Sec. 4.1) settings. We highlight alignment with associative memories as a key mechanism, without excluding other contributing factors.
>
>
> >We could also expect this imbalance from the distribution of tokens in text data, which follows Zipf's law as (Kunstner et al., 2024)
>
>
> * **Our results are aligned with Kunstner et al. (2024).** Although the seminal work of Kunstner et al. (2024) primarily focuses on heavy-tailed effects at the **single-token** level, our work shows that their main **claims and analyses about Adam extend to the learning of associative memories**. Specifically, Adam learns the tail class much better than SGD in the knowledge learning task (Figure 3), discussed in Lines 333-335 in the original version and Lines 319-321 in the revised version. Moreover, this can be theoretically proved when the embedding is support-decoupled (Theorem 4.3), which is exactly the setting considered in Kunstner et al. (2024). In the revised version (Appendix G.9), we also reproduce the Wikitext103 setting of Kunstner et al. (2024) and analyze the loss of each individual token; the results show a consistent advantage of Muon on tail tokens, in line with our overall claims.
>
> >The issue may stem from uneven token frequencies or uneven fact frequencies; though facts map to tokens, the two can still be distinguished.
>
> * **The benefit of Muon is tied to the heavy-tailedness of token associations and can change even when the single-token distribution remains close to Zipf.** Following the reviewer’s suggestion, Appendix G.8 evaluates Muon on datasets that mix the same bio-information under different levels of heavy-tailedness by explicitly varying association frequencies (e.g., co-occurrences like "Ashton Hilda Older"–"Jersey City") while leaving single-token frequencies almost unchanged. Figure 13 shows that Muon’s gain over Adam decreases as the mixture becomes more uniform. The mixed datasets still (i) slightly change the single-token distributions (facts and tokens are not perfectly decoupled; Fig. 14(a)) and (ii) yield token distributions that approximately follow Zipf’s law (Fig. 14(b)). Thus, Muon’s advantage depends on the heavy-tailedness of associations rather than only on the marginal token distribution.

---

> ### Author Response · Authors · 2025-11-20
>
> ### **W2. Presentation of the results**
>
> Thanks for the kind presentation suggestion.
>
> ---
>
> **Figure 1:** To make the main conclusion clearer, we have followed your suggestion and added bar plots summarizing the final validation loss at 10,000 steps in Figures 1\(c\)  and 1(d). Figure 1 now exclusively shows results for the non-gated FFN model. All experiments for the gated FFN variant have been moved to a dedicated section in the Appendix G.3.
>
> ---
>
> **Figure 2:** All experiments for the gated FFN variant have been moved to a dedicated section in Appendix G.3. The SVD spectrum of QK will be explained in our response to your comment Q1.
>
> ---
>
> **Figure 4 (b, c):** We have flipped the x-axis in Figures 4(b) and 4\(c\), and added an explanation of the step size in the captions. We use different step sizes to obtain different levels of population loss and $\Delta(W)$.
>
> ---
>
> **Theoretical results in Section 4.2:** To make the presentation clearer, we add a paragraph after Assumption 4.2 as follows. "Throughout Section 4 we will also refer to the **imbalance ratio**, defined as the ratio between the minimal and maximal frequencies of triplets, i.e., $r = \frac{\min_{k \in [K]} p_k}{\max_{k \in [K]} p_k} \in (0, 1].$ Under Assumption 4.2 with parameters $\alpha$ and $\beta = L/K$, this reduces to $r(\alpha, \beta) = \min \{ \frac{\alpha(1-\beta)}{\beta(1-\alpha)}, \frac{\beta(1-\alpha)}{\alpha(1-\beta)} \}$. We keep the two-mass $(\alpha, \beta)$ parametrization because it allows us to write the gradient and its SVD in closed form while capturing the same dependence on class imbalance as using $r$ directly; the multiclass case follows from the same SVD calculation."
>
> ---
>
>
>
> **Could some of the notation be simplified by assuming that the frequencies are sorted?**
> The min/max in equation (4.1) are taken over the model's predicted correct-class probabilities $[f_{W_\eta}(E_k)]_k$; they are not the sampling probabilities $\{p_k\}$.
>
> ---
>
> **The description of the stopping condition needs a slightly more wordy explanation.**
> To make the presentation clearer, we add a sentence before the equation (4.1) as follows. " For each optimizer, we choose a step size $\eta$ so that some class already attains correct-class probability at least $1-\epsilon$ after one update, and then we report the smallest correct-class probability across classes at the same $\eta$. Equation (4.1) formalizes this procedure."
>
> ---
>
> ### **W3. Additional References**
>
> Thank you for your valuable comments and for bringing these two relevant papers to our attention. We have now added both works to our Related Works section as follows.
>
> For instance, Zhao et al. 2024 and Zhang et al. 2024 conduct detailed ablations on Adam's hyperparameters and components, identifying that its benefits are particularly pronounced for the first and last embedding layers of language models.
>
> ---
>
> ### **W4. Minors/typos**
>
> Thank you for pointing out these typos.
>
> **"intrinsically heavy-tailed":** We changed it to "... which are intrinsically heavy-tailed:  a few 'head' classes are extremely frequent, while a vast number of 'tail' classes are individually rare."
>
> **It is not clear where Theorem 4.3 ends.** We replace "For GD, ... For Muon, ... For Adam, ..." in an itemize block in Theorem 4.3.
>
> **"highly sensitive" as a description of Adam's behavior in Figure 2 seems exaggerated.** We changed "highly sensitive" to "more sensitive".

---

> ### Author Response · Authors · 2025-11-20
>
> ### **Q1. Why does Muon help more on the VO/FFN than on the QK weights?**
>
> Thanks for this insightful question about the difference between QK and the associative memory parameters, i.e., VO and FFN. We would like to clarify this by responding to the following concerns.
>
>
> >The QK weights are also part of the attention mechanism and form an outer-product based on the representation of the tokens. Why would the associative-memory explanation not apply to them as well?_
>
> * **QK parameters cannot be straightforwardly modeled as associative memories due to positional embeddings.** Intuitively, QK parameters in attention capture relationships between tokens. Most pretrained LLMs, such as the LLaMA and Qwen series, adopt positional embeddings (e.g., RoPE) that bias attention toward nearby tokens. **We conjugate these positional embeddings break the clean associative-memory structure of the QK gradient** assumed in existing theory papers (Zhang, Frei, and Bartlett, 2023) and make the 'ground-truth' form of the QK parameters difficult to analyze, even in single-layer models.
>
>
>
>
>
> >The FFN weights do not seem to follow this outer-product structure, so why is the associate-memory explanation relevant for them?
>
> >It might be that the associative-memory explanation is most relevant for the VO weights and do not explain the benefits on FFN weights or the reduced benefits on QK weights. But if that is the case, this should be acknowledged more explicitly in the submission.
>
> * **Existing work on knowledge editing shows that FFNs can be modeled as linear associative memories that store knowledge and facts (Meng et al., 2022; Fang et al., 2024).** These studies demonstrate, in a *causal* sense, that FFNs function as linear associative memories: they explicitly extract key and value vectors from pretrained models and modify the stored knowledge (i.e., the value vectors) via **linear regression**. These findings support our modeling choice of treating FFNs as linear associative memories. We have added a clarification of this point in Lines 137–139 of the revised version.
>
>
> "In fact, they demonstrate that we can manually update the knowledge in Large Language Models (LLM)s using least squares on the FFN parameters (Meng et al., 2022a;b; Fang et al., 2024)."
>
>
> >Beyond this theoretical explanations, it would be helpful for future work if the authors could look at the experiments and data they have collected to see if there is some empirical difference between the QK weights and the VO/FFN weights that could explain the different behavior, for example a different spread in the spectrum of the singular values of the inputs of each layer?
>
>
> * **Although QK parameters learned by Muon are more isotropic than those learned by Adam, Muon does not perform substantially better than Adam on QK-related heavy-tailed tasks,** as discussed in Lines 309–311 of the original version and Lines 297–299 of the revised version. However, **this does not translate into a significant advantage for Muon on QK-related heavy-tailed tasks**. In Appendix G.10 of the revised version, we train LLMs with Muon and Adam on an in-context linear regression task. The LLM is asked to infer a vector $w$ for a query $x_{\text{query}}$ from demonstrations $\{(x_i, y_i)\}_{i=1}^{K}$ in the prompt, where $y_i = w^\top x_i$. The value of $w$ varies across prompts, and we assign different frequencies to $w$ in different subspaces. Intuitively, the LLM must rely on QK parameters to extract $w$ from the demonstrations. However, our results in Figure 17 show that Muon **does not yield a clear improvement in tail-class learning for this task**. Our conjecture is that the positional embedding breaks the associative memory structure in QK parameter, and we leave the verification of this as our future work.
>
>
> ---
>
> ### **Q2. Why the focus on Gated vs. Ungated FFN?**
>
> Thanks for the question about the experimental setup. There is no prior reason to believe that Muon would behave differently on gated versus non-gated LLMs. We include both to demonstrate that **our conclusions hold across a diverse set of LLM architectures**: Llama-2, Llama-3, and Qwen-2 adopt gated FFNs, while GPT-2 uses a non-gated FFN.
>
> ---
>
> ### **Q3. How is the step-size set in Figure 1?**
>
> Thanks for the question about the experimental setup. We set the learning rates (LRs) for Adam and Muon via **grid search**, as detailed in Appendix F. 1. Specifically, we first search across a grid for the best LR for Adam and then do the same for the LR for Muon based on this value. Since the original implementations of Adam and Muon use a single LR for all modules, we follow this practice: in Section 4.1, we do not tune separate LRs for different subsets of weights, and we reuse the LRs found for the full-Adam and full-Muon settings. This makes our protocol representative of common practice.

---

> ### Author Response · Authors · 2025-11-20
>
> We thank the reviewer again for reading our response. We appreciate it if the reviewer could consider raising the score, if our rebuttal has addressed your concerns.
> ___
> **Reference**
>
> Meng K, Bau D, Andonian A, et al. Locating and editing factual associations in gpt[J]. Advances in neural information processing systems, 2022, 35: 17359-17372.
>
> Dai D, Dong L, Hao Y, et al. Knowledge neurons in pretrained transformers[C]//Proceedings of the 60th Annual Meeting of the Association for Computational Linguistics (Volume 1: Long Papers). 2022: 8493-8502.
>
> Kunstner F, Milligan A, Yadav R, et al. Heavy-tailed class imbalance and why adam outperforms gradient descent on language models[J]. Advances in Neural Information Processing Systems, 2024, 37: 30106-30148.
>
>
> Bietti A, Cabannes V, Bouchacourt D, et al. Birth of a transformer: A memory viewpoint[J]. Advances in Neural Information Processing Systems, 2023, 36: 1560-1588.
>
> Zhang R, Frei S, Bartlett P L. Trained transformers learn linear models in-context[J]. Journal of Machine Learning Research, 2024, 25(49): 1-55.
>
> Fang J, Jiang H, Wang K, et al. Alphaedit: Null-space constrained knowledge editing for language models[J]. arXiv preprint arXiv:2410.02355, 2024.

---

> ### Author Response · Authors · 2025-11-27
>
> Dear Reviewer,
>
> As the discussion period is approaching its end, **with less than a week remaining**, we would like to ensure that we have fully addressed all of your concerns. If you have any additional comments or suggestions, please feel free to share them. We are happy to clarify any remaining points and further improve our work.
>
> Thank you so much for your time and effort in reviewing our work.

---

### Author Response · Authors · 2025-11-20
**Summary of our main revisions**

Dear AC and Reviewers,

We sincerely thank you for the time and effort you have devoted to reviewing our work. We are very grateful for your insightful comments, which have helped us further improve the quality and clarity of the paper. In response to the reviewers’ constructive feedback, we have made the following main revisions:

- **In response to Reviewer E1d8**'s suggestion about writing, all experiments for the gated FFN variant have been moved to a dedicated section in Appendix G.3.

- **In response to Reviewer E1d8 Weakness 1** concerning the distinction between frequency imbalance in "facts" and general imbalance in text. Appendix G.8 now evaluates Muon on datasets that contain the same bio-information under different degrees of heavy-tailedness, by explicitly varying association frequencies while keeping single-token frequencies distributed according to Zipf’s law. These results indicate that the heavy-tailedness of facts can be **approximately independent of the tokens’ Zipf marginals** and instead depends on associations, i.e., co-occurrences, of tokens.

- **In response to Reviewer W4iz Weakness 1** which raises the concern that the task relies heavily on the QK parameters. Appendix G.10 of the revised version now includes experiments where we train LLMs with Muon and Adam on an in-context linear regression task that primarily depends on learning the QK parameters well. Their similar performance on this task indicates that **they behave similarly when optimizing QK parameters**, consistent with our conclusion.

- **In response to Reviewer Yhie Weakness 3** regarding the point that our heavy-tail tasks focus primarily on knowledge-acquisition tasks. We have added a new experiment on the WikiText-103 language modeling benchmark in Appendix G.9, whose token frequencies follow a heavy-tailed distribution. The benefit of Muon over Adam on the tail classes in this task further verifies that our conclusions hold beyond the knowledge-acquisition setting and **extend to general text**.

- **In response to Reviewer sugx Weakness 4 and Question 1** concerning FFN over QK may be the result of parameter counting. We have added a new analysis in Appendix G.2 that explicitly controls for the number of parameters when comparing the influence of FFN and QK. The gain of Muon over Adam, normalized by parameter count, indicates that the greater importance of FFN over QK does not stem from their larger parameter size **but from their inherent structure**.

We greatly value all of the reviewers’ comments and sincerely appreciate how they have helped us strengthen the paper. We hope that these revisions satisfactorily address the concerns raised.

Sincerely,
The Authors

---

### Author Response · Authors · 2025-12-01
**Summary of Rebuttal and Discussion (Part III)**

### **Reviewer sugx's main concerns and our responses are summarized as follows.**

* **The relationship between the heavy-tailedness of data and the heavy-tailedness of parameter updates in real LLMs (W1 & Q2):** In Appendix K of the revised version, we discuss how **our formally proven insights for the one-layer model in Section 4 generalize to multi-layer LLMs**. Specifically, this generalization relies on two facts: (1) the gradient induced by the $k$-th token in the alphabet, $\nabla_W \log  [f_{W}(E_{k})]_ {k}$, is a sum of outer products, which aligns with the SVD-based structure exploited by Muon; and (2) the heavy-tailed data distribution $p_k$ enters the gradient as   $\nabla_W \mathcal{L}(W)=  -\sum_{k=1}^{K}p_{k} \nabla_W \log [f_{W}(E_{k})]_{k}$, so the gradient itself is also heavy-tailed.


* **The justification of $W_V$ as associative memory (W2):** In Lines 138–141 of the revised version, we add references to existing work showing that $W_V$ and $W_O$ **play similar roles** and can be analyzed jointly in practice, including in multi-query attention (MQA) and grouped-query attention (GQA) settings.

* **Novelty of our observation 2 (W3):** Our results reveal the **instability of Adam with respect to random initialization** and, moreover, our experiments **precisely localize and functionally disentangle the parameters** that benefit most from Muon. These are not covered by the existing works.

* **FFN over QK may be the result of parameter counting (W4&Q1):** In Figure 5 (Appendix G.2), we add two additional results showing that **the parameter-count–normalized benefit of FFN is much larger than that of QK**. Specifically, we (i) report the parameter-count–normalized gains when all FFN and all QK parameters are optimized by Muon, and (ii) compare the gains when Muon is applied to equally sized subsets of FFN and QK parameters.

* **Released code does not include the model file (W5):** We have now **included all files**, including the model file, in the newly submitted supplementary materials.

* **Logic gap on associative-memory mechanism (Q3):** In Lines 236–244 of the revised version, we clarify the logic underlying our associative-memory mechanism. Concretely, our work adopts the **hypothetico-deductive method**: we formulate the mechanism as a hypothesis, derive testable predictions, and then verify these predictions experimentally. The experiments in Sections 3.2 and 3.3 confirm the deduced consequences of the proposed mechanism at both the parameter level and the end-to-end performance level.

 **Dicussion results**:
* **Reviewer sugx claims that we have addressed most of their concerns**, writing:
    >I can confirm my concerns W2, W4 & Q1, Q3 are resolved. I still want to discuss about W1&Q2, W5.
* For W5, we reorganized the code folders to provide a runnable version for our **Round 2** discussion with the reviewer. Since then, the reviewer has not raised this concern again.

* For W1 & Q2, we discuss how our intuitions generalize to multi-layer LLMs. Because of the notational complexity of multi-layer architectures, we were actively clarifying these points with Reviewer sugx. However, due to the ICLR leakage issue, **this discussion has been interrupted**.

---

### Author Response · Authors · 2025-12-01
**Summary of Rebuttal and Discussion (Part II)**

### **Reviewer W4iz's main concerns and our responses are summarized as follows.**

* **Additional experiment on a QK-heavy task (W1):** We addressed this comment by adding an in-context linear regression experiment in Appendix G.10, which relies heavily on QK parameters to extract information. The results show that **Adam and Muon achieve very similar performance on both head and tail classes of this task**, reinforcing our claim that the primary benefits of Muon do not stem from QK parameters.
* **Main ablation (VO+FFN) at a larger scale (W2):** We clarified that our work includes experiments on 0.7B models (Appendix G.4), which display the exact same pattern as the 160M model.
* **Intuition on diminishing returns with scale (Q1):** We explained that while our work focuses on principles that should apply across scales, **practical performance at large scales depends crucially on hyperparameter tuning protocols**. We cited some works (e.g., Wen et al. vs. Liu et al.) to illustrate that observed diminishing returns may stem from tuning differences, noting that a systematic study of these protocols is orthogonal to our current focus.

* **Varying Newton-Schulz coefficients/iterations (Q2):** Our experiments apply **both the standard Newton-Schulz implementation and exact SVD**, with both approaches consistently supporting our conclusions on associative memory. We have added relevant references regarding Newton-Schulz to the related work section (Lines 751-753), though we consider the specific influence of coefficients to be orthogonal to our main contribution.

 **Discussion Results:**

* Reviewer W4iz is **positive toward our work** and notes that our additional ablation results provide a cleaner view, saying:
> I think the ablation study is useful and clearly executed, and while the associative-memory interpretation feels somewhat narrow, it is reasonable within the scope of the work. I'd lean towards acceptance.
* For the remained concerns, i.e., concerns about scaling (Q1) and NS implementation (Q2), the reviewer acknowledges that resolving them is **potentially beyond or orthogonal to the scope of our work**, saying:
> but I understand if increasing the scale is beyond the scope of your current submission.

  > But this is not concerning any issue I have with your current paper.

---

### **Reviewer Yhie's main concerns and our responses are summarized as follows.**
* **Novelty regarding isotropy observations (Liu et al., 2025) (W1):** Our results reveal the **instability of Adam with respect to random initialization** and, moreover, our experiments **precisely localize and functionally disentangle the parameters** that benefit most from Muon. These are not covered by the existing works.

* **Causal link between Muon's effectiveness on VO+FFN and associative memory(W2):** Our response clarifies that our main contribution is precisely this causal explanation: because Muon **normalizes the amplitude of each outer product in the gradient SVD**, it ensures that the linear associative memories learn all stored facts at a similar pace. We verify this relationship empirically by showing that Muon induces a more isotropic singular value spectrum in the VO/FFN parameters (Section 3.2), which in turn leads to more balanced learning across head and tail classes (Section 3.3).

* **Whether Muon's heavy-tail ability relates to its effectiveness on VO+FFN (W2):** **We answer this question affirmatively with two pieces of evidence**. First, on the knowledge acquisition task (Section 3.3), whose success relies on associative-memory learning, whether the VO and FFN parameters are optimized by Muon has a strong impact on performance. Second, on the in-context linear regression task (Appendix G.10), whose success relies on QK parameter learning, Muon and Adam achieve similar performance, even if the data is heavy-tailed.

* **Muon might be restricted to specific tasks; more experiments on various tasks(W3):** The additional experiment on WikiText-103 (Appendix G.9) **verifies that our conclusions extend to general tasks**. The results on this real-world dataset closely mirror our synthetic findings: Muon is stable and significantly outperforms Adam on tail-token groups, while the Muon(VO+FFN) hybrid matches the performance of the full model, confirming that our theory holds in realistic settings.

* **The toy thoery seems to intuitive (W4):** We respectfully argue that **intuitiveness is a desirable property** of a good theory, **not a weakness**. We also emphasize that our contribution goes beyond intuition: (i) we provide the **first quantitative, constructive proof of why and how Adam fails** in an offline optimization setting; and (ii) we present, to our knowledge, the first analysis of Muon in a **non-convex, logistic-regression-style associative-memory model** for LLM knowledge recall.

**Discussion results:**

* Despite our active responses to the raised concerns, the reviewer **did not engage in the discussion**.

---

### Author Response · Authors · 2025-12-01
**Summary of Rebuttal and Discussion (Part I)**

Dear AC,

We sincerely thank you for the time and effort you have devoted to reviewing our work. We would like to summarize our discussion with reviewers.

### **Reviewer E1d8's main concerns and our responses are summarized as follows.**

Reviewer E1d8 is satisfied with our main experimental and theoretical results and agrees that our work provides strong evidence for our central claim. In the review, the reviewer writes:

> I am generally positive about the submission. My initial score for the review is low due to the issues listed below. Those should be addressable with some revisions to the presentation and an additional experiment. I am expecting to increase my score if that can be added.



To address the reviewer's concerns, we have made the following revisions:

* **The distinction between frequency imbalance in "facts" and general imbalance in text (W1):** To address this concern, Appendix G.8 now evaluates Muon on datasets that contain the same bio-information under different degrees of heavy-tailedness, by explicitly varying association frequencies while keeping single-token frequencies distributed according to Zipf’s law. These results indicate that the heavy-tailedness of facts can be **approximately independent of the tokens’ Zipf marginals** and instead depends on associations, i.e., co-occurrences, of tokens.
* **Presentation (W2):** We have followed reviewer’s suggestion including: i) Add bar plots summarizing the final validation loss at 10,000 steps in Figures 1; ii) All experiments for the gated FFN variant have been moved to a dedicated section in Appendix G.3. iii) Flip the x-axis in Figure 4 and add an explanation of the step size. iv) Introduce the notation $r=\frac{\min_{k\in[K]} p_k}{\max_{k\in[K]} p_k}$ throughout Section 4 as suggested by the reviewer to show the imbalance ratio more clear.
* **Why does Muon help more on the VO/FFN than on the QK weights?(Q1):**  We clarify the distinction between parameter types by explaining that **QK weights are difficult to model as associative memories** because positional embeddings (e.g., RoPE) introduce biases that disrupt the clean gradient structure assumed in existing theory. In contrast, we justify **modeling FFNs as linear associative memories** as prior research that demonstrates that FFNs function as key-value stores for knowledge that can be updated via linear regression. Empirically, we further support this by showing that while Muon improves the isotropy of QK parameters, it **does not significantly outperform Adam on QK-dominated tasks** (such as in-context linear regression task in Appendix G. 10), reinforcing the hypothesis that positional embeddings limit the applicability of the associative memory framework to QK weights.
* **Additional References and typos (W3&4) and experimental setting explanation (Q2&3):** We modified the manuscript according to the reviewer's suggestion.

**Discussion results:**

* Despite our active responses to the raised concerns, the reviewer **did not engage in the discussion**.

---

### Meta-Review · Area_Chair_qZct · 2026-01-21

**Summary:**

This paper analyzes why the Muon optimizer outperforms Adam in transformers, showing that the gains arise primarily from optimizing associative-memory components (VO and FFNs) rather than QK weights. The authors connect Muon’s updates to the outer-product structure of associative memories and improved tail-class learning under heavy-tailed data. Reviewers were generally positive about the core empirical and theoretical results. The rebuttal added controlled experiments disentangling fact imbalance from token imbalance, clarified why QK behaves differently, and strengthened the empirical link between isotropy and tail-class performance, resolving the reviewer’s main concerns.

**Reviewer Concerns:**

Resolved: control exp on different frequencies, QK-heavy experiment, presentation

No significant outstanding issue

**Reviewer Scores:**

Most reviewers should raise the scores by 1 to 2 after rebuttal

---

### Decision · Program_Chairs · 2026-01-26

Accept (Poster)